# Evidence of the interplay of genetics and culture in Ethiopia

Saioa López [1,2,11✉], Ayele Tarekegn[3,11✉], Gavin Band [4], Lucy van Dorp [1,2], Nancy Bird [1,2], Sam Morris[1,2], Tamiru Oljira [5], Ephrem Mekonnen [6], Endashaw Bekele[7], Roger Blench[8,9], Mark G. Thomas [1,2], Neil Bradman[10] & Garrett Hellenthal [1,2✉]

The rich linguistic, ethnic and cultural diversity of Ethiopia provides an unprecedented opportunity to understand the level to which cultural factors correlate with–and shape–genetic structure in human populations. Using primarily new genetic variation data covering 1,214 Ethiopians representing 68 different ethnic groups, together with information on individuals' birthplaces, linguistic/religious practices and 31 cultural practices, we disentangle the effects of geographic distance, elevation, and social factors on the genetic structure of Ethiopians today. We provide evidence of associations between social behaviours and genetic differences among present-day peoples. We show that genetic similarity is broadly associated with linguistic affiliation, but also identify pronounced genetic similarity among groups from disparate language classifications that may in part be attributable to recent intermixing. We also illustrate how groups reporting the same culture traits are more genetically similar on average and show evidence of recent intermixing, suggesting that shared cultural traits may promote admixture. In addition to providing insights into the genetic structure and history of Ethiopia, we identify the most important cultural and geographic predictors of genetic differentiation and provide a resource for designing sampling protocols for future genetic studies involving Ethiopians.

[1] Research Department of Genetics, Evolution & Environment, University College London, London, UK. [2] UCL Genetics Institute, University College London, London, UK. [3] Department of Archaeology and Heritage Management, College of Social Sciences, Addis Ababa University, New Classrooms (NCR) Building, Second Floor, Office No. 214, Addis Ababa University, Addis Ababa, Ethiopia. [4] Wellcome Centre for Human Genetics, University of Oxford, Oxford, UK. [5] Genomics & Bioinformatics Research Directorate (GBRD), Ethiopian Biotechnology Institute (EBTi), Addis Ababa, Ethiopia. [6] Institute of Biotechnology, Addis Ababa University, Addis Ababa, Ethiopia. [7] College of Natural and Computational Sciences, Addis Ababa University, Addis Ababa, Ethiopia. [8] McDonald Institute for Archaeological Research, University of Cambridge, Cambridge, UK. [9] Department of History, University of Jos, Jos, Nigeria. [10] Henry Stewart Group, London, UK. [11]These authors contributed equally: Saioa López, Ayele Tarekegn. ✉email: saioa.lopez@ucl.ac.uk; ayele.tarekegn@aau.edu.et; g.hellenthal@ucl.ac.uk

Ethiopia is one of the world's most ethnically and culturally diverse countries, with over 70 different languages spoken across more than 80 distinct ethnicities (www.ethnologue.com). Its geographic position and history (briefly summarised in Supplementary Note 1) motivated geneticists to use blood groups and other classical markers to study human genetic variation[1,2]. More recently, the analysis of genomic variation in the peoples of Ethiopia has been used, together with information from other sources, to test hypotheses on possible migration routes at both 'Out of Africa' and more recent "Migration into Africa" timescales[3,4]. The high genetic diversity in Ethiopians facilitates the identification of novel variants, and this has led to the inclusion of Ethiopian data in studies on the genetics of elite athletes[5–7], adaptation to living at high elevation[8–11], milk drinking[12–14], tuberculosis[15,16], and drug metabolising enzymes[17–19].

While the relationships of Beta Israel with other Jewish communities have been the subject of focused research following their migration to Israel[20–22], studies involving genomic analyses of the history of wider sets of Ethiopian groups have been more limited[23,24]. Although as early as 1988 Cavalli-Sforza et al.[25], drew attention to the importance of bringing together genetic, archaeological and linguistic data, there have been few attempts to systematically do so in studies of Ethiopia[4,26–31]. Generally, studies have been limited to analysing data from single autosomal loci, non-recombining portion of the Y chromosome and mitochondrial DNA[24,26,32–34] and/or relatively few ethnic groups[3,27,30,31,35,36], which has limited the inferences that can be drawn. Furthermore, hitherto there has been little exploration of how genetic similarity is associated with shared cultural practices (see however van Dorp et al.[37]) despite the considerable variation known to exist in cultural practices, particularly in the southern part of the country (The Council of Nationalities, Southern Nations and Peoples Region, 2017[38]). For example, Ethiopian ethnic groups have a diverse range of religions, social structures and marriage customs, which may impact which groups intermix, and hence provide an on-going case study of socio-cultural selection[39,40], The Council of Nationalities, Southern Nations and Peoples Region, (2017)[38] that can be explored using DNA.

Here we analyse autosomal genetic variation data at 534,915 single nucleotide polymorphisms (SNPs) in 1214 Ethiopian individuals that include 1082 previously unpublished samples and 132 samples from Lazaridis et al.[41], Gurdasani et al.[42], and Mallick et al.[28,41,42]. Our study includes people from 68 distinct self-reported ethnicities (8–73 individuals per ethnic group) that comprise representatives of most of the major language groups spoken in Ethiopia, including Nilo-Saharan (NS) speakers and three branches (Cushitic, Omotic, Semitic) of Afroasiatic (AA) speakers, as well as languages of currently uncertain classification (Chabu, and the speculated, possibly extinct language of the Negede-Woyto) (www.ethnologue.com) (Fig. 1a, Supplementary Fig. 1, Supplementary Data 1, 2, Supplementary Note 2). Newly genotyped individuals were selected from a larger collection on the basis that their self-reported ethnicity, and typically birthplace, matched that of their parents, maternal grandmother, paternal grandfather, and any other grandparents recorded, analogous to recent studies of population structure in Europe[43,44]. For these individuals we also recorded their self-reported religious affiliation (four categories), first language (66 total classifications) and/or second language (40 total classifications) (Supplementary Data 1). Furthermore, some of the authors of this study (A.T., N.B.) translated into English and edited a compendium (originally published in Amharic) that documented the oral traditions and cultural practices of 56 ethnic groups of the Southern Nations, Nationalities and Peoples' Region (SNNPR) of Ethiopia through interviews with members of different ethnic groups (The Council of Nationalities, Southern Nations and Peoples Region, 2017)[38]. From this new resource, we compiled a list of 31 practices that were reported as cultural descriptors by members of 47 different ethnic groups out of the 68 in this study (see "Methods"). These practices include self-declared cultural practices such as male and female circumcision, and 29 different pre-marital and marriage customs, including arranged marriages, polygamy, gifts of beads or belts, and covering the bride in butter.

We compared SNP patterns in each present-day Ethiopian to those in all other present-day Ethiopians and to the 4500 year-old Ethiopian sample "Mota", a forager from southern Ethiopia that

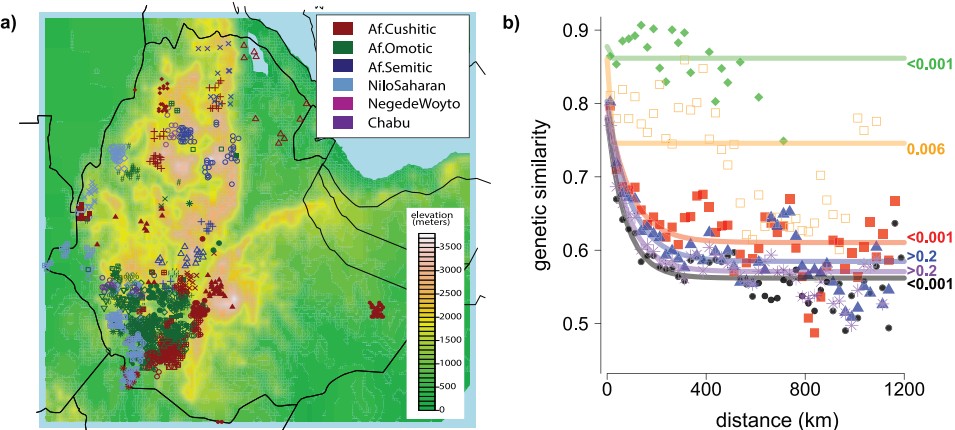

**Fig. 1 Genetic similarity decays with spatial distance among Ethiopians and correlates with shared reported ethnicity and language. a** Locations of sampled Ethiopians based on birthplace (in some cases slightly moved due to overlap), with landscape colours showing elevation and coloured symbols depicting the language category (plus unclassified languages Negede Woyto, Chabu) of each individual's ethnic identity. The legend for the symbols is provided in Supplementary Fig. 1. **b** Fitted model for genetic similarity (1-TVD; under the "Ethiopia-internal" analysis) between pairs of individuals versus geographic distance, with points depicting the average genetic similarity within 25 km bins, for all individuals (black; dots) or restricting to individuals who report the same group label (green; diamonds), same first language (orange; open squares), same second language (blue; triangles), same religious affiliation (purple; asterisks), or whose reported ethnicities are from the same language group (red; closed squares). Labels at right give permutation-based p values when testing the null hypothesis of no increase in genetic similarity among individuals sharing the given trait (see "Methods").

represents the only presently available ancient genome from the country[4]. We also compared them to a further 16 labelled groups comprised of 39 ancient individuals[36,45–53] (Supplementary Table 1) and 264 present-day non-Ethiopian groups[28,41,42,51,54,55] comprised of 2678 individuals (average group sample size = 10, range: 1–100), including 106 unpublished samples from nine groups (Supplementary Data 3). We focus on inferring patterns of haplotype sharing among individuals, which has increased resolution over commonly-used allele-frequency based techniques[56,57] when identifying latent population structure and inferring the ancestral history of peoples sampled from relatively small geographic regions, such as within a country[43,58,59].

Our results provide a comprehensive understanding of the relative strength to which different socio-cultural factors are associated with genetic distance in present-day Ethiopians. We provide evidence that recent intermixing is increased among groups, sometimes from distantly-related linguistic affiliations, that live nearby and/or share cultural practices. We also provide an inferred recent admixture history for members of 68 ethnic groups.

## Results

**Genetic distance is broadly associated with geography, ethnicity, linguistics and shared culture in Ethiopia.** Principal components analysis (PCA)[57,60] applied to sampled African individuals revealed Ethiopians to be more genetically similar to each other and sampled groups from other east African countries (Kenya, Somalia, Sudan, Tanzania) than to other African populations (Supplementary Fig. 2b). Runs-of-homozygosity[61] and inferred proportions of genome that are identical-by-descent (IBD)[62] among individuals of the same ethnicity vary substantially across Ethiopian groups (Supplementary Fig. 3a, b). Ethiopia's two largest ethnic groups, Amhara and Oromo, have the lowest levels of within-group IBD-sharing (Supplementary Fig. 3a), and we observe a significant ($p$ val < 0.001) decrease of homozygosity with increasing population census size across ethnic groups in the SNNPR (Supplementary Fig. 3c; census from 2007: The Council of Nationalities, Southern Nations and Peoples Region, 2017[38]).

To measure genetic similarity between pairs of individuals, we calculated the total variation distance (TVD)[43] between their haplotype-sharing patterns inferred by CHROMOPAINTER[63] (see "Materials and Methods"). Mimicking van Dorp et al.[37], we performed two CHROMOPAINTER analyses in order to infer the broad time periods over which lines of ancestry between individuals diverged (see schematic of approach in Supplementary Fig. 4). The first, which we call "Ethiopia-internal," compares haplotype patterns in each Ethiopian to those in all other sampled individuals. TVD based on this analysis can be thought of as a haplotype-based analogue of the commonly-used $F_{ST}$[64] genetic distance measure, and the two are correlated in our analyses (Pearson's $r = 0.63$; Mantel-test $p$ value < 0.00001). However, TVD estimates have been shown to be more powerful at distinguishing subtle genetic differences among e.g., African groups[65]. The second, which we call "Ethiopia-external," instead compares patterns in each Ethiopian only to those among individuals in non-Ethiopian groups. As the "Ethiopia-internal" analysis compares haplotype patterns in each Ethiopian to those in other Ethiopians, including other members of the same ethnic group, it is more sensitive for detecting endogamy effects and admixture among Ethiopian groups[37,59]. In contrast, the "Ethiopia-external" analysis mitigates signals related to both of these factors, while remaining sensitive for inferring whether Ethiopians having varying proportions of ancestry related to non-Ethiopian sources due to e.g., different admixture histories[43]. We

illustrate this in simulations mimicking our real data (Supplementary Fig. 5, Supplementary Note 3).

We first considered how pairwise genetic similarity among Ethiopians is related to several factors. Under both the "Ethiopia-internal" and "Ethiopia-external" analyses, we found significant associations ($p$ val < 0.05) between genetic distance and each of geographic distance, elevation difference, ethnicity and first language, after controlling each factor for the others where possible (Fig. 1b, Supplementary Figs. 6, 7, Supplementary Tables 2–6). In contrast, we found no significant association ($p$ val > 0.2) between genetic distance and each of religion and second language (Fig. 1b, Supplementary Fig. 6, Supplementary Tables 2–6). However, within six of 16 groups for which we sampled at least five individuals from different religions, we found some nominal evidence (permutation-based $p$ val < 0.05) of genetic isolation between people reporting as Christians versus those reporting as Muslims or those reporting as practicing traditional religions (Supplementary Table 7).

We next averaged pairwise genetic similarity values among individuals from the same versus different group labels (Supplementary Fig. 8). Consistent with the relationships depicted by PCA (Supplementary Fig. 2), on average Ethiopian groups are more genetically similar to other Ethiopian groups than they are to the non-Ethiopian groups included in this study (Supplementary Fig. 8, Supplementary Data 5, 6). We found a significant association between genetic similarity and reporting shared cultural traits among SNNPR groups under the "Ethiopia-internal" analysis (Mantel-test $p$ value < 0.03), which remained after accounting for geographic or elevation distance (partial Mantel-test $p$ value < 0.05) or language group (partial Mantel-test $p$ value < 0.03) (Supplementary Table 8).

To facilitate comparisons of genetic patterns among groups, we generated an interactive map that graphically displays the genetic similarity among groups under each of the "Ethiopia-internal" and "Ethiopia-external" analyses (https://www.well.ox.ac.uk/~gav/projects/ethiopia/), with averages summarised in Supplementary Fig. 8 and Supplementary Data 5, 6. As examples, we provide three observations based on these findings. The first observation is that, under the "Ethiopia-internal" analysis, Ari and Wolayta people who work as cultivators or weavers are more genetically similar to members of other ethnicities on average than they are to people from their own ethnicities who work as potters, blacksmiths and tanners (top left squares in Fig. 2a). This is consistent with the social marginalisation reported to be associated with occupational classes in these ethnic groups[66,67]. Despite this, under the "Ethiopia-external" analysis, Ari and Wolayta are more genetically similar to members of their own ethnicities on average, regardless of occupation (bottom right of squares in Fig. 2a). Therefore, in contrast to indications given from the "Ethiopia-internal" analysis (Supplementary Data 5) and $F_{ST}$ (Supplementary Data 11), the "Ethiopia-external" results suggest that individuals from different occupations within the same ethnic group are more recently related to each other than they are to any other ethnic group.

The second example concerns the two sampled groups in our study for which Ethnologue ascribes no linguistic classification, the Chabu and Negede-Woyto. Each are significantly differentiable ($p$ val < 0.001) from all other ethnic groups under the "Ethiopia-internal" analysis (Fig. 2b, Supplementary Fig 8a, Supplementary Data 5). The Chabu, a hunter-gatherer group and linguistic isolate, exhibit the strongest overall degree of genetic differentiation from all other ethnic groups, consistent with previous analyses highlighting their genetic distinctiveness[30,31]. However, under the "Ethiopia-external" analysis, the Chabu show similar genetic patterns to NS speaking groups, while the Negede-Woyto are not significantly distinguishable from multiple ethnic

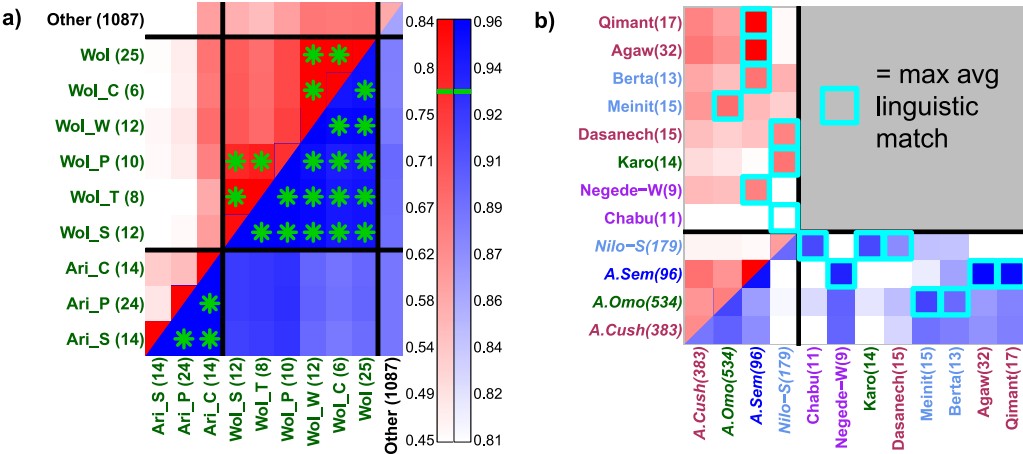

**Fig. 2 Genetic similarity suggests recent endogamy within occupational groups and shows shared ancestry among some linguistically divergent groups.** Average pairwise genetic similarity (1-TVD) between individuals from different Ethiopian labelled groups (coloured on axis by language category–see Fig. 1a), under the "Ethiopia-internal" analysis (top left, red colour scale) versus the "Ethiopia-external" analysis (bottom right, blue colour scale). **a** Genetic similarity between Ari and Wolayta (Wol) occupational groupings (C = cultivator, P = potter, S = blacksmith, T = tanner, W = weaver), with green asterisks denoting relatively high similarity above the green lines in legend at right. This illustrates how the "Ethiopia-external" analysis shows increased similarity between groups of the same ethnicity relative to that seen under the "Ethiopia-internal" analysis. **b** Average pairwise genetic similarity among individuals from four language classifications (italics), and the average genetic similarity between individuals from these four language groups and those from eight ethnic groups (non-italics). For each of the eight ethnic groups, the cyan squares denote the language group with the highest average genetic similarity to that ethnic group under each analysis. This illustrates how the linguistically-unclassified Chabu and Negede-Woyto are most genetically similar on average to Nilo-Saharan and Afroasiatic Semitic speakers, respectively, and highlights six other groups that are more genetically similar to members from a different language group than they are to members of their own language group.

groups representing all three branches of AA (Fig. 2b, Supplementary Fig 8b, Supplementary Fig. 9). The Chabu's similarity to NS speakers reflects previous findings based on genetics[30], where the Chabu were referred to as Sabue[31]), and linguistics[68,69]. However, we clarify this further by showing the Chabu to be significantly more genetically similar to the Mezhenger sample than other samples examined here (Supplementary Fig. 8a), with whom they have been suggested to share recent origins[70].

Third, we find unexpectedly high genetic similarities among groups classified into distantly related linguistic categories (Fig. 2b, Supplementary Fig. 8). For example, the AA-speaking Karo and Dasanech are on average more genetically similar to NS speakers than to other AA speakers. In contrast, the NS speaking Meinit and Berta are more similar to AA speakers. At a finer linguistic level, the AA Cushistic-speaking Agaw and Qimant are most genetically similar to sampled AA Semitic-speakers, with the Qimant and AA Semitic-speaking Beta Israel having been reported previously to be related linguistically to the Agaw[71]. These observations demonstrate that shared linguistic affiliation, even using broad categories, is not always a reliable predictor of relatively higher genetic similarity. However, on average individuals from the AA Cushitic, AA Omitic, AA Semitic, and NS classifications, as well as individuals from separate sub-branches within each of these categories, are genetically distinguishable from each other under both the "Ethiopia-internal" and "Ethiopia-external" analyses (p val < 0.001; Supplementary Note 4; Supplementary Fig. 9; Supplementary Data 9-10), consistent with Pagani et al.[27]. This suggests that speakers of the first three tiers of Ethiopian language classifications at www.ethnologue.com are genetically –distinguishable on average, and that these genetic differences are not solely attributable to endogamy effects but also to differential ancestry related to non-Ethiopians. We also find that several groups spanning the three AA classifications of Cushitic, Omotic, and Semitic show high genetic similarity to each other on average and less genetic similarity to NS speakers (Fig. 2b, Supplementary Figs. 8, 9). We find no clear genetic evidence Omotic is an outgroup to other AA

language groups, as previously claimed[29], at least among Ethiopians.

**The recent admixture history of Ethiopia.** We explore the ancestry of different Ethiopian groupings by comparing their haplotype sharing patterns under the "Ethiopia-external" analysis to those in a set of reference populations intended to reflect ancestral source populations. To do so, we first used fineSTRUCTURE[63] to assign Ethiopians into 78 clusters of relative genetic homogeneity (Supplementary Fig. 10, Supplementary Data 4). Unsurprisingly, given our previous genetic similarity results (Fig. 1b, Supplementary Fig. 6, Supplementary Fig. 8), these clusters were associated with ethnic label (Supplementary Fig. 10), with clusters inferred using the alternative approach ADMIXTURE (Alexander et al.[56]) also often categorising genomes according to ethnic group (Supplementary Fig. 11). However, using clusters rather than self-reported label can increase power to infer ancestral histories by merging ethnic groups with similar genetic variation patterns. This also can clarify ancestry inference, as it does not assume that all individuals reporting the same ethnicity share recent ancestry. We applied SOURCEFIND[72] and GLOBETROTTER[58] to infer and describe admixture events in each of these 78 clusters (see "Methods", Supplementary Note 5). Simulations mimicking patterns we observe showed that our approach accurately infers sources and dates of admixture (Supplementary Fig. 5d, Supplementary Note 3).

GLOBETROTTER infers clear admixture events in 68 of the 77 Ethiopian clusters containing more than one individual, with dates ranging from ~100 to 4200 years ago (Supplementary Note 5, Fig. 3, Supplementary Fig. 12, Supplementary Data 7). Out of 275 reference populations, SOURCEFIND infers only 13 contributed >5% towards describing ancestry patterns within any of these 68 clusters: the 4500-year-old Ethiopian Mota and 12 present-day groups from Chad, Egypt, Kenya, Saudi Arabia, Somalia, Sudan, Tanzania, Uganda and Yemen (Fig. 3, Supplementary Fig. 12, Supplementary Data 7). Strikingly, the percentage of matching to Mota decreases with increasing spatial

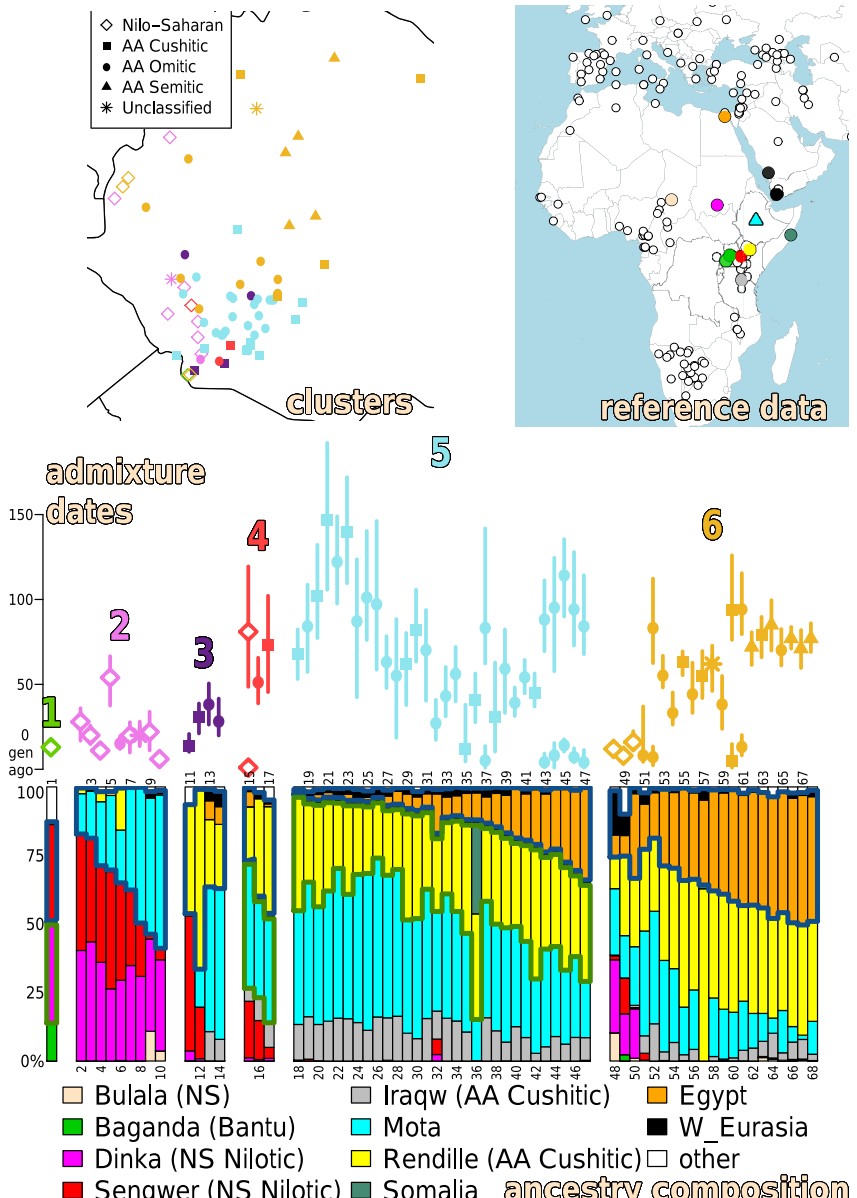

**Fig. 3 Inferred ancestral composition and recent admixture events in each Ethiopian cluster.** (top-left) FINESTRUCTURE-inferred genetically homogeneous clusters of Ethiopians, with location placed on the map by averaging the latitude/longitude of each cluster's individuals. Colours denote which of six types of admixture event (1-6 below) each cluster falls into and symbols provide the most-represented language group among individuals' ethnicities in each cluster. (top-right) A subset of the 264 non-Ethiopian present-day reference populations, plus the 4.5 kya Ethiopian Mota (Gallego-Llorente et al.[4]; cyan triangle), that DNA patterns in each Ethiopian cluster were compared to under the "Ethiopian-external" analysis. Filled circles (legend at bottom) indicate reference populations that contributed >5% of ancestry to at least one Ethiopian using SOURCEFIND. (Middle) Inferred admixture dates in generations from present (symbols give means and correspond to legend in the top-left panel, line = 95% CI, sample sizes given in Supplementary Fig. 12), coloured by the six types of admixture event. (Bottom) SOURCEFIND-inferred ancestry proportions for each Ethiopian cluster (key for numbers in Supplementary Data 7). Blue and green borders in the ancestry composition highlight different admixing sources. In particular we enclose the reference populations representing one of the inferred admixing sources with a thick blue line. In Ethiopian groups with >2 inferred sources, we also enclose the reference populations representing the second source with a thick green line. Using this information, we highlight six types of inferred admixture events among: (1) three sources related to the Baganda, Dinka and Sengwer, (2) two sources related to Mota and Dinka/Sengwer, (3) two sources related to Rendille and Mota or Sengwer, (4) three sources related to Rendille, Mota and Sengwer, (5) three sources related to Egypt/W.Eurasia, Rendille and Mota/Iraqw, and (6) two sources related to Egypt/W.Eurasia and Rendille/Mota.

(geographic and elevation) distance between where Mota was discovered and the average location of individuals in each cluster (linear regression *p* value < 0.0005, Supplementary Fig. 13, Supplementary Table 9).

We infer six broad categories of admixture, correlated with both geography and linguistics (Fig. 3, Supplementary Fig. 12). For example, 12 clusters primarily containing individuals from

NS-speaking groups (clusters 1–5, 7, 9, 10, 15, 48–50 on Fig. 3, Supplementary Fig. 12) show evidence of admixture involving a source related to Bantu (Baganda) and/or NS Nilotic (Sengwer, Dinka) speakers, with date estimates <30 generations ago in all but two of these clusters. Similar admixture is inferred in the AA Omotic speaking Karo (cluster 6), AA Cushitic speaking Dasanech (cluster 11) and linguistically-unclassified Chabu

(cluster 8), which each show relatively high genetic similarity to NS-speakers (Fig. 2a). In contrast, clusters primarily containing AA speakers, including all Ari and Woylata clusters (clusters 22, 24, 25, 39, 41, 43, 45, 54, 56) and a cluster containing the linguistically-unclassified Negede-Woyto (cluster 58), typically show evidence of admixture between two or more sources related to the 4.5 kya Ethiopian Mota, Cushitic-speaking Rendille from Kenya and Egypt/W.Eurasian groups, over a broader range of dates (5–147 generations ago). Among these, five northern clusters containing AA Semitic-speakers and the AA Cushitic-speaking Agaw (clusters 62, 64, 66–68), plus two geographically nearby clusters containing the AA Cushitic-speaking Qimant (cluster 63) and AA Omotic-speaking Shinasha (cluster 65), show the highest amounts of Egypt-like ancestry in our dataset and similar admixture dates (point estimates 71–85 generations ago).

**Pervasive recent intermixing among groups is associated with geographic proximity and shared cultural practices.** The surprising genetic similarity among people speaking dissimilar languages may be attributable in part to relatively recent language adoption and/or high levels of recent intermixing among distinct Ethiopian groups. To test for the latter, we also applied GLOBETROTTER to each of the 77 Ethiopian clusters under the "Ethiopia-internal" analysis, which includes Ethiopians as surrogates for admixing sources and hence can characterize intermixing that has occurred among Ethiopian groups. GLOBETROTTER found evidence of admixture in 61 clusters in this analysis, 46 (75.4%) of which had estimated dates <30 generations ago (<900 years ago) (Supplementary Data 8). Across clusters, inferred dates under the "Ethiopia-internal" analysis typically are more recent than those inferred under the "Ethiopia-external"

analysis (Fig. 4a). This indicates that the "Ethiopia-internal" analysis captures recent intermixing among Ethiopian groups that is missed under the "Ethiopia-external" analysis; otherwise dates under the two analyses would be similar. Furthermore, we inferred that recent intermixing occurred more frequently than expected among clusters whose individuals reside geographically near to each other ($p$ value < 0.00002, Fig. 4b).

We next explored whether groups that share cultural practices also show evidence of recent intermixing. Supporting this, we found a significant association ($p$ value < 0.05) between genetic similarity and shared cultural practices only under the "Ethiopia-internal" analysis that is sensitive to intermixing among Ethiopian groups (Supplementary Table 8). Six traits out of the 20 reported by more than one ethnic group exhibited nominally higher ($p$ value < 0.05) genetic similarity among ethnic groups participating in the practice relative to those who did not participate or whose participation in the practice was unknown (Fig. 5). These practices include male and female circumcision and four different marriage practices (see Supplementary Note 6 for details). The average genetic similarity among groups sharing one of these six cultural traits in common was higher than that expected based on linguistic affiliation and spatial distance (Fig. 5), and we see increased evidence of recent intermixing among groups reporting male/female circumcision and sororate/cousin marriages relative to other SNNPR groups (Fig. 5, Supplementary Table 10, see "Methods", Supplementary Note 5). As an example, GLOBETROTTER infers admixture occurring 16 generations ago (95% CI: 11–21) in a cluster of the AA Cushitic-speaking Dasanech (cluster 11 in Fig. 4b), from a source most genetically related to a cluster containing the NS-speaking Murle and Nyangatom that share practices of arranged and abduction marriages (Fig. 4b, Supplementary Data 8).

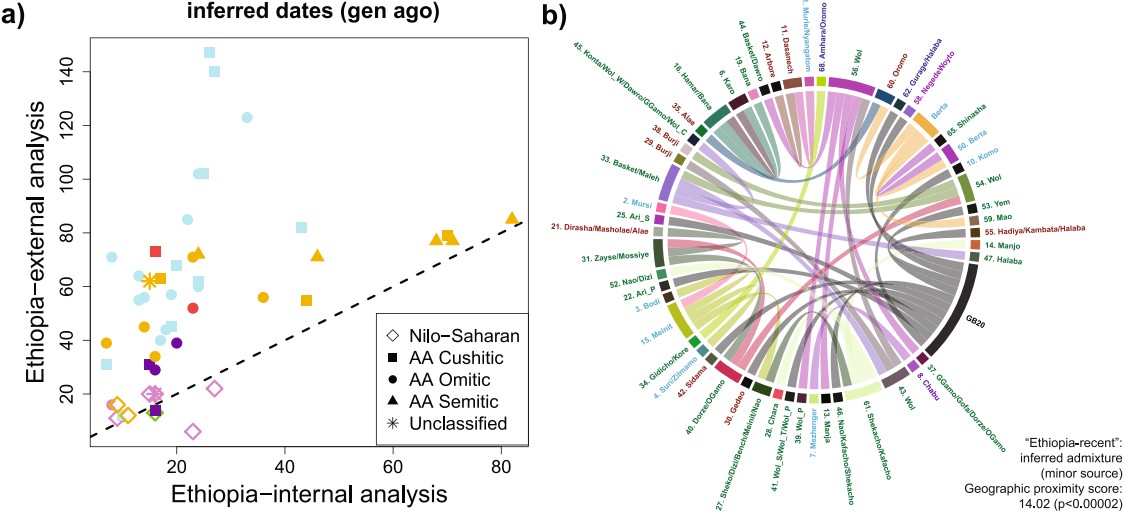

**Fig. 4 Evidence of recent intermixing among nearby Ethiopian groups. a** GLOBETROTTER-inferred dates (generations from present) for each Ethiopian cluster inferred to have a single date of admixture under each of the "Ethiopia-external" and "Ethiopia-internal" analyses. Inferred dates typically are more recent under the latter, indicating this analysis is picking up relatively more recent intermixing among sources represented by present-day Ethiopian clusters. Colours match those in Fig. 3 for these clusters. **b** GLOBETROTTER-inferred ancestry sources under the "Ethiopia-internal" analysis. Each Ethiopian cluster *X*, also including Mota, has a corresponding colour (outer circle). Lines of this colour emerging from *X* indicate that *X* was inferred as the best surrogate for the admixing source contributing the minority of ancestry to each other cluster it connects with. The thickness of lines is proportional to the contributing proportion. Ethiopian clusters, with labels coloured by language category according to Fig. 1a, are ordered by the first component of a principal components analysis applied to the geographic distance matrix between groups, i.e., so that geographically close groups are next to each other. The "geographic proximity score" gives the average ordinal distance between an admixture target and the surrogate that best represents the source contributing a minority of the admixture, with the one-sided *p* value testing the null hypothesis that admixture occurs randomly between groups (i.e., independent of the geographic distance between them) based on permuting cluster labels around the circle.

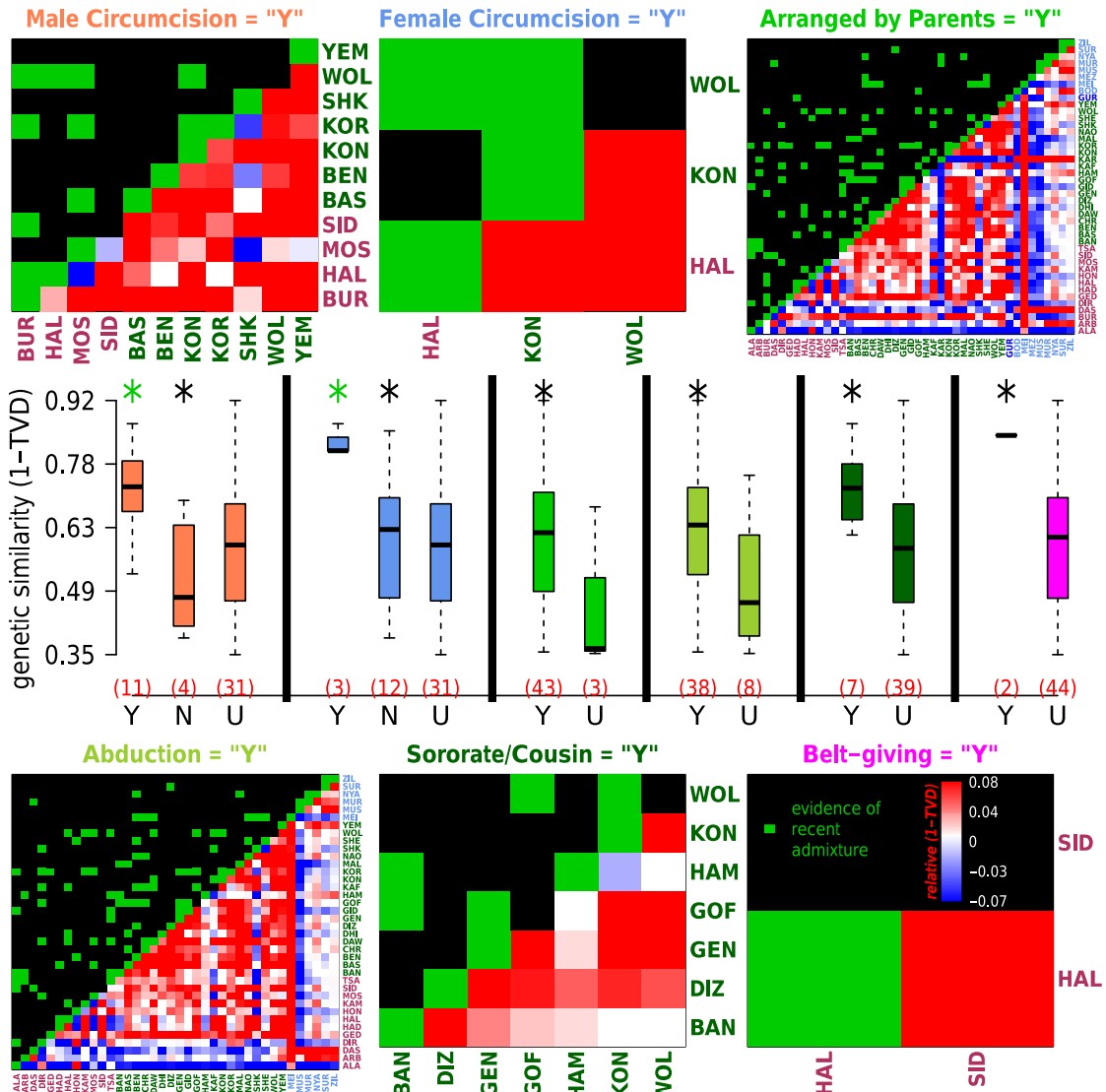

**Fig. 5 Sharing of self-declared cultural practices is associated with genetic similarity.** Boxplots depict the pairwise genetic similarity (under the "Ethiopia-internal" analysis) among ethnic groups that reported practicing ("Y"), not practicing ("N") or gave no information about practicing ("U") each of six different cultural traits (labelled above heatmaps, with text colours matching the corresponding boxplots' colours). Numbers of groups in each category are in parentheses in red. Each boxplot depicts the median (horizontal black bar), interquartile range (box), minimum and maximum (endpoints) values across pairwise comparisons. Stars above the boxplots denote whether there is a significant increase (one-sided empirical p val < 0.05, based on re-sampling groups, and without adjusting for multiple comparisons) in genetic similarity among groups in (black) "Y" versus "U" or (green) "Y" versus "N". The bottom right of each heatmap shows the increase (red) or decrease (blue) in average genetic similarity relative to that expected based on the ethnicities' language classifications (key in bottom right heatmap), after accounting for the effects of spatial distance, between every pairing of ethnicities who reported practicing ("Y") the given trait (axis labels coloured by language group as in Fig. 1a; group labels given in Supplementary Data 1). Green squares in the top left portion of the heatmaps indicate whether > =1 pairings of individuals from different ethnic groups share atypically long DNA segments relative to all other comparisons of people from the two groups, which is indicative of recent intermixing between the two groups (see Supplementary Note 5); p values provided in Supplementary Table 10.

## Discussion

Here we analyse a large-scale Ethiopian cohort densely sampled across ethnicities and geography, and annotated for cultural practices (Supplementary Note 2). This resource enabled us to disentangle several factors shaping genetic structure in Ethiopians. Wherever possible we only included individuals whose ethnicity matched that reported for parents and grandparents, which—if accurate—should exclude instances of ethnic re-identification and between-group intermixing occurring within the last two generations. This inclusion criterion implies that the patterns we have inferred reflect genetic patterns in Ethiopia

approximately two generations prior to the present-day. This plausibly underrepresents genetic similarity and intermixing among ethnic groups that would be observable in a random sample, though our results support widespread recent intermixing among ethnic groups nonetheless (Fig. 4).

Our simulations demonstrate how two different types of analyses, which we term "Ethiopia-internal" and "Ethiopia-external", can disentangle relatively recent from ancient shared ancestry to better understand the origins of different ethnic groups (Supplementary Figs. 4, 5). In the real data, groups referred to as socially marginalised occupational minorities in the social

anthropology literature, such as the Manjo from Kefa Sheka[66], the Manja from Dawro[73], the Ari/Wolayta Blacksmiths/Potters/Tanners[39,74] the Chabu and the Negede-Woyto[70,75,76], each show relatively high genetic distance from other Ethiopians using $F_{ST}$ (Supplementary Data 11) and under the "Ethiopia-internal" analysis (Fig. 2a, Supplementary Fig. 8a, Supplementary Data 5). However, under the "Ethiopia-external" analysis, these genetic distances become relatively small (Fig. 2a, Supplementary Fig. 8b, Supplementary Data 6), suggesting that the high levels of genetic differentiation between marginalised and other Ethiopians groups (e.g. measured by $F_{ST}$) have arisen through their relatively recent isolation. Consistent with this isolation, these groups also exhibit signatures of recent endogamy as reflected by higher degrees of genetic homogeneity (Supplementary Fig. 3a, b), with each forming a distinct cluster in ADMIXTURE analysis (Supplementary Fig. 11, Lawson et al.[77]).

In the Ari, we infer very similar sources and dates of admixture in independent analyses of distinct clusters that correspond to occupational groups (clusters 22, 24 and 25 in Fig. 3, Supplementary Fig. 12) under the "Ethiopia-external" analysis, with overlapping 95% confidence intervals spanning 42–146 generations (Supplementary Data 7). A parsimonious explanation of these findings, consistent with our simulations (Supplementary Fig. 5), is that the ancestors of the Ari were a single population when these admixture events occurred. This in turn suggests the ancestors of different Ari occupational groups became isolated from one another only within the past ~146 generations (<4200 years, assuming 28 years per generation[78]). This corresponds to the time period during which iron working is thought to have first appeared in Ethiopia[79] and supports the marginalisation theory of their origins[80] consistent with previous genetic studies[31,37].

Analogous to this, in the Chabu, who are not linguistically classified by Ethnologue, we infer admixture events (dated to 300–900 years ago) and ancestry proportions that are similar to those inferred in the Mezhenger (Fig. 3, Supplementary Fig. 12, Supplementary Data 7). These inferences are consistent with a high degree of intermarrying among the Chabu and Mezhenger, as has been proposed[31,81], and/or that these two groups split within the last ~900 years and had subsequently distinct linguistic trajectories. Nonetheless, among the Ethiopian groups, the Chabu are the strongest outliers under $F_{ST}$ and "Ethiopia-internal" analyses, consistent with previous claims of a decline in genetic diversity over the past 1000 years in the Chabu[31]. For the Negede-Woyto, the other group in this study for which there is no established linguistic classification in Ethnologue, we infer a relatively high amount of Egyptian-related ancestry (Fig. 3, Supplementary Fig. 12), which is consistent with the group's own origin narrative of a migration from Egypt by way of the Abay river[75]. The ancestry proportions and admixture dates inferred in the Negede-Woyto are similar to those for the Beta Israel and Agaw, whom some scholars have proposed possible genealogical relationships with[76], and show the highest average similarity to AA Semitic speakers ($p$ val > 0.05; Supplementary Fig. 9).

A caveat to the interpretation that groups with similar inferred admixture sources and proportions under the "Ethiopian-external" analysis share similar recent ancestry is that this analysis will have reduced (or no) power to discriminate between Ethiopian groups that indeed have separate ancestral sources if we have not included relevant non-Ethiopian groups to represent these sources. The large number of non-Ethiopian groups included in this sample, particularly those geographically proximal to Ethiopia, diminishes this possibility, but more samples from other sources, in particular from ancient individuals in Ethiopia, may increase our ability to identify older ancestral differences between Ethiopians using these techniques.

Both the "Ethiopia-internal" and "Ethiopia-external" analyses show a strong concordance between genetic differences and geographic distance among individuals (Fig. 1b, Supplementary Fig. 6), analogous to that shown previously among peoples sampled from European[43,82], African[30,83] and worldwide countries[84]. We also identify a correlation between genetic similarity and elevation difference, even after correcting for genetic similarity over geographic distances. Strikingly, we also see a correlation between spatial distance and the degree of genetic ancestry related to Mota, an ancient individual [4]whose remains were found in the Gamo Highlands of present-day Ethiopia 4500 years ago (Supplementary Fig. 13, Supplementary Table 9). This suggests a notable preservation of some population structure in parts of Ethiopia over the intervening period[4,31].

The "Ethiopia-external" SOURCEFIND and GLOBE-TROTTER results indicate that Ethiopians in the southwest, typically NS speakers plus a few non-NS speaking groups (Chabu, Dasanech, Karo), share more recent ancestry with non-Ethiopian Bantu and NS Nilotic speakers. In contrast, Ethiopian AA speakers in the northeast share more recent ancestry with Egyptians and West Eurasians (Fig. 3, Supplementary Fig. 12). The inferred timing and sources of admixture related to Egypt/W. Eurasian-like sources, starting around 100–125 generations ago (~2800–3500 years ago; Fig. 3, Supplementary Fig. 12), as in previous findings[27,85], is consistent with significant contact and gene flow between the peoples of present day Ethiopia and northern Africa even before the rise of the kingdom of D'mt and interactions with the Saba kingdom of southern Yemen which traded extensively along the Red Sea[79]. This timing is also consistent with trading ties between the greater Horn and Egypt. dating back only to 1500 BCE, when a well-preserved wall relief from Queen Hateshepsut's Deir el-Bahari temple shows ancient Egyptian seafarers heading back home from an expedition to what was known as the Land of Punt (Supplementary Note 1A). On the other hand, inferred admixture dates in groups with varying amounts of ancestry related to Bantu and NS Nilotic speakers are dated to <1100 years ago, with the exceptions of the NS-speaking Kwegu (~1500 years ago) and a second inferred older date (>1400 years ago) in the NS-speaking Meinit, which may reflect recent intermixing of NS-speakers with other Ethiopians. Such recent intermixing is consistent with mixed ancestry signals we see in some NS groups (e.g., see clusters containing Berta, Meinit and Nyangatom in clusters 15, 48–50 in Fig. 3, Supplementary Fig. 12).

To facilitate comparison, our SOURCEFIND analysis included reference groups related to the four proxies used for ancestry sources in ancient and present-day East African groups reported in Prendergast et al.[36] (see Supplementary Note 5). We excluded their aDNA samples as reference groups, because they reported them to have admixture from these four sources. While using different reference groups and techniques complicates direct comparisons, our inferred sources of ancestry broadly agree with that study. For example, the Agaw (clusters 66, 67) have relatively more Levant-like ancestry (which we match most closely to Egypt), the Ari (clusters 22, 24, 25; called Aari in Prendergast et al.[36]), have relatively more Mota-like ancestry, and the Ethiopian Mursi (cluster 2) have relatively more Dinka-like ancestry (Fig. 3, Supplementary Fig. 12). Simulations mimicking the admixture inferred here show high accuracy in inferred dates and sources, though illustrate a limitation whereby older dates of admixture (e.g., those reported in Prendergast[36]) may be masked by more recent admixture (Supplementary Fig. 5). Thus complex intermixing events, such as those exhibited here, can be difficult to dissect fully with these approaches and sample sizes, e.g., distinguishing between multiple pulses or continuous admixture.

A potential example are the NS-speaking Berta (clusters 48, 50), in which we infer only a single recent date of admixture but whom have complicated sources of ancestry that suggest multiple events (Fig. 3, Supplementary Fig. 12).

Interestingly, the association between cultural and genetic similarity is only apparent under the "Ethiopia-internal" analysis, which is more sensitive to recent shared ancestry (Supplementary Table 8). Another example consistent with this trend is that the NS-speaking Suri, Mursi, and Zilmamo, the only three Ethiopian ethnic groups that share the practice of wearing decorative lip plates, show atypically high genetic similarity under the "Ethiopia-internal" analysis but similarity levels comparable to other NS speakers under the "Ethiopia-external" analysis (Supplementary Fig. 14, Supplementary Data 13). This suggests a recent separation of these groups, i.e., more recently than they separated from all other sampled Ethiopian groups, and/or recent intermixing among them.

Overall the above examples illustrate how genetic data provide a rich additional source of information that can either corroborate or conflict with claims from other disciplines (linguistics, geography, archaeology, anthropology, sociology and history) while adding further details and/or novel insights and directions for future investigation. Our interactive map is designed to facilitate evaluation of genetic evidence for such claims, providing results from both the "Ethiopia-internal" and "Ethiopia-external" analysis to enable comparisons analogous to the examples above. Future work can compare these and other published genetic results (e.g.,[30,31,36]) to oral histories recorded for various ethnic groups. For example, some Mezhenger report that their ancestors originally migrated from Sudan to the present-day Gambella Regional State where Anuak lived, after which they migrated with the AA Omotic-speaking Sheko for a period before settling in their present-day homeland (The Council of Nationalities, Southern Nations and Peoples Region[37]). Consistent with this, in the "Ethiopia-external" analysis the Mezhenger have high inferred ancestry matching to the Sudanese Dinka (Fig. 3, Supplementary Fig. 12, Supplementary Data 7), and in the "Ethiopia-internal" analysis they have an inferred admixture event ~300–600 years ago among three sources that are best represented by clusters containing the Anuak, Sheko and other NS-speaking groups near the Mezhenger (Supplementary Data 8).

Our study also highlights the importance of considering topographical and cultural factors, in particular language, ethnicity and in some cases occupation, when designing sampling strategies for future Ethiopian genetic studies, e.g. genome-wide association studies (GWAS), which our interactive map can also assist with. Similar sampling strategies may be necessary to capture the genetic structure of peoples in some other African countries that also exhibit relatively high levels of genetic diversity and structure[65,83]. Finally, our analyses illustrate how cultural practices, e.g., participation in certain cultural and marriage customs, can operate as both a barrier and a facilitator of gene flow among groups, and consequently act as an important factor shaping human diversity and evolution.

## Methods

**Samples.** DNA samples from the 1082 Ethiopians whose autosomal genetic variation data are newly reported in this study (following quality control, see below) were collected in several field trips from 2000 to 2010, through a long-standing collaboration including researchers at University College London and Addis Ababa University. All study participants, including non-Ethiopians whose genetic variation data are newly reported in this study, gave their informed consent. Local permissions were obtained in all cases where applicable local ethical approval and regulations existed, e.g., Cameroon, Ministry of Higher Education and Scientific Research, Permits 0188/MINREST/B00/D00/D10/ D12 and 317/MINREST/B00/ D00/D10 and University of Yaounde I; Ethiopia, Ethiopian Science and Technology Commission and National Ethics Review Committee. Sample collection/usage for all unpublished data included in this study were approved by the UK ethics

committee London Bentham REC (formally the Joint UCL/UCLH Committees on the Ethics of Human Research: Committee A and Alpha, REC reference number 99/0196, Chief Investigator MGT). The analyses reported here were approved by UCL REC (Project ID: 5188/001).

Buccal swab samples were collected from anonymous donors over 18 years of age, unrelated at the paternal level. For all individuals we recorded their, their parents', paternal grandfather's and maternal grandmother's village of birth, language, cultural ethnicity and religion. In order to mitigate the effects of admixture from recent migrations that may be causing any genetic distinctions between ethnic groups to blur, analogous to Leslie et al.[43], where possible we genotyped those individuals whose grandparents' birthplaces and ethnicity were coincident[43]. However, for a few ethnic groups (Bana, Meinit, Negede Woyto, Qimant, Shinasha, Suri), we did not find any individuals fulfilling this birthplace condition; in such cases we randomly selected individuals whose grandparents had the same ethnicity. In these cases, the geographical location was calculated as the average of the grandparents' birthplaces (see Supplementary Note 2). We did not have geographic or birthplace information for Beta Israel individuals whose genetic variation data is newly released in this study. Information about elevation was obtained using the geographic coordinates of each individual in the dataset with the "Googleway" package. All the Ethiopian individuals included in the dataset are classified into 75 groups based on self-reported ethnicity (68 ethnic groups) plus occupations (Blacksmith, Cultivator, Potter, Tanner, Weaver) within the Ari and Wolayta ethnicities. Supplementary Data 1 shows the number of samples from each Ethiopian population and ethnic group that passed genotyping QC and were used in subsequent analyses. Figure 1a shows the geographic locations (i.e., birthplaces) of the Ethiopian individuals, though jittered to avoid overlap.

For comparison, we also incorporated 2678 non-Ethiopians (after quality control below) from 264 labelled present-day populations, and 40 high coverage aDNA genomes (including Mota), as described in this paragraph. Among these, non-Ethiopian samples newly released in this study include 23 Arabs from Israel, 13 Arabs from Palestine, 8 Bedouins from Saudi Arabia, 18 Berbers from Morocco, 7 Kotoko from Cameroon, 6 Muganda/Baganda from Uganda, 6 Mussese from Uganda, 13 Senegalese and 12 Syrians. All newly reported DNA samples in this study were genotyped using the Affymetrix Human Origins SNP array, which targets 627,421 SNPs (prior to our quality control), and merged with the Human Origin datasets published by Lazaridis et al.[41] and Lazaridis et al.[86], excluding their haploid samples (some ancient humans and primates)[41,86]. To these data we added present-day Indians and Iranians published by Broushaki et al.[52], and Lopez et al.[55], and genomes from present-day Africans published by Skoglund et al.[54], Gurdasani et al.[42] and Mallick et al.[28] (Supplementary Data 3)[28,42,52,54,55]. We also included 21 high coverage published ancient samples (>1X average coverage) from Africa[36,51,87], including GB20 'Mota' from Ethiopia[4], and 19 high coverage (>5X) published ancient non-African samples[45,46,48,52,53] (Supplementary Table 1).

BAM files for ancient samples were downloaded from the ENA website (https://www.ebi.ac.uk/ena), with each file checked for correct format and metadata using PicardTools. We estimated post-mortem damage using ATLAS[88] with "pmd", recalibrating each BAM file using ultra-conserved positions from UCNE (https://ccg.epfl.ch/UCNEbase/) and running ATLAS with "recal", and then generated maximum likelihood genotype calls and phred-scaled genotype likelihood (PL) scores for each position using ATLAS with "call". We used Conform-GT (https://faculty.washington.edu/browning/conform-gt.html) to ensure that strand was consistent with 1000 Genomes[89] across present-day and ancient datasets, merging the data and running Beagle 4.1[90] with "modelscale = 2" and the genetic maps at http://bochet.gcc.biostat.washington.edu/beagle/genetic_maps/plink.GRCh37.map.zip to re-estimate genotypes and impute missingness. We used vcf2gprobs, gprobsmetrics and filterlines (https://faculty.washington.edu/browning/beagle_utilities/utilities.html) to filter SNPs with an imputation accuracy of less than 0.98, and then we phased all samples using shapeit4[91] with "–pbwt-depth 16" and using their provided genetic maps.

To identify putatively related individuals, we used PLINK v1.9[61] with "–genome" to infer pairwise PI_HAT values, after first pruning for linkage disequilibrium using "–indep-pairwise 50 10 0.1". Instead of using the same fixed PI_HAT threshold value for all populations, we identified individuals with outlying PI_HAT values relative to other members of the same group label, in order to avoid removing too many individuals from populations with relatively low genetic diversity. Specifically, we found all pairings of individuals from populations (i,k) that had $PI\_HAT > 0.15$ and $PI\_HAT > \min(X_i + 3*\max\{0.02, S_i\}, Y_i + 3*\max\{0.02, D_i\}, X_k + 3*\max\{0.02, S_k\}, Y_k + 3*\max\{0.02, D_k\})$, where $\{X_i, Y_i, S_i, D_i\}$ are the {mean, median, standard deviation, median-absolute-deviation}, respectively, of pairwise PI_HAT values among individuals from population i. For populations with $<=2$ sampled individuals, the standard deviation and median-absolute-deviations are undefined or 0; therefore in such cases we added to the list any pairings with $PI\_HAT > 0.15$ that contained $>=1$ person from that population. Using a stepwise greedy approach, we then selected individuals from this list that were in the most pairs to be excluded from further analysis, continuing until at least one individual had been removed from every pair. This resulted in a total of 234 individuals removed, including 62 Ethiopians. All remaining Ethiopian pairs after this procedure had $PI\_HAT < 0.2$.

Following the quality control described above, the total number of samples in the merge was 3892, analyzed at 534,915 autosomal SNPs. We performed a principal-components-analysis (PCA) on the SNP data using smartpca[57,60] from

EIGENSOFT version 7.2.0, with standard parameters and the lsqproject option. For the PCA of all individuals (Supplementary Fig. 2a), we performed PCA on all individuals and used five outlier removal iterations (default). For the PCA of only African individuals (Supplementary Fig. 2b), we performed PCA on 2,110 present-day Africans and 8 Saudi-Bedouins without performing any outlier removal iterations to prevent excluding more isolated populations, subsequently removing the Saudi Bedouins from the plot and projecting the 21 ancient African samples including Mota.

**Genetic diversity and homogeneity.** We used three different approaches to assess within-group genetic homogeneity in the Ethiopian ethnic groups. First, we computed the observed autosomal homozygous genotype counts for each sample using the –het command in PLINK v1.9[61], taking the median value within each group. Second, we pruned SNP data based on linkage disequilibrium (–indep-pairwise 50 5 0.5), which left us with 359,281 SNPs, and used PLINK v1.9 to detect runs of homozygosity (ROH). This ROH procedure find runs of consecutive homozygous SNPs within groups that are identical-by-descent; here we report the total length of these runs per individual (Supplementary Fig. 3b). Third, we used FastIBD[62], implemented in the software BEAGLE v3.3.2, to find tracts of identity by descent (IBD) between pairs of individuals. For each population and chromosome, fastIBD was run for ten independent runs using an IBD threshold of $10^{-10}$, as recommended by Browning and Browning[62], for every pairwise comparison of individuals[62]. For each population, we report the fraction of the genome that each pair of individuals shares IBD (Supplementary Fig. 3a).

We assessed whether the degree of genetic diversity in Ethiopian ethnic groups was associated with census population size, by comparing different measures of genetic diversity described above (homozygosity, IBD and ROH) with the census population size using standard linear regression (Supplementary Fig. 3c). As population census are not always available and can be inaccurate, we limited this analysis to ethnic groups in the SNNPR, for whom census information was recently reported (The Council of Nationalities, Southern Nations and Peoples Region, 2017).

**Using chromosome painting to evaluate whether genetic differences among ethnic groups are attributable to recent or ancient isolation.** To quantify relatedness among individuals, we employed a "chromosome painting" technique, implemented in CHROMOPAINTER[63], that identifies strings of matching SNP patterns (i.e., shared haplotypes) between a phased target haploid and a set of phased reference haploids. By modelling correlations among neighboring SNPs (i.e., "haplotype information"), CHROMOPAINTER has been shown to increase power to identify genetic relatedness over other commonly-used techniques such as ADMIXTURE and PCA[43,58,63]. In brief, at each position of a target individual's genome, CHROMOPAINTER infers the probability that a particular reference haploid is the one which the target shares a most recent common ancestor (MRCA) relative to all other reference haploids. These probabilities are then tabulated across all positions to infer the total proportion of DNA for which each target haploid shares an MRCA with each reference haploid. We can then sum these total proportions across the reference haploids assigned to each of $K$ pre-defined groups.

Following van Dorp et al.[37], we used two separate CHROMOPAINTER analyses that differed in the $K$ pre-defined groups used:

1. "Ethiopian-external", which matches (i.e., paints) DNA patterns of each sampled individual to that of non-Ethiopians from $K = 264$ groups only (Supplementary Data 3).

2. "Ethiopia-internal", which matches DNA patterns of each sampled individual to that of all sampled groups, comprising 264 non-Ethiopian groups plus the 78 Ethiopian clusters defined in Supplementary Fig. 10 and the 4 Ethiopian groups from Mallick et al.[28], leading to $K = 346$ groups total[28].

Relative to our genetic similarity score (1-TVD, described in the next section) under the "Ethiopia-internal" analysis, our score under the "Ethiopia-external" analysis mitigates the effects of any recent genetic isolation (e.g., endogamy) that may differentiate a pair of Ethiopians. This is because individuals from groups subjected to such isolation typically will match relatively long segments of DNA to only a subset of Ethiopians (i.e., ones from their same group) under analysis (1). However, this isolation will not affect how the same individuals match to each non-Ethiopian under analysis (2), for which they typically share more temporally distant ancestors. Consistent with this, in our sample the average size of DNA segments that an Ethiopian individual matches to another Ethiopian is 0.68 cM in the "Ethiopia-internal" analysis, while the average size that an Ethiopian matches to a non-Ethiopian is only 0.23 cM in the "Ethiopia-external" analysis, despite the latter analysis matching to substantially fewer individuals overall and hence having a higher a priori expected average matching length per individual.

Following López et al.[55], van Dorp et al.[59], and Broushaki et al.[52], for each analysis (1) and (2) we estimated the CHROMOPAINTER algorithm's mutation/emission (Mut, "-M") and switch rate (Ne, "-n") parameters using ten steps of the Expectation-Maximisation (E-M) algorithm in CHROMOPAINTER applied to chromosomes 1, 8, 15 and 22 separately, analysing only every ten of 4081 individuals as targets for computational efficiency[52,55,59]. This gave values of {321.844, 0.0008304} and {178.8922, 0.0006667} for {Ne, Mut} in CHROMOPAINTER analyses (1) and (2), respectively, after which these values were fixed in a subsequent CHROMOPAINTER run applied to all chromosomes

and target individuals. The final output of CHROMOPAINTER includes two matrices giving the inferred genome-wide total expected counts (the CHROMOPAINTER ".chunkcounts.out" output file) and expected lengths (the ".chunklengths.out" output file) of haplotype segments for which each target individual shares an MRCA with every other individual.

**Inferring genetic similarity among Ethiopians under two different CHROMOPAINTER analyses.** Separately for each of the "Ethiopia-internal" and "Ethiopia-external" CHROMOPAINTER analyses, for every pairing of Ethiopians $i,j$ we used total variation distance (TVD)[43] to measure the genetic differentiation (on a 0-1 scale) between their $K$-element vectors of CHROMOPAINTER-inferred proportions (with $K$ defined above for both analyses), i.e:

$$TVD_{ij} = 0.5 \sum_{k=1}^{K} |f_k^i - f_k^j| \qquad (1)$$

where $f_k^i$ is the total proportion of genome-wide DNA that individual $i$ is inferred to match to individuals from group $k$ (see schematic in Supplementary Fig. 4). Throughout we report $1 - TVD_{ij}$, which is a measure of genetic similarity. When calculating the genetic similarity between two groups, we average $(1 - TVD_{ij})$ across all pairings of individuals $(i,j)$ where the two individuals are from different groups (e.g., for Figs. 2, 5, Supplementary Figs. 8, 9, Supplementary Data 5, 6). We note an alternative approach to measure between-group genetic similarity is to first average each $f_k^i$ across individuals from the same group, and then use (1) to calculate TVD between the groups by replacing each $f_k^i$ with its respective average value. Potentially this could give more power by reducing noise in the inferred copy vector for each group through averaging. However, here we instead use our approach of averaging $(1 - TVD_{ij})$ across individuals because of the considerable reduction in computation time when performing large numbers of permutations when assessing significance.

To test whether individuals from group $A$ are more genetically similar on average to each other than an individual from group $A$ is to an individual from group $B$, we repeated the following procedure 100 K times. Let $n_A$ and $n_B$ be the number of sampled individuals from $A$ and $B$, respectively, with $n_X = min(n_A, n_B)$. First we randomly sampled $floor(n_X/2)$ individuals without replacement from each of $A$ and $B$ and put them into a new group $C$. If $n_X/2$ is a non-integer, we added an additional unsampled individual to $C$ that was randomly chosen from $A$ with probability 0.5 or otherwise randomly chosen from $B$, so that $C$ had $n_X$ total individuals. We then tested whether the average genetic similarity, $\sum_{i,j} \frac{1-TVD_{ij}}{(n_X choose2)}$, among all $(n_X choose2)$ pairings of individuals $(i,j)$ from $C$ is greater than or equal to that among all $(n_X choose2)$ pairings of $n_X$ randomly selected (without replacement) individuals from group $Y$, where $Y \in \{A,B\}$ (tested separately). We report the proportion of 100 K such permutations where this is true as our one-sided $p$ value testing the null hypothesis that an individual from group $Y$ has the same average genetic similarity with someone from their own group versus someone from the other group (Supplementary Fig. 8, Supplementary Data 5, 6). Overall this permutation procedure tests whether the ancestry profiles of individuals from $A$ and $B$ are exchangeable, while accounting for sample size and avoiding how some permutations may by chance put an unusually large proportion of individuals from the same group into the same permuted group.

For each Ethiopian group $A$, in Supplementary Fig. 8 and Supplementary Data 5, 6 we also report the other sampled group $A_{max}$ with highest average pairwise genetic similarity to $A$. To test whether $A_{max}$ is significantly more similar to group $A$ than sampled group $B$ is, we permuted the group labels of individuals in $A_{max}$ and $B$ to make new groups $A_{max}^p$ and $B^p$ that preserve the respective sample sizes. We then found the average genetic similarity between all pairings of individuals where one in the pair is from $B^p$ and the other from $A$, and subtracted this from the average genetic similarity among all pairings of individuals where one is from $A_{max}^p$ and the other from $A$. Finally, we found the proportion of 100 K such permutations where this difference is greater than that observed in the real data (i.e., when replacing $B^p$ with $B$ and $A_{max}^p$ with $A_{max}$), reporting this proportion as a $p$ value testing the null hypothesis that individuals from group $A_{max}$ and group $B$ have the same average genetic similarity to individuals from group $A$. For each $A$, any group $B$ where we cannot reject the null hypothesis at the 0.001 type I error level (not adjusting for multiple testing) is enclosed with a white rectangle in Supplementary Fig. 8 and reported in Supplementary Data 5, 6.

As individuals are not allowed to match to themselves under the CHROMOPAINTER model, one potential issue with our paintings of Ethiopians under the "Ethiopian-internal" analysis is that each Ethiopian is allowed to match to one less individual in the cluster to which it is assigned relative to Ethiopians outside that cluster. For example, if cluster $A$ contains ten Ethiopians, each of those Ethiopians are allowed to match to nine people from cluster $A$ under the "Ethiopia-internal" analysis, while Ethiopians outside of cluster $A$ are matched to all ten. This may create a slight discrepancy in the $f_k^i$ values among Ethiopians for the 78 elements of $k$ representing the Ethiopian clusters, which in turn may affect differences in TVD among Ethiopian group labels. To test this, we repeated the above using an alternative "Ethiopia-internal" painting where each Ethiopian is matched to all other Ethiopians from their cluster and $n_k - 1$ Ethiopians from each other Ethiopian cluster $k$ after randomly removing one individual, while matching

to all individuals from every non-Ethiopian group as before. This gives a $K = 346$ length vector of $f_k^i$ values for each Ethiopian $i$ as before, but where each Ethiopian now has been painted against the same numbers of individuals from the $K$ groups. We found that results change very little, e.g., with the TVD values among all pairwise combinations of Ethiopian groups (Supplementary Fig. 8A, Supplementary Data 5) having correlation $r > 0.999$. This likely reflects how, for the given sample sizes in the $k$ clusters, removing one individual from a cluster $k$ results in people matching slightly more to the remaining $n_k - 1$ individuals in that cluster, so that the total matching to $k$ remains relatively unchanged. For comparison, in Supplementary Data 5 we provide columns at the far right end showing which groups were the closest match under this alternative "Ethiopia-internal" analysis; we note there are few changes relative to the original "Ethiopia-internal" analysis.

**Testing for associations between genetic similarity and spatial distance, shared group label, language and religious affiliation**. To test for a significant association between genetic similarity and spatial distance, we used statistical tests that are analogous to the commonly-used Mantel test[92] but that account for the non-linear relationships between some variables and/or adjust for correlations among more than three variables. We calculated genetic similarity ($G_{ij}$) between individuals $i$ and $j$ as $G_{ij} = 1 - TVD_{ij}$, geographic distance ($d_{ij}$) using the haversine formula applied to the individuals' location information, and elevation distance ($h_{ij}$) as the absolute difference in elevation between the individuals' locations. We assessed the significance of associations between $G_{ij}$ and $d_{ij}$ and between $G_{ij}$ and $h_{ij}$ using 1000 permutations of individuals' locations.

When using distance bins of 25 km, we noted that the mean genetic similarity across pairs of individuals showed an exponential decay versus geographic distance in the "Ethiopia-internal" analysis (Fig. 1b). Therefore, we assumed

$$G_{ij} = \alpha + \beta exp(-\lambda d_{ij}) + e_{ij}. \quad (2)$$

To infer maximum likelihood estimates (MLEs) for ($\alpha,\beta,\lambda$), we first used the "Nelder-Mead" algorithm in optim() in R to infer the value of $\lambda$ that minimizes the sum of $e_{ij}^2$ across all pairings of individuals $i,j$ when $\alpha = 0$ and $\beta = 1$, and then found the MLE for $\alpha$ and $\beta$ under simple linear regression using this fixed value of $\lambda$. As the main observed signal of association between genetic and spatial distance is the increased $G_{ij}$ at small values of $d_{ij}$, (e.g. $d_{ij} = 0$, which is not always accurately fit via the Nelder-Mead algorithm), our reported $p$ values are the proportion of permutations for which the mean $G_{ij}$ among all $(i,j)$ with permuted $d_{ij} < 25$ km is greater than or equal to that of the (unpermuted) real data (Supplementary Table 5a).

In contrast, we noted a linear relationship between mean $G_{ij}$ and $d_{ij}$ in the "Ethiopia-external" analysis (Supplementary Fig. 6b) and between mean $G_{ij}$ and $h_{ij}$ when using 100 km elevation bins under both analyses (Supplementary Fig. 6a, c). Therefore, for these analysis we assumed:

$$G_{ij} = \gamma + \delta x_{ij} + \varepsilon_{ij}, \quad (3)$$

where $x_{ij} = d_{ij}$ or $h_{ij}$. Separately for each analysis, we found the MLEs for ($\gamma,\delta$) using lm() in R. When testing for an association with elevation, we only included individual pairs $(i,j)$ whose elevation distance was less than 2500 km, which occurred in 730,880 (99.6%) of 733,866 total comparisons, to avoid undue influence from outliers. As we expect (and observe) the change in genetic similarity $\delta$ to be negative as spatial distance increases, our reported $p$ values provide the proportion of permutations for which the MLE of $\delta$ in the 1000 permutations is less than or equal to that of the real data (Supplementary Table 5b–d).

As $d_{ij}$ and $h_{ij}$ are correlated (r = 0.22, Supplementary Fig. 7c, d), we also assessed whether each was still significantly associated with $G_{ij}$ after accounting for the other under the "Ethiopia-internal" analysis. To test whether geographic distance was still associated with genetic similarity after accounting for elevation difference, we assumed:

$$d_{ij} = \eta + \theta h_{ij} + \kappa_{ij} \quad (4)$$

and used lm() in R to infer maximum likelihood estimates for ($\eta,\theta$). Then to test for an association between genetic similarity and geographic distance after accounting for elevation, we used Eq. (2) but replacing $G_{ij}$ with the fitted residuals $\varepsilon_{ij} = G_{ij} - \gamma - \delta h_{ij}$ from Eq. (2) and replacing $d_{ij}$ with the fitted residuals $\kappa_{ij} = d_{ij} - \eta - \theta h_{ij}$ from Eq. (4). We then repeated the procedure described above to calculate permutation-based $p$ values, first shifting $\kappa_{ij}$ to have a minimum of 0 (Supplementary Table 5a, c). Similarly, to test for an association between genetic similarity and elevation difference after accounting for geographic distance, we replaced $x_{ij}$ in Eq. (3) with the fitted residuals from an analogous model to (4) that instead regresses elevation on geographic distance, and replaced $G_{ij}$ in Eq. (3) with the fitted residuals $e_{ij} = G_{ij} - \alpha - \beta exp(-\lambda d_{ij})$ from Eq. (2). We used the same permutation procedure described above to generate $p$ values (Supplementary Table 5b, d).

We tested whether sharing the same (A) self-reported group label, (B) language category of reported ethnicity, (C) self-reported first language, (D) self-reported second language, or (E) self-reported religious affiliation were significantly associated with increased genetic similarity after accounting for geographic distance or elevation difference. We used 75 group labels for (A) (Supplementary Data 1), 66 first languages for (C), and 40 s languages for (D). For (B), we used the four labels in the second tier of linguistic classifications at www.ethnologue.com for

which we have data (i.e., Afroasiatic Omotic, Afroasiatic Semitic, Afroasiatic Cushitic, Nilo-Saharan Core-Satellite), excluding the Negede-Woyto and Chabu as they have not been classified into any language family. For (E), we compared genetic similarity across three religious affiliations (Christian, Jewish, Muslim), excluding religious affiliations recorded as "Traditional" as practices within these affiliations may vary substantially across groups.

To test whether each of these factors are associated with genetic similarity, we repeated the above analyses that use Eqs. (2)–(4) while restricting to individuals (including permuted individuals) that share the same variable Y, separately for Y= {A,B,C,D,E}. Our reported $p$ values give the proportion of permutations for which genetic similarity among permuted individuals sharing the same Y is more extreme than or equal to that of the real (un-permuted) data. For the "Ethiopia-internal" analysis when testing genetic similarity against geographic distance, this is the same $p$ value procedure as above, i.e., the proportion of permutations for which the mean $G_{ij}$ among all $(i,j)$ with permuted $d_{ij} < 25$ km is greater than or equal to that of the (unpermuted) real data (Supplementary Table 5a). When testing genetic similarity against geographic distance under the "Ethiopia-external" analysis, or testing genetic similarity against elevation difference under either analysis, this was instead defined as having any fitted value of $G_{ij}$, at 48 equally-spaced bins of $d_{ij} \in \{12.5, 1187.5$ km$\}$ or 25 equally-spaced bins of $h_{ij} \in \{50, 2450$ m$\}$, greater than or equal to that of the observed data.

As group label, language and religion can also be correlated with spatial distance and with each other (e.g. see Supplementary Fig. 7a, b), we performed additional permutation tests where we fixed each of (A)-(E) when carrying out the permutations described above. For example, when fixing (A), we only permuted birthplaces and each of (B)-(E) across individuals within each group label, hence preserving the effect of group label on $G_{ij}$. Applying this permutation procedure for each of (A)-(E), we repeated all tests described above, reporting $p$ values in Supplementary Table 5.

For each of geographic distance, elevation difference, and (A)-(E), our final $p$ values reported in the main text and Fig. 1b and Supplementary Fig. 6 that test for an association with genetic similarity are the maximum six permutation tests that permute all individuals freely or fix each of (A)-(E) while permuting (i.e., the maximum values across rows of Supplementary Table 5), with the following two exceptions. First, relative to the distances between birthplaces among all individuals, Ethiopians who share the same group label or who share the same first language live near each other (Supplementary Table 6), so that permuting birthplaces while fixing group label or first language do not permute across large spatial distances. Therefore, we ignore those permutations when reporting our final $p$ values for geographic distance and elevation difference (i.e., in the main text and Fig. 1b, Supplementary Fig. 6). Second, the high correlation between group label and first language (Supplementary Fig. 7a, b) makes accounting for one challenging (in terms of loss of power) when testing the other. Furthermore, few permutations are possible when testing language group while accounting for group label (0 permutations available) or first language. Therefore, we excluded permutations fixing group and fixing first language when testing each of group, first language and language group when reporting our final $p$ values in the main text and Supplementary Fig. 6. Note we do observe a significant association with genetic similarity and ethnicity after accounting for spatial distance (geographic or elevation) and major language group, suggesting ethnicity explains genetic similarity beyond that of classifications according to the second language tier of at Ethnologue. We caution that these analyses assume that the relationships among genetic, geographic and elevation distance can be modelled with simple linear or exponential functions, which is sometimes debatable (Supplementary Fig. 7c, d), indicating larger sample sizes may reveal deviations from these assumptions.

**Classifying Ethiopians into genetically homogeneous clusters**. We used fineSTRUCTURE[63] to classify 1268 Ethiopians (which includes all sampled Ethiopians except the eight Ethiopians from Mallick et al.[28] that were added later) into clusters of relative genetic homogeneity[28]. To do so, we first used SHAPEIT[93] to jointly phase individuals using default parameters and the linkage disequilibrium-based genetic map build 37 (available at https://github.com/johnbowes/CRAFT-GP/find/master). We then employed CHROMOPAINTER to paint each individual against all others, i.e., in a manner analogous to the "Ethiopian-internal" analysis, though using a slightly different set of reference populations (e.g., samples from Mallick et al.[28] were not included due to una-vailability at the time) and hence slightly different {Ne, Mut} values of {192.966, 0.000801}. We used default parameters, with the fineSTRUCTURE normalisation parameter "c" estimated as 0.20245. To focus on the fine-scale clustering of Ethiopians, we fixed all non-Ethiopian samples in the dataset as seven super-individual populations (Africa, America, Central Asia Siberia, East Asia, Oceania, South Asia and West Eurasia) that were not merged with the rest of the tree. We performed 2,000,000 sample iterations of Markov-Chain-Monte-Carlo (MCMC), sampling an inferred clustering every 10,000 iterations. Following Lawson et al.[63], we next used fineSTRUCTURE to find the single MCMC sampled clustering with highest overall posterior probability[63]. Starting from this clustering, we then per-formed 100,000 additional hill-climbing steps to find a nearby state with even higher posterior probability. This gave a final inferred number of 180 clusters containing Ethiopians. Results were then merged into a tree using

fineSTRUCTURE's greedy algorithm. We used a visual inspection of this tree to merge clusters, starting at the bottom level of 180 clusters, that had small numbers of individuals of the same ethnicity, as shown in Supplementary Fig. 10. After merging, we ended up with a total of 78 Ethiopian clusters.

We followed Leslie et al.[43] to generate a measure of cluster certainty using the last 100 fineSTRUCTURE MCMC samples[43]. In particular for each of these 100 MCMC samples, we assigned a certainty score for each individual i being assigned to each final cluster j (out of 78) as the percentage of individuals assigned to the same cluster as individual i in that MCMC sample that are found in final cluster j. (For each individual i, note these percentages sum to 100% across the 78 final clusters.) For each combination of individual and final cluster, we averaged these certainty scores across all 100 MCMC samples. For each of our 78 final clusters, in Supplementary Data 4 we report the average certainty score of being assigned to that cluster across all individuals assigned to that cluster. This average certainty score had a mean of 44.7% across all clusters (range: 5.6–88.8%). For comparison, the average certainty score of being assigned to a cluster other than the final classification we used had a mean of 0.7% across all clusters (range: 0.1–1.2%). We note that clusters do not necessarily correspond to distinct groups that split from one another in the past, but instead provide a convenient means to increase power and clarity of ancestry inference by (i) merging people with similar genetic variation patterns, and (ii) separating individuals of the same self-identified label that have different genetic variation patterns.

**Clustering Ethiopians using ADMIXTURE.** We also used ADMIXTURE v.1.3.0[56] to cluster Ethiopians. To do so, we first pruned the dataset for SNPs in linkage disequilibrium using PLINK v.2[61], removing SNPS with an $r^2 > 0.1$ within a 50-SNP window, which left 139,032 SNPs. We then applied ADMIXTURE to the Ethiopians using these SNPs and a varying number of clusters K = 2 − 15 and default parameters.

**Describing the genetic make-up of Ethiopians as a mixture of recent ancestry sharing with other groups.** We applied SOURCEFIND[72] to each of the 78 clusters to infer the proportion of ancestry that each clusters' individuals share most recently with 275 ancestry surrogate populations, consisting of 264 present-day non-Ethiopian populations and aDNA samples from 11 populations including Mota (Supplementary Note 5). Briefly, SOURCEFIND identifies the reference groups for which each Ethiopian cluster shares most recent ancestry, and at what relative proportions, while accounting for potential biases in the CHROMO-PAINTER analysis e.g. attributable to sample size differences among the surrogate groups. To do so, first each surrogate group and Ethiopian cluster $k$ is described as a vector of length 264, where each element $i$ in the vector for group $k$ contains the total amount of genome-wide DNA that individuals from $k$ are, on average, inferred to match to all individuals in group $i$ under the "Ethiopia-external" CHROMOPAINTER analysis. These elements are proportional to the $f_k^i$ described in the section "Inferring genetic similarity among Ethiopians under two different CHROMOPAINTER analyses" above. SOURCEFIND then uses a Bayesian approach to fit the vector for each Ethiopian cluster as a mixture of those from the 275 surrogate populations, inferring the mixture coefficients via MCMC[72]. In particular SOURCEFIND puts a truncated Poisson prior on the number of non-Ethiopian groups contributing ancestry to that Ethiopian cluster. We fixed the mean of this truncated Poisson to 4 while allowing 8 total groups to contribute at each MCMC iteration, otherwise using default parameters. For each Ethiopian cluster, we discarded the first 50 K MCMC iterations as "burn-in", then sampled mixture coefficients every 5000 iterations, averaging these mixture coefficients values across 31 posterior samples. In Supplementary Data 7 and Fig. 3, Supplementary Fig. 12, we report the average mixture coefficients as our inferred proportions of ancestry by which each Ethiopian cluster relates to the 275 reference groups, though noting only 13 of these 275 contribute >5% to any cluster in these results.

**Identifying and dating admixture events in Ethiopia.** Under each of the "Ethiopia-internal" and "Ethiopia-external" analyses, we applied GLOBETROTTER[58] to each Ethiopian cluster to assess whether its ancestry could be described as a mixture of genetically differentiated sources who intermixed (i.e., admixed) over one or more narrow time periods (Supplementary Note 5). GLO-BETROTTER assumes a "pulse" model whereby admixture occurs instantaneously for each admixture event, followed by the random mating of individuals within the admixed population from the time of admixture until present-day. When testing for admixture in each Ethiopian cluster under the "Ethiopia-external" analysis, we used 130 groups (119 present-day groups and 11 ancient groups) as potential surrogates to describe the genetic make-up of the admixing sources, excluding non-African groups that contributed little in the SOURCEFIND analysis for computational efficiency. When testing for admixture under the "Ethiopia-internal" analysis, we added as surrogates 64 of the 78 inferred Ethiopian clusters, removing 14 clusters (marked by asterisks in the first column of Supplementary Data 4) that contained small numbers of individuals from several ethnic groups and hence would confuse interpretation of results.

GLOBETROTTER requires two paintings of individuals in the target population being tested for admixture: (1) one that is primarily used to identify the genetic make-up of the admixing source groups (used as "input.file.copyvectors" in GLOBETROTTER), and (2) one that is primarily used to date the admixture event (used as the "painting_samples_filelist_infile" in GLOBETROTTER). For both the "Ethiopia-external" and "Ethiopia-internal" analyses, we used the respective paintings described in "Using chromosome painting to evaluate whether genetic differences among ethnic groups are attributable to recent or ancient isolation" above to define the genetic make-up of each group for painting (1). For (2), following Hellenthal et al.[58], we painted each individual in the target cluster against all other individuals except those from the target cluster, using ten painting samples inferred by CHROMOPAINTER per haploid of each target individual[58]. For the "Ethiopia-external" analysis, by design the painting in (2) is the same as the one used in (1). For the "Ethiopia-internal" analysis, we had to repaint each individual in the target cluster for step (2); to do so we used the previously estimated CHROMOPAINTER {Ne, Mut} parameters of {180.5629, 0.000610556}.

In all cases, we ran GLOBETROTTER for five mixing iterations (with each iteration alternating between inferring mixture proportions versus inferring dates) and performed 100 bootstrap re-samples of individuals to generate confidence intervals around inferred dates. We report results for null.ind = 1, which attempts to disregard any signals of linkage disequilibrium decay in the target population that is not attributable to genuine admixture when making inference[58]. All GLOBETROTTER results, including the inferred sources, proportions and dates of admixture, are provided in Supplementary Data 7-8 and summarized in Fig. 3 and Supplementary Fig. 12; see Supplementary Note 5 for more details. To convert inferred dates in generations to years in the main text, we used years ~= 1975 − 28 x (generations + 1), which assumes a generation time of 28 years[78] and uses an average birthdate of 1975 for sampled individuals that matches our recorded information.

**Permutation test to assess significance of genetic similarity among individuals from different linguistic groups.** To test whether individuals from language classification A are more genetically similar to each other than an individual from classification A is to an individual from classification B, we followed an analogous procedure to that detailed above to test for genetic differences between group labels A and B. Again let $n_A$ and $n_B$ be the number of sampled individuals from A and B, respectively, with $n_X = min(n_A, n_B)$. For each of 100 K permutations, we first randomly sampled $floor(n_X/2)$ individuals without replacement from each of A and B and put them into a new group C. If $n_X/2$ is a fraction, we added an additional unsampled individual to C that was randomly chosen from A with probability 0.5 or otherwise randomly chosen from B, so that C had $n_X$ total individuals. We then tested whether the average genetic similarity, $\sum_{i,j} \frac{1 - TVD_{ij}}{(n_X choose 2)}$, among all ($n_X choose 2$) pairings of individuals (i,j) from C is greater than or equal to that among all ($n_X choose 2$) pairings of $n_X$ randomly selected (without replacement) individuals from group Y, where $Y \in \{A, B\}$ (tested separately).

Individuals from the same ethnic/occupation label (i.e., those listed in Supplementary Data 1) are often substantially genetically similar to one another (Supplementary Fig. 8, Supplementary Data 5, 6), which may in turn drive similarity among individuals within the same language classification. Therefore, whenever a language classification contained more than two different ethnic/occupation labels, we restricted our averages to only include pairings (i,j) that were from different ethnic/occupation labels (including in permuted group C individuals). We report the proportion of 100 K such permutations where this is true as our one-sided p value testing the null hypothesis that an individual from language classification Y has the same average genetic similarity with someone from their own language group versus someone from the other language group (Supplementary Fig. 9, Supplementary Data 9, 10). To test whether classifications A and B are genetically distinguishable, we take the minimum such p value between the tests of Y = A and Y = B (Supplementary Fig. 9), which accounts for how some linguistic classifications include more sampled individuals and/or more sampled ethnic groups that therefore may decrease their observed average genetic similarity.

**Genetic similarity versus cultural distance.** Between each pairing of 46 sampled SNNPR ethnic groups, we calculated a cultural similarity score as the number of practices, out of 31 reported in the SSNPR book (The Council of Nationalities, Southern Nations and Peoples Region, 2017) and described in Supplementary Note 6, that the pair reported either both practicing or both not practicing (see Supplementary Data 12 for all groups' recorded practices). Despite the SSNPR book also containing information about the Ari, we did not include them among these 46 because of the major genetic differences among occupational groups (Fig. 2a). For the Wolayta, we included individuals that did not report belonging to any of the occupational groups analysed here.

We also calculated a second cultural similarity score whereby practices shared by many groups contributed less to a pair's score than practices shared by few groups. To do so, if H ethnic groups in total reported participating in a practice, any pair of ethnicities that both reported participating in this practice added a contribution of 1.0/H to that pair's cultural similarity score, rather than a contribution of 1 as in the original cultural similarity score. Similarly, if Z ethnic groups in total reported not participating in a practice, any pair of ethnicities that both reported not participating in this practice added a contribution of 1.0/Z to that pair's cultural similarity score.

Genetic similarity, geographic distance and elevation difference between two ethnic groups $A$, $B$ were each calculated as the average such measure between all pairings of individuals where $i$ is from $A$ and $j$ from $B$. We then applied a mantel test using the *mantel* package in the *vegan* library in R with 100,000 permutations to assess the significance of association between genetic and cultural similarity across all pairings of ethnic groups (Supplementary Table 8). We also used separate partial mantel tests, using the mantel.partial function in R with 100,000 permutations, to test for an association between genetic and cultural similarity while accounting for one of (i) geographic distance, (ii) elevation difference, or (iii) shared language classification (Supplementary Table 8). To account for shared language classification, we used a binary indicator of whether $A$,$B$ were from the same language branch: AA Cushitic, AA Omotic, AA Semitic, NS Satellite-Core.

For each of the 31 cultural practices, all 46 ethnic groups were classified as either (i) reporting participation in the practice, (ii) reporting not participating in the practice or (iii) not reporting whether they participated in the practice. For cultural practices where at least two of (i)-(iii) contained > =2 groups, we tested the null hypothesis that the average genetic similarity among groups assigned to category X was equal to that of groups assigned to Y, versus the alternative that groups in X had a higher average genetic similarity to each other. To do so, we calculated the difference in mean genetic similarity among all pairs of groups assigned to X versus that among all pairs assigned to Y. We then randomly permuted ethnic groups across the two categories 10,000 times, calculating $p$ values as the proportion of times where the corresponding difference between permuted groups assigned to X versus Y was higher than that observed in the real data. For 16 of 31 cultural practices, we tested X=(i) versus Y=(iii). For one cultural practice, we tested X=(ii) versus Y=(iii). For three cultural practices, we tested {X=(i) versus Y=(ii)}, {X=(i) versus Y=(iii)}, and {X=(ii) versus Y=(iii)}.

Six practices gave a $p$ value < 0.05 for one of the above permutation tests (Fig. 5). These $p$ values remained after first adjusting for spatial distance as described in this paragraph. We calculated the average genetic similarity between all ethnic groups sharing these six practices after accounting for the effects of spatial distance and language classification. To account for spatial distance, we used Eqs. (2)–(4) above, first adjusting geographic distance out of each of genetic similarity and elevation difference, and then regressing the residuals from the genetic similarity versus geographic distance regression against the residuals from the elevation difference versus geographic distance regression. We take the residuals for individuals $i,j$ from this latter regression as the adjusted genetic similarity between individuals $i$ and $j$ (denoted $G^*_{ij}$). In each of the above regressions, we fit our models using all pairs of Ethiopians that were not from the same language classification at the branch level (i.e., AA Cushitic, AA Omotic, AA Semitic, NS Satellite-Core), in order to account for only spatial distance effects that are not confounded with any shared linguistic classification. We calculate the average spatial-distance-adjusted genetic similarity between each ethnic group $A$,$B$ as the average $G^*_{ij}$ between all pairings of individuals where $i$ is from $A$ and $j$ is from $B$. Then to adjust for language classification, we calculated the expected spatial-distance-adjusted genetic similarity for each pairing of language branches $C$,$D$ as the average adjusted genetic similarity across all pairings of ethnic groups $A$, $B$ where $A$ is from $C$ and $B$ is from $D$. For each pair of ethnic groups that share a reported cultural trait shown in Fig. 5, we show the adjusted genetic similarity between that pair minus the expected spatial-distance-adjusted genetic similarity based on their language classification. This therefore illustrates the genetic similarity between the two groups after adjusting for that expected by their spatial distance from each other and their respective language classifications (lower right triangles of heatmaps in Fig. 5).

For each of these six cultural practices shown in Fig. 5, we also assessed whether there was evidence of recent intermixing among people from pairs of groups that both reported the given practice (see Supplementary Note 5). To do so, we indicate in the upper left triangles of the heatmaps in Fig. 5 whether >=1 pairings of individuals, one from each group, have average MRCA segments >= 2.5 cM longer than the median length of average inferred MRCA segments across all such pairings of individuals from the separate groups. We calculated the average MRCA segment length between two individuals as the total inferred cM length of matching between the two divided by the total inferred number of segments matching between the two, as inferred by CHROMOPAINTER under the "Ethiopia-internal" analysis. We calculated the proportion out of 10,000 random samples of $n$ groups (sampled from the 46 SNNPR groups analysed here) where a greater or equal number of group pairings showed this trend, also considering various different values of excess average MRCA segment size (Supplementary Table 10).

**Reporting summary**. Further information on research design is available in the Nature Research Reporting Summary linked to this article.

## Data availability

Genotype data, birthplace information and self-reported group label, first language, second language and religious affiliation for newly genotyped individuals are available for non-commercial use at the European Genome-phenome Archive (EGA), which is hosted by the EBI and the CRG, under accession number EGAS00001005171. Previously published data were obtained from: www.ebi.ac.uk/ena/browser/view/PRJEB32086, www.ebi.ac.uk/ena/browser/view/PRJEB21878, https://doi.org/10.6084/m9.figshare.5223583.

v1, www.ebi.ac.uk/ena/browser/view/PRJEB11848, www.ebi.ac.uk/ena/browser/view/PRJEB11450, www.ebi.ac.uk/ena/browser/view/PRJEB22660, www.ebi.ac.uk/ena/browser/view/PRJEB2830, www.ebi.ac.uk/ena/data/view/PRJEB9021, www.ncbi.nlm.nih.gov/sra?term=PRJNA230689, genetics.med.harvard.edu/reichlab/Reich_Lab/Datasets.html, africangenome.org/Main_Page, www.ebi.ac.uk/ena/browser/view/PRJEB32086, www.ebi.ac.uk/ena/browser/view/PRJEB31373, www.ebi.ac.uk/ena/browser/view/PRJEB22660, www.ebi.ac.uk/ena/browser/view/PRJEB14180, www.ebi.ac.uk/ena/browser/view/PRJEB8448.

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

## Acknowledgements

This work is funded by BBSRC (Grant Number BB/L009382/1). GH is supported by a Sir Henry Dale Fellowship jointly funded by the Wellcome Trust and the Royal Society (Grant Number 098386/Z/12/Z) and supported by the National Institute for Health Research University College London Hospitals Biomedical Research Centre. We also acknowledge the UCL Biosciences Big Data equipment grant from BBSRC (BB/R01356X/1). GB is a member of the MalariaGEN resource centre, supported by Wellcome [204911/Z/16/Z]. We thank David Reich and the Children's Hospital of Philadelphia for genotyping the samples on the Human Origins array. We thank Karl Skorecki for assistance with sampling. Samples analysed in this study are drawn from a collection assembled and managed over many years. We thank the very many collectors, donors, students and technical staff who have contributed to this enterprise.

## Author contributions

A.T., T.O., E.M., E.B., and N. Bradman performed sample collection. N. Bradman oversaw and managed the sample collection programme. M.G.T. designed and managed post-collection sample processing procedures. S.L., L.v.D., N. Bird, S.M. and G.H. performed the analyses. S.L., A.T., N. Bradman, and G.H. wrote the paper with input from co-authors including R.B. G.B. designed the webpage resource.

## Competing interests

The authors declare no competing interests.
