## [Peer Review File · Nature Communications]

Reviewers' Comments:

Reviewer #1:

Remarks to the Author:

The genetic landscape of Ethiopia: diversity, intermixing 1 and the association with culture

The authors have assembled genome-wide SNP array dataset from 1268 Ethiopians representing 68 ethnic groups and characterized the impact of geographic and cultural affinities on genetic substructure across groups. Using previously published methods, like fineSTRUCTURE, SOURCEFIND and Globetrotter, the authors characterize the population history of Ethiopians. They also investigate signals of selection and adaptation to high altitude in Ethiopians. The analysis performed is compelling but technical concerns that I discuss below.

Major concerns:

1. Dates of admixture - In the analysis for dating admixture, the authors report the timing of admixture using Globetrotter. In previous work, this method has been tested for dating mixtures involving up to 3 ancestral populations. The Ethiopian samples that the authors analyze have ancestry for 4-6 ancestral groups. Thus, its not clear which mixture is being dated and reported here. More generally, its not clear if Globetrotter is applicable in this setting and gives unbiased results. Indeed, a recent paper explores this scenario and concludes that current method are inapplicable for dating admixtures in case of >2 admixture events (Chimusa et al. 2018, <https://www.ncbi.nlm.nih.gov/pubmed/30462157>). Thus it would be useful if authors can provide simulations mimicking the population history of Ethiopians (4-5way mixtures) and showing that the method provide reliable results for the timing of multi-way mixtures.

2. Analysis of "Ethiopian-internal" and "Ethiopia-external"

In order to study how internal and external populations have impacted the population history of Ethiopians, authors perform analysis of GLOBETROTTER and SOURCEFIND under two settings "Ethiopian-internal" and "Ethiopia-external" using ancestral groups either from inside or outside Ethiopia. While this is an interesting idea, the interpretation of the results is less clear. For instance, the internal analysis should largely pick up Identity by descent sharing across Ethiopians groups and hence founder events in Ethiopia vs. admixture between various Ethiopian groups. Here again it would be useful if authors can perform simulations and highlight how the history of founder events and endogamy within Ethiopia would impact the results. Comparing these Ethiopia internal results to IBD analysis will also be helpful to understand the source of the population structure.

3. Selection scan

The details of the selection scan applied to identify signals of adaptation seems unclear to me. Given that Ethiopian populations are so structured, how was this analysis performed? In particular, how was population structure accounted in selection scan - since long haplotypes and increased F_{st} differences would be expected due to admixture alone.

Minor concerns:

1. Figures are too crowded and often hard to follow - would be useful to split these into multiple figures so that the results are clearer.
2. Discussion is too long and summarizes large parts of the results. Would be useful to cut down a little so that message is clearer.
3. Figures - reference populations used for analysis, say GLOBETROTTER, etc. are often unclear and not specified in the legend.

Reviewer #2:

Remarks to the Author:

López et al analyzed a large SNP array dataset genotyped in >1000 Ethiopians from 68 ethnic groups. They used Chromopainter and fineSTRUCTURE to analyze genetic structure and Globetrotter to analyze sharing of haplotypes between Ethiopian groups and between Ethiopians and a reference dataset from global populations. This allowed them to make inferences about recent migration events between Ethiopian groups and older migration events with non-Ethiopian populations. They present an interactive website to view sharing of haplotypes between groups. The authors also analyze correlations between genetic, geographic, linguistic and cultural variation. This is a large and impressive dataset from ethnically diverse African populations, which are under-represented in human genetics research and can potentially be an important resource if individuals are properly consented and data is shared. The analysis of genetic and cultural diversity is also of considerable interest. However, there are many serious concerns about the study, which would need to be addressed before a manuscript could be published in this or any journal.

Major Concerns:

1. Informed consent and ethical considerations of this study

a. Page 28 Lines 11-17. What is the informed consent obtained from these study participants?

Documentation must be provided indicating whether all individuals included in the study signed consent forms. Are these consent conditions consistent with the public release of these data and analyses performed on these data?

b. Page 6 Lines 2-4: "including 8 unpublished Kotoko individuals from Cameroon, 20 Moroccan Berbers, 13 individuals from Senegal and 11 Chagga individuals from Tanzania (Fig 1b, 3Table S2)." There is no mention of research permits from Cameroon or the other countries. Again, demonstration of documented informed consent for every individual is critical.

c. Page 50 Lines 5-8: "Genotype data, birthplace information and self-reported group label, first language, second language and religious affiliation for each individual are available at the Gene Expression Omnibus database online (www.ncbi.nlm.nih.gov/geo), series accession number XXXX." The Gene Expression Omnibus accession number for access to the data is not listed here. These data must be made publicly available with a clear description of data use restrictions prior to publication. Otherwise, the importance of the study is greatly reduced.

d. Page 28 Lines 14-17: I was not able to find this REC record. Could you include a link to the appropriate website to identify the REC? I checked on this website: <https://www.hra.nhs.uk/about-us/committees-and-services/res-and-recs/search-research-ethics-committees/>. This needs to be easily accessible. A direct link to this record in addition to its description would be helpful.

Page 5 Lines 1-6: "Furthermore, some of the authors of this study (A. Tarekegn, N. Bradman) translated into English and edited a compendium (originally published in Amharic) that documented the oral traditions and cultural practices of 56 ethnic groups of the Southern Nations, Nationalities and Peoples' Region (SNNPR) of Ethiopia through interviews with members of different ethnic groups (The Council of Nationalities, Southern Nations and Peoples Region, 2017)." Given that the cultural classification are central to the analyses in this study, the methods should be presented and they should provide a copy of this document which, ideally, should be assessed by a cultural anthropologist/linguist for accuracy.

e. Page 11 Lines 1-4: "However, we found evidence ($p\text{-val} < 0.05$) of genetic isolation between Christians and either Muslims or people practicing traditional religions within seven of 16 groups for which we sampled at least five individuals from each religion (Table S6)." It is not clear how strong this "genetic isolation" is (this is where standard F_{st} analyses would be helpful) or how you are defining people practicing traditional religions. It is very common that indigenous Africans will practice traditional religion in addition to identifying as Christian or Muslim. Further, the religious classification is likely to be highly correlated with ethnicity and language classification. Indeed, it would be informative to know how they correlated with the "clusters" inferred from fineSTRUCTURE. If those are

controlled for, do you still see differences between groups based on language? Analyses of genetic and religious affiliation (in addition to many of the other cultural practices they describe) has the potential to be highly inflammatory in Ethiopia, a region that is currently experiencing conflict between ethnic groups. I would want to know that individuals gave consent for that analysis and that it was included in the ethical review at the country level. A copy of the informed consent form used would help clarify.

2. There are multiple instances in this paper where there are missing citations directly relevant to their study. For example, Scheinfeldt et al. 2012 presented a study of adaptation to high altitude in Ethiopia (page 3, lines 12 – 14). They should cite additional studies in Africa (including Ethiopia) that have examined correlation between genetic, geographic, and/or linguistic diversity (e.g. Tishkoff et al. 2009, Verdu et al. 2017; Baker et al. 2017; Scheinfeldt et al. 2019, etc) when they describe similar results (e.g. page 3 lines 19 – 21, page 13 lines 7 – 9, page 18 lines 3 – 6, page 19 lines 3 -4 and 5 – 8.).

3. The use of ethnic groups and clusters used in the study is not always clear. It seems to change depending on the analysis. This needs to be clarified and explained why in some cases they use ethnic groups for analyses and others they refer to clusters. Furthermore, they identify 78 clusters and have 68 ethnic groups. If they are separating ethnic groups into essentially their own clusters, then how is this analysis informative? Is your approach overfitting the data given that there are more clusters than ethnic groups? Looking at the large supplemental tree (Figure S3), how confident are you that each node in your tree is distinct?

4. Page 25 Line 24 to Page 26 Line 3 “When testing for admixture in each Ethiopian cluster under the “Ethiopia- internal” analysis, we used all other Ethiopian clusters (excluding 11 clusters – those with the suffix ‘adm’ in Extended Table 2 – that each consisted of small numbers of individuals from multiple ethnic groups)”. Some clusters have very small sample size (some are $N = 1$), what was your sample size cutoff when excluding clusters from an analysis. For analyses where you did not exclude clusters, do you expect the small sample size to impact your results?

5. This paper is based on analyses of haplotype sharing between populations using Globetrotter. How would other standard approaches compare to the analyses presented here (e.g. ADMIXTURE, Fst, and f4/D statistics)? They present ROH and IBD analyses in a Figure but only discuss in the methods. This should be discussed in the results. What do these results imply about inbreeding and relatedness among individuals in the dataset? The hunter-gatherer populations appear to have high IBD and ROH. How might that effect the admixture analyses?

6. How does demographic history impact your inference of migration timing with Globetrotter? They mention that Globetrotter assumes single pulses of migration for up to two events, which is then followed by random mating. What happens if this model is not correct? Indeed, prior studies have shown that many ethnic groups in Ethiopia may have 3-way admixture. Yet, it appears that they only considered 2-way admixture. Further, they assume a single initial pulse of admixture followed by constant gene flow. But that is unlikely to be the case for all of the populations included in this study. Additionally, they mention on page 10 of the supplement that 26 of 53 clusters’ admixture patterns are unclear. Isn’t this a problem for interpreting their results?

7. Page 7 lines 2 – 6: “Importantly, if two clusters infer admixture events with matching dates, sources and proportions, a parsimonious explanation is that the ethnic groups contained in these clusters were a single combined population when this admixture occurred, hence placing an upper bound on their split time (van Dorp et al., 2015, van Dorp et al., 2019). Conversely, if two clusters show different admixture inferences, they were likely not a combined population when the most recent admixture occurred.” The authors state that if two populations share a similar ancestry and time of admixture, they most likely share a common ancestry and were “a single combined population”. Indeed, this is the crux of the entire study. But how do we know that there isn’t another model that’s equally as likely? Again, without modeling the many possible demographic scenarios that could also explain this

pattern, it's not possible to know this and, therefore, the conclusions are over-stated. It should be noted that in the van Dorp 2019 paper, they do simulations based on testing models specific to populations in the DRC.

8. Supplement Page 10: "A likely explanation for these type-(a) versus type-(b) patterns is that at least two separate types of intermixing occurred in the ancestors of present-day Ethiopians. The first involved older intermixing between a Sub-Saharan African source versus a West-Eurasian source (type (a) clusters), after which these admixed ancestors intermixed again with individuals from West-Central Africa (type (b) clusters)." Again, are there other equally likely explanations? Without simulations and demographic modeling, this explanation is not convincing.

9. Supplement Page 11: "each of the 61 clusters that inferred admixture events classified into (1)-(3), i.e. excluding clusters that concluded "uncertain" admixture, we took the GLOBETROTTER inference of the most strongly signalled event (see Supp Table S5-S6) This event infers that two distinct sources intermixed, one contributing a majority of ancestry and the other contributing a minority of ancestry to the cluster's individuals." How was the "most strongly signaled event" determined? And again, what about the case when there may be more than two source populations contributing to admixture in a population?

10. The description of reference populations is confusing in this paper

a. It appears that for the GLOBETROTTER analysis there were 252 reference groups. Many of these are named by a country or broad geographic region and others by a particular ethnic group, making interpretation difficult. There are also many regions under-represented, particularly from Africa. With so many source populations, there are many potential combinations of admixture events. But it's not clear how these are influenced by sample size (seems that reference group sample sizes range from 2 – 100) and the sample sizes of "clusters" within Ethiopia vary. Will differences in sample size impact your analysis when inferring the most likely source population for an admixture event? Also, how might these results be influenced by drift in small/inbred populations such as the "Shabo", as seen in Figure S8?

a. Importantly, nearly all the "source" populations in Africa are, themselves, admixed (Tishkoff et al (2009), Pickrell et al (2014), Pagani et al (2015), Schuenemann et al (2017)). How might your approach be impacted when estimating admixture times and admixture coefficients due to admixture in your reference populations?

b. Why is the West Eurasian ancestry estimates presented in Figure 3B for the Amhara (~5%) so low compared to Pickrell et al (2014) (49.2%)?

c. Page 8 Lines 8-11: "SOURCEFIND ancestry inference under the "Ethiopia-external" analysis (Fig 3) showed matching primarily to Mota and six major geographic regions to which at least one Ethiopian cluster matched 9>5% of their DNA: Egypt, East Africa, North Africa, Somalia, West Africa and West Eurasia (Fig 101b)." Again, there is a concern about the reference populations used. West Africa and East Africa are so broad as to be uninformative given the high levels of genetic heterogeneity in those regions.

d. Page 21 Lines 2-7: "A caveat to the interpretation that groups with similar inference under the "Ethiopian-external" analysis share similar recent ancestry is that this 4 analysis will have decreased (or have no) power to discriminate between Ethiopian groups that indeed have separate ancestral sources if we have not included relevant non-Ethiopian groups to represent these separate sources. The large number of non-Ethiopian groups included in this sample, particularly those surrounding Ethiopia, diminishes this possibility". This is, indeed, a major potential problem and it is not correct that neighboring groups are heavily sampled. There are only four populations from Tanzania with sample sizes > 10 and there are no populations from Kenya included. The population from Somalia

has only 13 individuals, yet many of the conclusions in this study are based on analysis of that population which is concerning. It is not clear how this very sparse sampling on other regions of Africa may bias the study.

e. Page 21 Lines 13-14. "with results indicating intermixing between a source represented by (a) Sub Saharan African groups". Sub Saharan Africa is too vague to be informative.

f. Page 21 Lines 16-20: "Notably, Somalia differs among clusters in that it acts as a surrogate to source (a) in north/northeastern clusters with higher amounts of inferred ancestry related to Egyptian groups (type "a" clusters), while it acts as a surrogate to source (b) in west/southwest clusters with higher amounts of inferred ancestry related to East/West Sub-Saharan Africa (type "b" clusters) (Fig 3, Fig S9b-c, Table S10, Appendix B)" This is likely being impacted by Somalia being admixed with West Eurasian ancestry.

g. Supplement Page 9: "GLOBETROTTER concludes evidence of admixture in 69 of our 78 Ethiopian clusters, with inferred dates ranging from three to 118 generations ago. These inferred dates are robust across two alternative analyses that use different surrogates, one where we simply exclude Mota as a surrogate and another where we include only five modern groups as surrogates: Bantu_SA, Han, Japanese, Mende_Sierra_Leone_MSL, Yemen (Fig S9a)." Why choose these reference populations? They do not seem to fit well with the population history of Ethiopia.

11. The large heatmaps are challenging to interpret. I also recommend moving Figure 4, 1B, and 5A to the supplement. They also need to define what the colors mean in the figure S3 caption.

12. Page 9 Lines 4-7: "For example, under the former analysis Ari and Wolayta people who work as cultivators or weavers are more genetically similar to members of other ethnicities on average than they are to people from their own ethnicities who work as potters, blacksmiths and tanners (top left of squares in Fig 2c)". They make many statements like this throughout the manuscript. Their method of inferring genetic distances is rather "ad hoc" and it's not clear why they didn't use more traditional methods.

13. Page 43 Lines 2-4: "In general disentangling whether there is significant evidence of independent effects for each of language, group label and spatial distance while also mitigating effects of recent isolation, though suggested in Table S4c-d, will require larger sample sizes than those considered here." That is correct and it's not clear how they have disentangled these factors. Therefore, the conclusions are not well justified.

14. Page 28 Lines 18-19: "Buccal swab samples were collected from anonymous donors over 18 years of age, unrelated at the paternal level." How did you determine that individuals are not related at the paternal level given high incidence of non-paternity? Relationships should be inferred using the genetic data. Are their differences in the amount of 1st, 2nd, and 3rd degree relationships in different populations? If so, then why wasn't the dataset pruned to remove related individuals. How will including them influence the analyses?

15. The use of internal vs external analysis of reference populations reduces the impact of strong genetic drift. However, from the language used in this paper and prior papers, it is not clear by how much this approach removes this impact of drift: Page 6-7 Lines 23-3 "A key conceptual difference between the two analyses is that the "Ethiopia-external" analysis mitigates genetic differences between individuals attributable predominantly to recent isolation, e.g. endogamy, that can cause a groups' individuals to match large proportions of their haplotype patterns to each other (van Dorp et al., 2015; van Dorp et al., 2019)" As you state in the Van Doorp et al (2015) paper: "The differences we observe between ARIB and ARIC under CHROMOPAINTER analysis (A), ADMIXTURE and FST are no longer present under CHROMOPAINTER analyses (B) and (C) (Figs 1b-1c and 3, S7 and S11 Figs and

S10–S12 Tables). A key difference is that for each Ari group we disallow “self-copying” from individuals with the same label under analyses (B) and (C), which should reduce the magnitude of any differences seen in the other approaches that are attributable to bottleneck effects in the Blacksmiths.” Words such as “reduced” and “mitigates” need to be quantified. Have they performed simulations to determine the impact of strong drift on their analyses?

16. Page 18 Lines 16-22: An entirely new analysis of targets of natural selection cannot be introduced in the discussion section. Also, how much power do they have to identify selection signals? It was not sufficiently clear from the methods and sup material how these analyses were done and how to interpret them and, importantly, insufficient information about determining outliers. They appear to have pooled individuals into “high altitude” vs “low altitude” groups in an ad hoc manner without considering shared ancestry. Why not do something like a PBS analysis? These analyses were not well justified and should be removed from the paper.

17. Page 5 Lines 23-24: “We compared SNP patterns in each present-day Ethiopian to those in all other present-day Ethiopians and the 4,500 year-old Ethiopian sample “Mota” (Gallego-Llorente et al., 2015)” It would be useful to emphasize why the genetic ancestry of the Mota sample makes this individual an important reference group for your study, especially since this sample is often referred to elsewhere in the text. Also, what is the power/standard error for Chromopainter estimating ancestry contributions from only 1 individual? Could ancient DNA quality also impact this analysis?

18. Page 20 Lines 15-22: “Analogously, minority discriminated groups such as the Negede Woyto (Teclehaimanot, 1984; Legesse, 2013), Shabo (Dira & Hewlett, 2017), Manjo from Kefa Sheka (Freeman & Pankhurst, 2003), and Manja from Dawro (Dea, 2007) exhibit patterns of relatively low genetic similarity to other Ethiopians under the “Ethiopia-internal” analysis that disappear under the “Ethiopia-external” analysis (Fig 2b), in addition to showing higher degrees of genetic homogeneity (Fig S8a). Given this consistent pattern across discriminated groups, we argue it is most likely that discriminatory practices are directly responsible for having increased the genetic differentiation between discriminated peoples and other Ethiopians.” This conclusion is not well justified. These patterns may also be consistent with the small census sizes for some of these groups, such as the “Shabo”.

19. Page 13 Lines 5-7: “These observations suggest that the first three tiers of Ethiopian language classifications at www.ethnologue.com are genetically -- in addition to linguistically -- separable on average, and that these genetic differences may not solely be attributable to recent isolation”. I’m not following how they reached this conclusions. Do they observe statistically significant differences?

20. Are all cultural traditions independent of each other? If any are not, did you account for correlation between traditions in your analysis?

Minor Comments:

1. Page 17 Lines 5-8: “Therefore, to test for evidence of such recent intermingling between two 6 groups, we assess whether some pairings of individuals, one from each group, have average inferred 7 MRCA segments that are over 2.5 centimorgans (cM) longer than the median length of average inferred MRCA segments across all such pairings of individuals from the two groups (Fig 7).”, They should explain why they chose 2.5 cM.

2. Page 14 Lines 16-17: “The other group with uncertain linguistic classification in our study, the NegedeWoyto, are significantly differentiable ($p\text{-val} < 0.001$).” If all clusters, which basically correspond with ethnic groups, are statistically significant based on fineSTRUCTURE, how do we interpret this finding?

3. Page 22 Lines 12-17: "On the other hand, the increased amount of ancestry related to West and Central African groups (type (b) clusters) suggest independent recent DNA contributions (<4200 years ago) from individuals carrying ancestry related to the migrations of Bantu speaking peoples from West and Central Africa into East and South Africa, which started around 5,000 KYA but are not believed to have reached present-day Ethiopia during the initial migrations (Vansina, 1995; Ashley, 2010; Clist, 1987)". Which initial migrations are they referring to?
4. Page 22 Lines 17-21: "Clusters predominantly consisting of Nilo-Saharan speakers are either in type (b) clusters or unclear, with each showing a relatively high degree of ancestry related to West/Central Africans and an inferred admixture date more recent than 1600 years ago, perhaps reflecting these groups recently intermixing with other non-Nilo-Saharan speakers in Ethiopia." Which other non-Nilo-Saharan groups are they referring to?
5. Page 19 Lines 11-13: "We provide evidence that these patterns may be explained in part by the recent intermingling of individuals from groups sharing the same cultural practices". Does intermingling refer to admixture or cultural exchange? Can you define this more clearly?
6. Pages 14-15 Lines 23-2. "Scholars have also proposed possible genealogical relationships with the Beta Israel and/or Agaw (Legesse, 2013); we observe a high genetic similarity between Negede Woyto and each of these groups (Fig S2b) though with some degree of recent isolation among them (Fig S2a)." Is this genetic sharing higher compared to sharing with other groups?
7. Your website(https://www.well.ox.ac.uk/~gav/work_in_progress/ethiopia/v5/index.html) has the potential to be very beneficial for other researchers. However, the labels on the graphs are too small and there should be some added documentation on how to make comparisons between groups. Is it possible to download any data or can you only look at the graphs?
8. Page 27 Lines 18-19: "In each analysis, for painting (2) we used ten painting samples inferred by CHROMOPAINTER per haploid of each target individual". Why did you choose 10 painting samples?
9. Page 11 Lines 4-11: "Strikingly, the SOURCEFIND-inferred proportion of DNA that each Ethiopian cluster matches to the 4,500-year-old Ethiopian Mota also showed a significant (linear regression p-value < 0.0001) decrease with increasing spatial distance between the average location of individuals in that cluster (based on birthplace) and where Mota was discovered (Fig 4cd, Table S7). This is consistent with the preservation of substantial population structure over the past 4.5k years in the region of the Gamo Highlands where Mota was found (Gallego-Llorente et al., 2015, Gopalan et al., 2019)." Can you place this degree of population continuity in context with other examples of population continuity in the literature, such as Posth et al 2018? Is this the longest example of population continuity in Africa?
10. Page 13 Lines 16-18: "Meanwhile AA Cushitic and AA Omotic speakers display a wide range of inferred dates and admixture proportions that fall between those inferred for NS and AA Semitic speakers (Fig 3b, Extended Table 5)." Expand on why this is important.
11. Pages 32-33 Lines 22-2: "However, we instead use our approach of averaging $(1 - TVD_{ij})$ across individuals because of the considerable reduction in computation time when performing large numbers of permutations." Do you expect this to change your final results? If so, by how much? This is a rather ad hoc method for analyzing genetic distance between populations. It's not clear why other more standard approaches weren't used.
12. Page 2 Line 7-10: "we disentangle the effects of geographic distance, elevation, and social factors upon shaping the genetic structure of Ethiopians today. We provide examples of how social behaviours

have directly --and strongly --increased genetic differences among present-day peoples." This language is too strong ("disentangle" and "directly" and "strongly") given the many uncertainties discussed above.

13. Page 17 Line 5: "Therefore, to test for evidence of such recent intermingling between two". I think this is supposed to be "intermingling"

14. Page 28 line 19: "For all individuals we recorded their, their parents', paternal grandfather's". "their" is repeated twice

15. Page 30 Lines 8-9: Specify the exact file name that was downloaded from this ftp server "genetic map build 37 available on the 1000 Genomes Project website (<http://ftp.1000genomes.ebi.ac.uk/vol1/>)."

16. Page 6 Line 12: "For a set of pre-defined groups (see below)". Please specify the exact section that describes these groups

17. Page 9 Line 19: Cite a source that supports the use of a 28 year generation time. This is a very reasonable generation time, however there should be a cited source that supports this value.

18. Note that the anthropological literature refers to the "Shabo" as "Chabu". Shabo is considered derogatory.

Reviewer #3:

Remarks to the Author:

The manuscript by Lopez et al analyze the genetic landscape of Ethiopia by genotyping over 1,200 Ethiopian individuals and analyzing the data for population structure and admixture, using mostly chromopainter based techniques. They analyze the genetic data with respect to geography, linguistic data and various socio-cultural aspects. They define the most important factors governing population structure within Ethiopia by Ethiopia internal analyses. They also infer demographic history of Ethiopia in relation to external groups and identify and describe important past migrations into the region. The analyses are robust and very detailed and interesting novel aspects governing genetic structure in Ethiopia were uncovered and discussed. The manuscript is well written and the ideas and conclusions are explained well. However, I still felt there is some room for improvement with regards to some additional explanation, clarification and discussion. I added detailed comments below, which the authors may consider to improve clarity and information in the manuscript.

Comments:

1. In general the article feels very detailed and long. It is nice with detailed and well-motivated results and discussion, however, perhaps some whole sections can be moved to supplementary notes and just referenced with a line or two from the text.

2. At the start of the result section it feels like a lot of methods are explained – which can be shortened and moved to the method section

3. On page 8 line 11-13 Regarding the way the populations in Figure 3B were sorted and the sentence:

"Broadly, ancestry inference places the Ethiopian clusters on a northeast to southwest cline corresponding to increasing ancestry related to West African groups and decreasing ancestry related to present-day Egyptian groups, consistent with geography (Fig 3)."

This sorting feels very subjective. The authors don't support the correlation of these components with geography statistically and this correlation is also not very apparent when just looking where

populations are on the map in Figure 3A. If you sort by any component in the clustering analysis you will see a cline and the authors don't motivate why they chose West Africa and Egyptian components.

4. Page 12 lines 12-17. Are the admixture date inferences similarly sensitive to detect older dates compared to recent dates?

5. Page 14 line 7-14. The authors should outline more clearly how their inferences about the Chabu compare with Gopalan 2019

6. Regarding the sentence "The other group with uncertain linguistic classification in our study, the NegedeWoyto, are significantly differentiable ($p\text{-val} < 0.001$) from all other ethnic groups under the "Ethiopian internal" analysis, and cluster separately (Fig 2b, Fig S2a, Fig S3)."

In figure 2b this group have very similar cluster assignments to the groups that surrounds it on the clustering analysis plot.

7. P18 line 5. "In particular we show a strong concordance between genetic differences and geographic distance among individuals (Fig 4), similar to that shown previously among peoples sampled from European (Novembre et al., 2008, Leslie et al., 2015) and worldwide countries (Li et al., 2008)."

Are there perhaps some values to compare whether the correlations are similar magnitudes? I.e. Procrustes factors or mantel tests?

8. P18 line 8. I was surprised to see Mota quoted here as a farmer individual – is this correct?

9. RoH and selection scans are not included in the results section but only discussed in the Discussion section. Please introduce results first in result section and then discuss in discussion section.

10. Page 21-22. Results of the Ethiopian external analysis are discussed extensively in comparison to previously published results – however no mention or reference is made to the Prendergast 2019 aDNA study. This study on ancient HG, pastoralist and farmers from East Africa made various hypotheses about the movements of North African/Levant components into East Africa in relation to the introduction of pastoralism. These findings are important to discuss in the context of the current work. Also the current study only included the Mota aDNA individual and include none of the aDNA data for East Africa published by Skoglund 2017 and Prendergast 2019.

11. In the materials and methods it says the Malawi samples from Skoglund et al were included but they are not shown in the map.

12. For the PCAs in the supplement please include the variation explained by each PC. Also showing more PC's than just the first two can be very interesting regarding additional population structure

13. Please explain clearer in the introduction or discussion that the occupation specialization does not apply to all population groups in Ethiopia

14. It was not clear whether authors controlled for possible influential co-variates (e.g. language, geography) when correlation with cultural traits were tested.

15. Some of the colors on figure 1 are too close to each other and are difficult to distinguish in print

16. Figure 3. It is not clear how the number of sources to base the clustering on was chosen. From comparison to the box in figure 1b it seems the top 8 were chosen – but it is not clear why the cutoff was at 8. For instance, it caused the central Afr HGs to be grouped with the "other" fraction – which could have been interesting to have separate

17. Figure 4. It would be nice to include information on the gradients of the lines together with the residuals, in addition to p-values.

18. Figure 6 – Referring back to my comment 14 – it was not clear if confounding effects were controlled for here

19. Figure 7 – please add information on sample sizes. Again - referring to comment 14

We thank the reviewers for their comments, which we believe have greatly strengthened the manuscript. The changes we have made include re-doing all analyses after incorporating recently published relevant samples suggested by Reviewer 3, re-making every figure in light of new results, and adding a new set of simulations supporting some of the key insights we make (and important limitations). Furthermore, we have extensively re-written the paper to focus on these key points. While all conclusions are the same as in the previous submission, we believe our findings are now conveyed more clearly. We address and reply to all individual comments below, with reviewer comments in bold and our responses underneath.

Reviewer #1 (Remarks to the Author):

The genetic landscape of Ethiopia: diversity, intermixing 1 and the association with culture

The authors have assembled genome-wide SNP array dataset from 1268 Ethiopians representing 68 ethnic groups and characterized the impact of geographic and cultural affinities on genetic substructure across groups. Using previously published methods, like fineSTRUCTURE, SOURCEFIND and Globetrotter, the authors characterize the population history of Ethiopians. They also investigate signals of selection and adaptation to high altitude in Ethiopians. The analysis performed is compelling but technical concerns that I discuss below.

Major concerns:

1. Dates of admixture - In the analysis for dating admixture, the authors report the timing of admixture using Globetrotter. In previous work, this method has been tested for dating mixtures involving up to 3 ancestral populations. The Ethiopian samples that the authors analyze have ancestry for 4-6 ancestral groups. Thus, its not clear which mixture is being dated and reported here. More generally, its not clear if Globetrotter is applicable in this setting and gives unbiased results. Indeed, a recent paper explores this scenario and concludes that current method are inapplicable for dating admixtures in case of >2 admixture events (Chimusa et al. 2018, <https://www.ncbi.nlm.nih.gov/pubmed/30462157>). Thus it would be useful if authors can provide simulations mimicking the population history of Ethiopians (4-5way mixtures) and showing that the method provide reliable results for the timing of multi-way mixtures.

We have now included new simulations (SI section 3, Fig S5) that incorporate demography and multi-way mixture among Ethiopian-like populations. Also, as you suggest, GLOBETROTTER is only attempting to date up to 2 admixture events and describe at most 3 admixing sources. The confusion is that each inferred source is described as a mixture of reference populations. To make the inferred admixture for each Ethiopian cluster clearer, we have re-designed Figure 3 (also new Fig S11). In particular we enclose the reference populations representing one of the inferred admixing sources with a thick blue line. In Ethiopian groups with >2 inferred sources, we also enclose the reference populations representing the second source with a thick green line. Also in Fig 3 and Fig S11, we have denoted six different categories of admixture defined by inferred sources, summarised in the main text and SI section 5:

1. one-date of admixture between 3 sources related to (i) Nilotic-speaking Sudanese Dinka, (ii) Bantu-speaking Kenyan Sengwer and (iii) other Bantu-speaking Ugandan Buganda
2. one-date of admixture between 2 sources related to (i) the 4.5kya Ethiopian Mota and (ii) Dinka+Sengwer
3. one-date of admixture between 2 sources related to (i) Cushitic-speaking Rendille and (ii) either

Mota or Sengwer

4. one-date of admixture between 3 sources related to (i) Rendille and (ii) Mota and (iii) Sengwer + Dinka
5. one-date of admixture between 3 sources related to (i) Egyptians and (ii) Rendille and (iii) Mota
6. either one or multiple-dates of admixture between 2 sources related to (i) Egyptians/West Eurasians and (ii) Rendille + Mota

As illustrated in Fig 3 and Fig S11, (1)-(6) show clear geographic and linguistic trends, e.g. with Nilo-Saharan speakers primarily falling into (1)-(2) and northeastern Ethiopians (including all Afroasiatic Semitic speaking groups) primarily falling into (6). We also highlight limitations of these approaches in our revised Discussion: “Simulations mimicking the admixture inferred here show high accuracy in inferred dates and sources, though illustrate an inherent limitation that older dates of admixture (e.g. those reported in Prendergast 2019) may be masked by more recent admixture (Fig S5). Thus complex intermixing events, such as those exhibited here, can be difficult to dissect fully with these approaches and sample sizes, e.g. distinguishing between multiple pulses or continuous admixture. A potential example are the NS-speaking Berta (clusters 49, 50), in which we infer only a single recent date of admixture but whom have complicated sources of ancestry that suggest multiple events (Fig 3, Fig S11).”

2. Analysis of “Ethiopian-internal” and “Ethiopia-external”

In order to study how internal and external populations have impacted the population history of Ethiopians, authors perform analysis of GLOBETROTTER and SOURCEFIND under two settings “Ethiopian-internal” and “Ethiopia-external” using ancestral groups either from inside or outside Ethiopia. While this is an interesting idea, the interpretation of the results is less clear. For instance, the internal analysis should largely pick up Identity by descent sharing across Ethiopians groups and hence founder events in Ethiopia vs. admixture between various Ethiopian groups. Here again it would be useful if authors can perform simulations and highlight how the history of founder events and endogamy within Ethiopia would impact the results. Comparing these Ethiopia internal results to IBD analysis will also be helpful to understand the source of the population structure.

We have now performed simulations that mimick endogamy by splitting a previously admixed population into two, and having one of the new populations experience a strong bottleneck, i.e. simulated populations “40% (Exp)” and “40% (BN)” in SI section 3 (Fig S5A). As expected, our results show major differences between the “Ethiopia-internal” and “Ethiopia-external” analyses, namely in that there is very weak genetic similarity between the two under the former and much stronger similarity under the latter (Fig S5B). Furthermore, the two simulated populations infer similar (and accurate) admixture dates and sources under the “Ethiopia-external” SOURCEFIND and GLOBETROTTER analyses (Fig S5C-D). As the reviewer suggests, genetic similarity patterns under our “Ethiopia-internal” analysis (Fig S8A) are associated with within-group IBD results (Fig S3A), for example with populations Negede-Woyto, Chabu and Ari Smiths that show low genetic similarity with other groups under “Ethiopia-internal” having the highest IBD sharing.

3. Selection scan

The details of the selection scan applied to identify signals of adaptation seems unclear to me.

Given that Ethiopian populations are so structured, how was this analysis performed? In particular, how was population structure accounted in selection scan - since long haplotypes and increased F_{st} differences would be expected due to admixture alone.

Following these comments and those of Reviewer 3, we have removed the selection analyses from this paper, and will make this part of a future manuscript.

Minor concerns:

1. Figures are too crowded and often hard to follow - would be useful to split these into multiple figures so that the results are clearer.

We have greatly simplified all main text figures to convey the main points, reducing the number of main text figures from 7 to 5 and cutting out several panels. In particular we greatly simplified Figure 2 and Figure 3, and now no main text figure contains more than two panels.

2. Discussion is too long and summarizes large parts of the results. Would be useful to cut down a little so that message is clearer.

Our Results and Discussion sections have been reduced from 5077 to 3853 words, and we have taken care not to repeat information.

3. Figures - reference populations used for analysis, say GLOBETROTTER, etc. are often unclear and not specified in the legend.

We have now specified the surrogate populations that were found to contribute >5% to any Ethiopian cluster in the SOURCEFIND analysis using colors at the bottom of Fig 3 (and Fig S11). In the top-right panel of our revised Fig 3, we show a subset of the 284 reference groups used, with this number mentioned in the legend now, mainly highlighting the reference groups in Africa.

Reviewer #2 (Remarks to the Author):

López et al analyzed a large SNP array dataset genotyped in >1000 Ethiopians from 68 ethnic groups. They used Chromopainter and fineSTRUCTURE to analyze genetic structure and Globetrotter to analyze sharing of haplotypes between Ethiopian groups and between Ethiopians and a reference dataset from global populations. This allowed them to make inferences about recent migration events between Ethiopian groups and older migration events with non-Ethiopian populations. They present an interactive website to view sharing of haplotypes between groups. The authors also analyze correlations between genetic, geographic, linguistic and cultural variation. This is a large and impressive dataset from ethnically diverse African populations, which are under-represented in human genetics research and can potentially be an important resource if individuals are properly consented and data is shared. The analysis of genetic and cultural diversity is also of considerable interest. However, there are many serious concerns about the study, which would need to be addressed before a manuscript could be published in this or any journal.

Major Concerns:

1. Informed consent and ethical considerations of this study

a. Page 28 Lines 11-17. What is the informed consent obtained from these study participants? Documentation must be provided indicating whether all individuals included in the study signed consent forms. Are these consent conditions consistent with the public release of these data and analyses performed on these data?

b. Page 6 Lines 2-4: “including 8 unpublished Kotoko individuals from Cameroon, 20 Moroccan Berbers, 13 individuals from Senegal and 11 Chagga individuals from Tanzania (Fig 1b, 3Table S2).” There is no mention of research permits from Cameroon or the other countries. Again, demonstration of documented informed consent for every individual is critical.

The (anonymised) samples analysed in the paper are part of an extensive collection sponsored over many years by Dr Neil Bradman that commenced in 1996 for use in the training of undergraduate, masters and PhD students and post-doctoral researchers in genetic anthropology in Ethiopia, Cameroon and the United Kingdom. Ethical approval for collection and analysis of samples was granted by The Joint UCL/UCLH Committees on the Ethics of Human Research 99/0196. At the time some samples analysed in this study (e.g. Cameroon, Senegal, Tanzania and Morocco) were donated there were no applicable local ethical approval procedures. Where local regulations existed, authority was obtained from relevant offices e.g. Cameroon, Ministry of Higher Education and Scientific Research, Permits 0188/MINREST/B00/D00/D10/ D12 and 317/MINREST/B00/D00/D10 and University of Yaounde 1; Ethiopia, Ethiopian Science and Technology Commission.

All collections were made with the informed consent of the study participants, with assurance that there would be no commercial use of the samples. No financial payment was made to donors of samples. Attached is an English translation of information provided to sample donors in Ethiopia (“Ethiopia Consent Explanation”). Note the explanations provided to participating individuals: “The purpose of this study is to investigate the genetic histories of human populations, and to describe patterns of genetic variation within and among human populations”, “the samples cannot be used for any commercial enterprise” and “any genetic material derived from the samples is retained by the biological owner, and cannot be patented”. Similar explanations in Amharic and local group languages were provided both in Ethiopia and elsewhere.

A list of peer reviewed publications in which analysis of samples in the collection, from all of these countries, are included is attached. This includes analyses of data from seven additional populations we are now including in this revised version: Israeli Arabs (Ingram et al 2007), Israeli Bedouins (Ingram et al 2007), Palestinian Arabs (Ingram et al 2007), Saudi Arabian Bedouins (Ingram et al 2007), Syrians (Thomas et al 2002) and Uganda Muganda/Mussese (Veeramah et al 2008).

In 2014 prior to commencement of the project, ethical approval was sought from and granted by UCL Research Ethics Committee (reference number: 5188/001, copy of approval attached). We have updated the Acknowledgements to reflect this.

c. Page 50 Lines 5-8: “Genotype data, birthplace information and self-reported group label, first language, second language and religious affiliation for each individual are available at the Gene Expression Omnibus database online (www.ncbi.nlm.nih.gov/geo), series accession number XXXX.” The Gene Expression Omnibus accession number for access to the data is not listed here. These data must be made publicly available with a clear description of data use restrictions prior to publication. Otherwise, the importance of the study is greatly reduced.

We will make all new data available upon publication. Reanalyses we have done based on the

suggestions of reviewers have slightly altered the QC done for these data, which will now be made available by David Reich (<https://reich.hms.harvard.edu/downloadable-genotypes-present-day-and-ancient-dna-data-compiled-published-papers>) in combination with other world-wide samples to be released on the Human Origins array used in this study. People who access the data will be required to state that they do so for non-commercial use and that no attempt will be made to identify study participants.

d. Page 28 Lines 14-17: I was not able to find this REC record. Could you include a link to the appropriate website to identify the REC? I checked on this website: <https://www.hra.nhs.uk/about-us/committees-and-services/res-and-recs/search-research-ethics-committees/>. This needs to be easily accessible. A direct link to this record in addition to it's description would be helpful.

We have provided a copy of the UCL REC.

Page 5 Lines 1-6: “Furthermore, some of the authors of this study (A. Tarekegn, N. Bradman) translated into English and edited a compendium (originally published in Amharic) that documented the oral traditions and cultural practices of 56 ethnic groups of the Southern Nations, Nationalities and Peoples’ Region (SNNPR) of Ethiopia through interviews with members of different ethnic groups (The Council of Nationalities, Southern Nations and Peoples Region, 2017).” Given that the cultural classification are central to the analyses in this study, the methods should be presented and they should provide a copy of this document which, ideally, should be assessed by a cultural anthropologist/linguist for accuracy.

An important objective of the study is to assess whether the genetic data correlates with cultural information obtained from elders of groups as recorded and published prior to the commencement of this study. The data is published in English and Amharic in hard copy publications, references to both, have now been added:

A Profile of the Nations, Nationalities and Peoples of the Southern Region, Second Edition (December 2004EC), Translated from Amharic and sub-edited by Ayele Tarekegn and Neil Bradman, Published by the Council of Nationalities with financial support from the German Co-operation (GIZ), Addis Ababa: Berhanena Selam Printing Enterprise, April 2017 CE.

Council of Nationalities (CoN), Southern Nations, Nationalities and Peoples’ Region (SNNPR), (in Amharic). Yedebubbiheroch, biheresebochna Hizboch Profile (title translated as A Profile of the Nations, Nationalities and Peoples of the Southern Region), Third Edition, Published by the CoN, SNNPR, Addis Ababa: BerhanenaSelam Printing Enterprise, Miazia (April) 2008EC.

e. Page 11 Lines 1-4: “However, we found evidence ($p\text{-val} < 0.05$) of genetic isolation between Christians and either Muslims or people practicing traditional religions within seven of 16 groups for which we sampled at least five individuals from each religion (Table S6).” It is not clear how strong this “genetic isolation” is (this is where standard F_{st} analyses would be helpful) or how you are defining people practicing traditional religions. It is very common that indigenous Africans will practice traditional religion in addition to identifying as Christian or Muslim. Further, the religious classification is likely to be highly correlated with ethnicity and language classification. Indeed, it would be informative to know how they correlated with the “clusters” inferred from fineSTRUCTURE. If those are controlled for, do you still see differences between groups based on language? Analyses of genetic and religious affiliation (in addition to many of the other cultural practices they describe) has the potential to be highly inflammatory in Ethiopia, a region that is currently experiencing conflict between ethnic groups. I would want to know that individuals gave consent for that analysis and that it

was included in the ethical review at the country level. A copy of the informed consent form used would help clarify.

The test mentioned here (reported in the new Table S8) looked separately at individuals within each ethnic group, exploring genetic affinities among individuals who reported the same versus different religious affiliations. As each ethnic group is classified into a linguistic category, this analysis accounts for ethnicity and language. Similarly, our permutation tests reported in (the new) Fig 1b account for religious affiliation when testing for associations between genetics and ethnicity and genetics and language (and vice versa), excluding “Traditional” due to the many different Traditional practices. For example, to account for religious affiliation when testing for an association between genetics and ethnicity, we only permute individuals' ethnic classifications within each religious affiliation (Christian, Jewish, Muslim).

We have amended the text to make it clear this is testing for relationships among peoples' self-reported religious affiliation: “However, within six of 16 groups for which we sampled at least five individuals from different religions, we found some nominal evidence (permutation-based p -val < 0.05) of genetic isolation between people reporting as Christians versus those reporting as Muslims or those reporting as practicing traditional religions (Table S8).”

While such analyses were the motivation for this sample collection, with the consent noting that we would “describe patterns of genetic variation within and among human populations” using their reported information, we acknowledge the potential sensitivities of such studies. After our preliminary analyses of these data, we gave lectures at eight Ethiopian universities in Adama, Addis Ababa, Arsi, Axum, Gondar, Haramaya, Harar and Hawassa, plus a public lecture at the Ethiopian Academy of Sciences (Addis Ababa). We discussed specifically these issues with several leading Ethiopian scientists and public members, and will continue to engage in these dialogues in Africa and elsewhere, noting that such issues are obviously relevant outside of Africa as well (e.g. research being co-opted by white supremacist groups in America: <https://www.nytimes.com/2018/10/17/us/white-supremacists-science-dna.html>).

2. There are multiple instances in this paper where there are missing citations directly relevant to their study. For example, Scheinfeldt et al. 2012 presented a study of adaptation to high altitude in Ethiopia (page 3, lines 12 – 14). They should cite additional studies in Africa (including Ethiopia) that have examined correlation between genetic, geographic, and/or linguistic diversity (e.g. Tishkoff et al. 2009, Verdu et al. 2017; Baker et al. 2017; Scheinfeldt et al. 2019, etc) when they describe similar results (e.g. page 3 lines 19 – 21, page 13 lines 7 – 9, page 18 lines 3 – 6, page 19 lines 3 -4 and 5 – 8).

Thank you for pointing out these important omissions. We have added a citations for these to supplement those we already had. In particular:

For Tishkoff et al 2009: “Both analyses show a strong concordance between genetic differences and geographic distance among individuals (Fig 1b, Fig S6), similar to that shown previously among peoples sampled from European (Novembre et al., 2008, Leslie et al., 2015), African (Tishkoff et al 2009, Scheinfeldt et al 2019) and worldwide countries (Li et al., 2008).”

For Scheinfeldt et al 2012: “The high genetic diversity in Ethiopians facilitates the identification of novel variants, and this has led to the inclusion of Ethiopian data in studies of the genetics of elite athletes (Rankinen et al., 2016, Ash et al., 2011, Scott et al., 2005), adaptation to living at high elevation (Huerta-Sanchez et al., 2013, Stobdan et al., 2015, Simonson, 2015, Ronen et al., 2014, Scheinfeldt et al 2012),..... ”

For Baker et al 2017, we added a citation when mentioning studies of genetic and linguistic data in Ethiopia: “Although as early as 1988 Cavalli-Sforza et al. (1988) drew attention to the importance of bringing together genetic, archaeological and linguistic data, there have been few attempts to do so in studies of Ethiopia (Boattini et al., 2013, Pagani et al., 2012, Gallego-Llorente et al., 2015, Baker et al 2017, Scheinfeldt et al 2019).” In Results we have also now added a comparison: “We also find individuals from the three AA classifications of Cushitic, Omotic, Semitic to be on average more genetically similar to one another than to NS speakers (Fig 2b, Figs S8-S9), contradicting previous suggestions that Omotic speakers should be an outgroup to AA (Baker et al 2017).”

For Scheinfeldt et al 2019, we added the above citation. The main text also contains: “The Chabu results are consistent with previous findings from genetics (Scheinfeldt et al., 2019, where the Chabu were referred to as Sabue) and linguistics (Blench, 2006; Ehret, 1992).”

As Verdu et al 2017 considers correlations between genetics and linguistics in non-Ethiopian groups, we did not find an obvious place to cite it after we substantially re-structured our manuscript.

Beyond these suggested citations, in our revised Discussion and SI section 5 we compare our ancestry inference to that reported for overlapping Ethiopian groups in Prendergast et al 2019.

We also added an additional reference to Gopalan et al 2019 regarding Ari occupational groups: “This corresponds to the time period during which iron working is believed to have first appeared in Ethiopia (Phillipson, 2005) and supports the marginalisation theory of their origins (Lewis, 1962), which is consistent with findings from previous genetic studies (van Dorp 2015, Gopalan 2019).”

3. The use of ethnic groups and clusters used in the study is not always clear. It seems to change depending on the analysis. This needs to be clarified and explained why in some cases they use ethnic groups for analyses and others they refer to clusters. Furthermore, they identify 78 clusters and have 68 ethnic groups. If they are separating ethnic groups into essentially their own clusters, then how is this analysis informative? Is your approach overfitting the data given that there are more clusters than ethnic groups? Looking at the large supplemental tree (Figure S3), how confident are you that each node in your tree is distinct?

We apologize for the lack of clarity. Self reported ethnicity is used in most analyses presented here, e.g. for inferring genetic distances among groups and correlations with genetics and linguistics, ethnicity, religious affiliation and cultural practices. However, when identifying/dating admixture events and inferring ancestral sources, we first classified individuals into genetic clusters with relatively homogeneous ancestry, and then analysed each cluster. We did so to avoid complicating these results, as individuals with the same self-reported ethnicity can have disparate ancestral backgrounds. As we now note in the main text: “Unsurprisingly, given genetic similarity observations (Fig 1b, Fig S6, Fig S8), these clusters were notably associated with ethnic label (Fig S10). However, using clusters rather than self-reported label can increase power by merging ethnic groups with similar genetics. This also can clarify ancestry inference, as it does not assume that all individuals reporting the same ethnicity share recent ancestry.”

As the clustering used is based on a Markov-Chain-Monte-Carlo procedure, one way of measuring whether nodes are distinct is considering how often individuals assigned to different clusters in our final assignments actually fall into the same cluster across MCMC samples. As we now note in Material and Methods, for the 70 final cluster assignments with ≥ 5 individuals, on average individuals from different clusters fell into the same cluster in 0.7% of MCMC samples. In contrast,

for individuals assigned to the same final cluster, this average is 32%.

4. Page 25 Line 24 to Page 26 Line 3 “When testing for admixture in each Ethiopian cluster under the “Ethiopia- internal” analysis, we used all other Ethiopian clusters (excluding 11 clusters – those with the suffix ‘adm’ in Extended Table 2 – that each consisted of small numbers of individuals from multiple ethnic groups)”. Some clusters have very small sample size (some are N = 1), what was your sample size cutoff when excluding clusters from an analysis. For analyses where you did not exclude clusters, do you expect the small sample size to impact your results?”

Smaller sample size will affect power to see signals, though our previous work has shown that one individual is sufficient to accurately infer the dates and sources of admixture in cases of e.g. recent mixture among continental groups (Chacon-Duque et al 2018). However, here one sample is too few, as we performed an analysis to account for isolation effects (e.g. due to consanguinity) that requires at least 2 individuals in the target population. We now make this clearer in the text, e.g. “GLOBETROTTER infers clear admixture events in 67 of the 77 Ethiopian clusters containing more than one individual,…”

5. This paper is based on analyses of haplotype sharing between populations using Globetrotter. How would other standard approaches compare to the analyses presented here (e.g. ADMIXTURE, Fst, and f4/D statistics)? They present ROH and IBD analyses in a Figure but only discuss in the methods. This should be discussed in the results. What do these results imply about inbreeding and relatedness among individuals in the dataset? The hunter-gatherer populations appear to have high IBD and ROH. How might that effect the admixture analyses?”

From the Reviewers recommendation, we have discussed ROH/IBD in the main text and included F_ST results for comparison (Extended Table S9). Simulations and analyses from our previous work demonstrate that high within-group IBD does not notably affect admixture date or source inference, in part because our test has been designed specifically to account for this. For example, our previous analyses (van Dorp et al 2015) and results reported here for the Ari Blacksmiths demonstrate a high degree of IBD and ROH consistent with recent isolation relative to the Ari Cultivators. Nonetheless both Ari Blacksmiths and Cultivators show near identical admixture inference when analysed independently (e.g. Fig 3, and see our new simulations mimicking these groups in Fig S5). As we now note in SI section 5: “In general we note that surrogate populations with high degrees of isolation (e.g. due to endogamy) may be less likely to be selected as representative of an ancestral source, which is one way SOURCEFIND likely differs from commonly-used f3/f4 statistics (Patterson et al 2012). But arguably such surrogates should be downweighted, as – due to recent isolation – the genetic make-up of such surrogates likely no longer well-reflects the ancestral source population.”

We avoid ADMIXTURE analyses here, as these have previously been shown to be prone to over-interpretation (e.g. van Dorp et al 2015, Lawson et al 2018, Chacon-Duque et al 2018), including in the context of Ethiopian populations (van Dorp et al 2015), as has F_ST (van Dorp et al 2015).

6. How does demographic history impact your inference of migration timing with Globetrotter? They mention that Globetrotter assumes single pulses of migration for up to two events, which is then followed by random mating. What happens if this model is not correct? Indeed, prior studies have shown that many ethnic groups in Ethiopia may have 3-way admixture. Yet, it appears that they only considered 2-way admixture. Further, they

assume a single initial pulse of admixture followed by constant gene flow. But that is unlikely to be the case for all of the populations included in this study. Additionally, they mention on page 10 of the supplement that 26 of 53 clusters' admixture patterns are unclear. Isn't this a problem for interpreting their results?

GLOBETROTTER classifies inferred admixture events into five possible categories: (i) one admixture date between 2 sources, (ii) one admixture date between 3+ sources or (iii) two admixture dates between 2 or more sources, (iv) no evidence of admixture or (v) unclear signals. In the cases of (ii) and (iii), due to the challenges in disentangling multiway admixture we attempt to only describe at most 3 sources, even though the admixture may be more complicated. To make our admixture inference for each Ethiopian cluster clearer, we have re-designed Figure 3 (and a new Fig S11) to illustrate which Ethiopian clusters show 2 versus 3 sources intermixing, and further separated clusters by which infer similar admixing source groups (see Response to Reviewer #1).

The GLOBETROTTER (and related ALDER/MALDER/ROLLOFF approaches) are similar in spirit to approaches that identify and date admixture by assigning individual tracts of DNA in target individuals to surrogates (e.g. RFMix), while being robust to tract mis-specification. These tract-identification approaches make the same assumption of admixture pulses followed by random mating. We have explored how bottlenecks and population expansions affect our inference, both previously (Hellenthal et al 2014) and in newly added simulations here (Fig S5, described in SI section 3). We have tried to clarify the writing, e.g. removing the statement referred to here. We also highlight limitations of these approaches in our revised Discussion: “Simulations mimicking the admixture inferred here show high accuracy in inferred dates and sources, though illustrate an inherent limitation that older dates of admixture (e.g. those reported in Prendergast 2019) may be masked by more recent admixture (Fig S5). Thus complex intermixing events, such as those exhibited here, can be difficult to dissect fully with these approaches and sample sizes, e.g. distinguishing between multiple pulses or continuous admixture. A potential example are the NS-speaking Berta (clusters 49, 50), in which we infer only a single recent date of admixture but whom have complicated sources of ancestry that suggest multiple events (Fig 3, Fig S11).”

7. Page 7 lines 2 – 6: “Importantly, if two clusters infer admixture events with matching dates, sources and proportions, a parsimonious explanation is that the ethnic groups contained in these clusters were a single combined population when this admixture occurred, hence placing an upper bound on their split time (van Dorp et al., 2015, van Dorp et al., 2019). Conversely, if two clusters show different admixture inferences, they were likely not a combined population when the most recent admixture occurred.” The authors state that if two populations share a similar ancestry and time of admixture, they most likely share a common ancestry and were “a single combined population”. Indeed, this is the crux of the entire study. But how do we know that there isn't another model that's equally as likely? Again, without modeling the many possible demographic scenarios that could also explain this pattern, it's not possible to know this and, therefore, the conclusions are over-stated. It should be noted that in the van Dorp 2019 paper, they do simulations based on testing models specific to populations in the DRC.

We agree that other scenarios can explain these findings. The key point is that this is a parsimonious explanation (and hence can be thought of as more likely), requiring only that they were a single source when the admixture occurred. We have now done simulations illustrating this point (see new SI section 3). In particular we simulated an admixed population (with dates matching those inferred in the Ari here) that subsequently split into two groups that experienced a population expansion and a strong bottleneck, respectively, i.e. simulated populations “40% (Exp)” and “40% (BN)” in SI section 3 (Fig S5A). As expected, our results show major differences between the “Ethiopia-

internal” and “Ethiopia-external” analyses, namely in that there is very weak genetic similarity between the two under the former and much stronger similarity under the latter (Fig S5B). Furthermore, the two simulated populations infer similar (and accurate) admixture dates and sources under the “Ethiopia-external” SOURCEFIND and GLOBETROTTER analyses (Fig S5C-D).

8. Supplement Page 10: “A likely explanation for these type-(a) versus type-(b) patterns is that at least two separate types of intermixing occurred in the ancestors of present-day Ethiopians. The first involved older intermixing between a Sub-Saharan African source versus a West-Eurasian source (type (a) clusters), after which these admixed ancestors intermixed again with individuals from West-Central Africa (type (b) clusters).” Again, are there other equally likely explanations? Without simulations and demographic modeling, this explanation is not convincing.

We agree and have removed this explanation in favor of something more focused on different numbers and types of ancestry sources, based on the new GLOBETROTTER/SOURCEFIND results presented in Fig 3 and Fig S11. See new Results section “The recent admixture history of Ethiopia”.

9. Supplement Page 11: “each of the 61 clusters that inferred admixture events classified into (1)-(3), i.e. excluding clusters that concluded “uncertain” admixture, we took the GLOBETROTTER inference of the most strongly signalled event (see Supp Table S5-S6) This event infers that two distinct sources intermixed, one contributing a majority of ancestry and the other contributing a minority of ancestry to the cluster’s individuals.” How was the “most strongly signaled event” determined? And again, what about the case when there may be more than two source populations contributing to admixture in a population?

We appreciate this was not clear and have re-written this part of SI section 5: “For each of the 61 clusters that did not infer “uncertain” admixture, we took the GLOBETROTTER inference of the most strongly signalled event (see Supp Table S5-S6). This is the event that explains the most variation in the set of GLOBETROTTER probability curve pairings across all reference populations (examples provided in Fig S14), i.e. as determined using the first (main) principal component of this probability curve set (see Hellenthal et al 2014 for details). (For clusters where admixture between more than 2 sources is inferred, the second, less clearly inferred admixture event is only partially captured by this first principal component if at all.) This event infers that two distinct sources intermixed, one contributing a majority of ancestry and the other contributing a minority of ancestry to the cluster’s individuals.”

10. The description of reference populations is confusing in this paper

a. It appears that for the GLOBETROTTER analysis there were 252 reference groups. Many of these are named by a country or broad geographic region and others by a particular ethnic group, making interpretation difficult. There are also many regions under-represented, particularly from Africa. With so many source populations, there are many potential combinations of admixture events. But it’s not clear how these are influenced by sample size (seems that reference group sample sizes range from 2 – 100) and the sample sizes of “clusters” within Ethiopia vary. Will differences in sample size impact your analysis when inferring the most likely source population for an admixture event? Also, how might these results be influenced by drift in small/inbred populations such as the “Shabo”, as seen in Figure S8?

To clarify, for simplicity reference groups were defined using population labels, and clusters were only used for Ethiopian samples. Surrogates of mixed ancestry may indeed confuse signals. All of our previous work, which included many thousands of simulations and hundreds of analysed groups (e.g. see Hellenthal et al 2014, Leslie et al 2015, Chacon-Duque et al 2018), suggests GLOBETROTTER is robust to population-specific drift when testing for admixture in small isolated populations (e.g. Ari Blacksmiths). Our new simulations mimicking admixture we observe in this study, and where we introduce a strong bottleneck via forward-simulating a population of size 100 haplotypes for 45 generations, reiterate this observation (Fig S5, SI section 3). Sample size may impact power, though as noted above, in certain cases we can identify admixture in single individuals (e.g. see Chacon-Duque et al 2018). However, in such cases we are unable to account for post-admixture drift; thus here we do not attempt to analyse clusters with only a single individual. Generally and importantly, our previous simulations (Hellenthal et al 2014) suggest that inferred proportions will be strongly influenced by admixture in the source populations, while estimates of the dates of admixture are not.

Motivated by the comments of this reviewer and others, we have incorporated an additional 119 published African samples from 27 present-day populations, including 23 East Africans from Ethiopia, Kenya and Tanzania (Mallick et al 2016, Lipson et al 2019). We have also added 74 unpublished samples from five Levant and Arabian groups. Including these additional data, out of 284 reference populations we use in analysis, only 11 contribute >5% to any Ethiopian cluster according to SOURCEFIND. Encouragingly, these 11 populations are all geographically proximal to Ethiopia, as illustrated in the top-right panel of Figure 3 and span sample sizes ranging from 1 (Mota) to 100 (Buganda) individuals. Even though these new populations feature prominently in our new GLOBETROTTER results, the general trends from our previous analyses still suggest greater Egyptian-like ancestry in northwestern Afroasiatic-speaking Ethiopian groups, relative to Bantu-like admixture in southwestern African (predominantly Nilo-Saharan-speaking) Ethiopian groups.

a. Importantly, nearly all the “source” populations in Africa are, themselves, admixed (Tishkoff et al (2009), Pickrell et al (2014), Pagani et al (2015), Schuenemann et al (2017)). How might your approach be impacted when estimating admixture times and admixture coefficients due to admixture in your reference populations?

In general, the inferred proportions of admixture will be strongly influenced by admixture in the reference populations, while dates are not (Hellenthal et al 2014). An exception is if a reference population has experienced very similar admixture to the target group, in which case admixture in the target may be masked. Though this is easily diagnosed – target groups will then match a very large proportion of their ancestry to this one group, which can subsequently be removed before re-analysis (though we do not do such re-analyses here). Admixture in surrogate groups is an ubiquitous issue when using present-day samples. For example, our previous work suggests that there are likely few, if any, extant human groups without evidence for recent admixture (Hellenthal et al 2014). In our revised main text, to provide a level of confirmation we compare recent ancestry composition results for three of our groups to the same groups analysed by Prendergast et al 2019: “Though using different surrogates and techniques complicates direct comparisons, our inferred sources of ancestry broadly agree with those reported for present-day Ethiopian groups by Prendergast 2019. For example, the Agaw (clusters 65, 67) have relatively more Levant-like ancestry (which we match most closely to Egypt), the Ari (clusters 22, 24, 27; called Aari in Prendergast 2019) have relatively more Mota-like ancestry and the Ethiopian Mursi (cluster 2) have relatively more Dinka-like ancestry (Fig 3, Fig S10). Simulations mimicking the admixture inferred here show high accuracy in inferred dates and sources, though illustrate a limitation that older dates of admixture (e.g. that reported in Prendergast 2019) may be masked by more recent admixture (Fig

S4).”

b. Why is the West Eurasian ancestry estimates presented in Figure 3B for the Amhara (~5%) so low compared to Pickrell et al (2014) (49.2%)?

Clusters 65/66 (Fig 3, Fig S11) contain several individuals who self-identify as Amhara. We infer 46-48% of ancestry related to Egyptians in these two clusters (see new Extended Table 5), similar to that observed in our original submission using different surrogate populations. Egyptians are closely related genetically to relevant West Eurasian populations, e.g. see Figure S13 clustering results of HGDP data in Hellenthal et al 2014, suggesting consistency with Pickrell et al 2014.

c. Page 8 Lines 8-11: “SOURCEFIND ancestry inference under the “Ethiopia-external” analysis (Fig 3) showed matching primarily to Mota and six major geographic regions to which at least one Ethiopian cluster matched 9>5% of their DNA: Egypt, East Africa, North Africa, Somalia, West Africa and West Eurasia (Fig 101b).” Again, there is a concern about the reference populations used. West Africa and East Africa are so broad as to be uninformative given the high levels of genetic heterogeneity in those regions.

We agree and have refined this to note the specific groups in the text and/or Fig 3 and Fig S11, given only 11 groups contribute >5% to any of the 67 Ethiopian clusters presented in those figures.

d. Page 21 Lines 2-7: “A caveat to the interpretation that groups with similar inference under the “Ethiopian-external” analysis share similar recent ancestry is that this analysis will have decreased (or have no) power to discriminate between Ethiopian groups that indeed have separate ancestral sources if we have not included relevant non-Ethiopian groups to represent these separate sources. The large number of non-Ethiopian groups included in this sample, particularly those surrounding Ethiopia, diminishes this possibility”. This is, indeed, a major potential problem and it is not correct that neighboring groups are heavily sampled. There are only four populations from Tanzania with sample sizes > 10 and there are no populations from Kenya included. The population from Somalia has only 13 individuals, yet many of the conclusions in this study are based on analysis of that population which is concerning. It is not clear how this very sparse sampling on other regions of Africa may bias the study.

Based on this suggestion, we have now included additional samples comprising six East African groups from Kenya (Elmolo, Kikuyu, Ogiek, Rendille, Sengwer) and Tanzania (Iraqw) using recently published data (Mallick et al 2016). As expected, ancestry proportions change, but the admixture dates and source interpretations remain similar to our original submission.

e. Page 21 Lines 13-14. “with results indicating intermixing between a source represented by (a) Sub Saharan African groups”. Sub Saharan Africa is too vague to be informative.

We have re-moved this, re-editing to describe admixture events in a different way as noted above.

f. Page 21 Lines 16-20: “Notably, Somalia differs among clusters in that it acts as a surrogate to source (a) in north/northeastern clusters with higher amounts of inferred ancestry related to Egyptian groups (type “a” clusters), while it acts as a surrogate to source (b) in west/southwest clusters with higher amounts of inferred ancestry related to East/West Sub-Saharan Africa (type “b” clusters) (Fig 3, Fig S9b-c, Table S10, Appendix B)” This is likely being impacted by Somalia being admixed with West Eurasian ancestry.

Similarly, we have removed this explanation of findings in our next text and figures, to try and simplify interpretations.

g. Supplement Page 9: “GLOBETROTTER concludes evidence of admixture in 69 of our 78 Ethiopian clusters, with inferred dates ranging from three to 118 generations ago. These inferred dates are robust across two alternative analyses that use different surrogates, one where we simply exclude Mota as a surrogate and another where we include only five modern groups as surrogates: Bantu_SA, Han, Japanese, Mende_Sierra_Leone_MSL, Yemen (Fig S9a).” Why choose these reference populations? They do not seem to fit well with the population history of Ethiopia.

We have removed this analysis in order to simplify the manuscript.

11. The large heatmaps are challenging to interpret. I also recommend moving Figure 4, 1B, and 5A to the supplement. They also need to define what the colors mean in the figure S3 caption.

We have moved Fig 4B-D to the supplement, simplified Fig 1B (and moved it to a panel of Fig 3), while also re-arranging and simplifying figures, ultimately reducing the number of main text figures from 7 to 5 and removing several sub-panels within figures that were in the original submission. We also edited the Fig S10 legend (formerly Fig S3): “Contiguous clusters of the same color were merged into one of the 78 final clusters we used in analysis; we alternate colors here to assist visualisation.”

12. Page 9 Lines 4-7: “For example, under the former analysis Ari and Wolayta people who work as cultivators or weavers are more genetically similar to members of other ethnicities on average than they are to people from their own ethnicities who work as potters, blacksmiths and tanners (top left of squares in Fig 2c)”. They make many statements like this throughout the manuscript. Their method of inferring genetic distances is rather “ad hoc” and it’s not clear why they didn’t use more traditional methods.

Our reasoning for focusing on haplotype-based genetic distance measures is because they have been shown to be more powerful than traditional approaches, e.g. when analysing African data (Busby et al 2016). However, we agree with the point that traditional techniques are more readily interpretable for most readers, and so we have included Fst metrics (Extended Table 9) to complement our findings.

13. Page 43 Lines 2-4: “In general disentangling whether there is significant evidence of independent effects for each of language, group label and spatial distance while also mitigating effects of recent isolation, though suggested in Table S4c-d, will require larger sample sizes than those considered here.” That is correct and it’s not clear how they have disentangled these factors. Therefore, the conclusions are not well justified.

Our point here is to note that it can be challenging to separate such factors due to their correlation. Nonetheless, our permutation procedures, designed to account for these correlations, often show statistical significance (see Table S6). To clarify these points, we have re-worded this sentence in Material and Methods: “Secondly, the high correlation between group label and first language (e.g. Fig S7ab) makes accounting for one challenging (in terms of loss of power) when testing the other. Furthermore, few permutations are possible when testing language group while accounting for group label (0 permutations available) or first language. Therefore, we excluded permutations fixing group and fixing first language when testing each of group, first language and language group when reporting our final p-values in the main text and Fig S6. Note we do observe a significant association with genetic similarity and ethnicity after accounting for spatial distance (geographic or elevation) and major language group, suggesting ethnicity explains genetic similarity beyond that of

classifications according to the second language tier of at Ethnologue. We caution that these analyses assume that the relationships among genetic, geographic and elevation distance can be modelled with simple linear or exponential functions, which is sometimes debatable (Fig S7cd), indicating larger sample sizes may reveal deviations from these assumptions.”

Also, in the main text we now write: “Under both the “Ethiopia-internal” and “Ethiopia-external” analyses, we found significant associations ($p\text{-val} < 0.05$) between genetic distance and each of geographic distance, elevation difference, ethnicity and first language, after controlling each factor for the others where possible (Fig 1b, Fig S6-S7, Tables S3-S7).”

14. Page 28 Lines 18-19: “Buccal swab samples were collected from anonymous donors over 18 years of age, unrelated at the paternal level.” How did you determine that individuals are not related at the paternal level given high incidence of non-paternity? Relationships should be inferred using the genetic data. Are their differences in the amount of 1st,2nd, and 3rd degree relationships in different populations? If so, then why wasn’t the dataset pruned to remove related individuals. How will including them influence the analyses?

Thanks for pointing this out. In the revised manuscript (“Materials and Methods”), we now make clear that we have removed individuals based on inferred IBD:

“To identify putatively related individuals, we used PLINK v1.9 (Chang et al., 2015) with “--genome” to infer pairwise PI_HAT values, after first pruning for linkage disequilibrium using “--indep-pairwise 50 10 0.1”. Instead of using the same fixed PI_HAT threshold value for all populations, we identified individuals with outlying PI_HAT values relative to other members of the same group label, in order to avoid removing too many individuals from populations with relatively low diversity. Specifically, we found all pairings of individuals from populations (i,k) that had $PI_HAT > 0.15$ and $PI_HAT > \min(X_i + 3 * \max\{0.02, S_i\}, Y_i + 3 * \max\{0.02, D_i\}, X_k + 3 * \max\{0.02, S_k\}, Y_k + 3 * \max\{0.02, D_k\})$, where $\{X_i, Y_i, S_i, D_i\}$ are the $\{\text{mean, median, standard deviation, median-absolute-deviation}\}$, respectively, of pairwise PI_HAT values among individuals from population i. For populations with ≤ 2 sampled individuals, the standard deviation and median-absolute-deviations are undefined or 0; therefore in such cases we added to the list any pairings with $PI_HAT > 0.15$ that contained ≥ 1 person from that population. Using a stepwise greedy approach, we then selected individuals from this list that were in the most pairs to be excluded from further analysis, continuing until at least one individual had been removed from every pair. This resulted in a total of 241 individuals removed, including 62 Ethiopians. (For comparison, all remaining Ethiopian pairs after this procedure had $PI_HAT < 0.2$.)”

15. The use of internal vs external analysis of reference populations reduces the impact of strong genetic drift. However, from the language used in this paper and prior papers, it is not clear by how much this approach removes this impact of drift: Page 6-7 Lines 23-3 “A key conceptual difference between the two analyses is that the “Ethiopia-external” analysis mitigates genetic differences between individuals attributable predominantly to recent isolation, e.g. endogamy, that can cause a groups’ individuals to match large proportions of their haplotype patterns to each other (van Dorp et al., 2015; van Dorp et al., 2019)” As you state in the Van Doorp et al (2015) paper: “The differences we observe between AR1b and AR1c under CHROMOPAINTER analysis (A), ADMIXTURE and FST are no longer present under CHROMOPAINTER analyses (B) and (C) (Figs 1b–1c and 3, S7 and S11 Figs and S10–S12 Tables). A key difference is that for each Ari group we disallow “self-copying” from individuals with the same label under analyses (B) and (C), which should reduce the magnitude of any differences seen in the other approaches that are attributable to bottleneck effects in the Blacksmiths.” Words such as “reduced” and “mitigates” need to be quantified. Have they performed simulations to determine the impact of strong drift on their analyses?

The new simulations we have included in the revised manuscript (SI section 3, Fig S5) demonstrate how the “Ethiopia-external” analysis mitigates endogamy (in this case strong bottleneck) effects relative to the “Ethiopia-internal” analysis (Fig S5B). The “Ethiopia-internal” and “Ethiopia-external” analyses use different surrogates, and hence the values (even comparing differences between two groups across the two analyses) cannot be directly compared. This makes vague terms like “mitigates” a necessary evil, at least without further research. Here we assess the degree to which groups that were previously distinguishable under one analysis are less so under the other, for example noting in the simulations (Fig S5) and real data (Fig 2a, Fig S8) how the population to which a group is most genetically related to can drastically change between the two analyses. For example, the Ari Cultivators are most genetically similar to Ari Smiths under the “Ethiopia-external” analysis, while they are much more genetically similar to groups other than the Ari Smiths under the “Ethiopia-internal” analysis (Fig 2a).

16. Page 18 Lines 16-22: An entirely new analysis of targets of natural selection cannot be introduced in the discussion section. Also, how much power do they have to identify selection signals? It was not sufficiently clear from the methods and sup material how these analyses were done and how to interpret them and, importantly, insufficient information about determining outliers. They appear to have pooled individuals into “high altitude” vs “low altitude” groups in an ad hoc manner without considering shared ancestry. Why not do something like a PBS analysis? These analyses were not well justified and should be removed from the paper.

To streamline the manuscript, and on the advice of the reviewer, we have removed the selection analyses from this submission.

17. Page 5 Lines 23-24: “We compared SNP patterns in each present-day Ethiopian to those in all other present-day Ethiopians and the 4,500 year-old Ethiopian sample “Mota” (Gallego-Llorente et al., 2015)” It would be useful to emphasize why the genetic ancestry of the Mota sample makes this individual an important reference group for your study, especially since this sample is often referred to elsewhere in the text. Also, what is the power/standard error for Chromopainter estimating ancestry contributions from only 1 individual? Could ancient DNA quality also impact this analysis?

The reason Mota is so important to our analysis is because it is the only available aDNA sample from Ethiopia and is also of high coverage (~12.5x). It was also reported by Prendergast et al 2019 to provide representation of a unique source of ancestry in East Africa and elsewhere. Using DNA from this individual in our analysis enables our results to be compared to the work of Prendergast et al 2019, especially now that we have deliberately used similar reference groups. Though we reiterate that our ancestry proportions only infer how best to form target Ethiopian clusters using the 284 surrogate groups we included here, rather than e.g. explicitly concluding these surrogate groups are separate admixing sources. The small sample size of Mota actually may favor it as an ancestry surrogate because we are unable to quantify genetic isolation in Mota using our techniques, a caveat we mention in our revised manuscript (SI section 5). Nonetheless, values are comparable across Ethiopian clusters (e.g. Fig 3, Fig S11-S12).

18. Page 20 Lines 15-22: “Analogously, minority discriminated groups such as the Negede Woyto (Teclehaimanot, 1984; Legesse, 2013), Shabo (Dira & Hewlett, 2017), Manjo from Kefa Sheka (Freeman & Pankhurst, 2003), and Manja from Dawro (Dea, 2007) exhibit patterns of relatively low genetic similarity to other Ethiopians under the “Ethiopia-internal” analysis that disappear under the “Ethiopia-external” analysis (Fig 2b), in addition to showing higher

degrees of genetic homogeneity (Fig S8a). Given this consistent pattern across discriminated groups, we argue it is most likely that discriminatory practices are directly responsible for having increased the genetic differentiation between discriminated peoples and other Ethiopians.” This conclusion is not well justified. These patterns may also be consistent with the small census sizes for some of these groups, such as the “Shabo”.

We have rewritten this in the revised Discussion, to make more clear the point we are trying to make:

“ We demonstrate how two different types of analyses, which we term “Ethiopia-internal” and “Ethiopia-external”, can disentangle relatively recent from ancient shared ancestry to better understand the origins of different ethnic groups (Fig S4-S5). As examples of this, minority discriminated groups included in this study, such as the Manjo from Kefa Sheka (Freeman & Pankhurst, 2003), the Manja from Dawro (Dea, 2007), the Ari/Wolayta Blacksmiths/Potters/Tanners (Biasutti, 1905; Pankhurst, 1999), the Chabu and the Negede-Woyto (Teclhaimanot, 1984; Legesse, 2013; Dira & Hewlett, 2017), each exhibit signatures of endogamy reflected by relatively high degrees of genetic homogeneity (Fig S3ab). Also consistent with endogamy, these groups each show relatively low genetic similarity to other Ethiopians under F_{st} (Extended Table 9) and the “Ethiopia-internal” analysis (Fig 2a, Fig S8a, Extended Table 3). However, these genetic differences disappear under the “Ethiopia-external” analysis (Fig 2a, Fig S8b, Extended Table 4), suggesting such endogamy/isolation has not persisted for long periods. This in turn suggests that recent practices are rapidly increasing the genetic differentiation (e.g. measured by F_{st}) between discriminated peoples and other Ethiopians.

For example, we infer very similar sources and dates of admixture in independent analyses of distinct clusters that correspond to the occupational groups within the Ari (clusters 22, 24 and 27 in Fig 3, Fig S11) under the “Ethiopia-external” analysis, with overlapping 95% confidence intervals spanning 53-130 generations (Extended Table 5). A parsimonious explanation of these findings, matching our simulations (Fig S5), is that the ancestors of the Ari were a single population when these admixture events occurred. This in turn suggests the ancestors of different Ari occupational workers became isolated from one another only within the past ~146 generations (<4200 years, assuming 28 years per generation). This corresponds to the time period during which

iron working is believed to have first appeared in Ethiopia (Phillipson, 2005) and supports the marginalisation theory of their origins (Lewis, 1962), which is consistent with findings from previous genetic studies (van Dorp 2015, Gopalan 2019).

Analogous to this, the Chabu, who are not linguistically classified by Ethnologue, share admixture events (dated to 300-900 years ago) and ancestry proportions similar to those inferred in the Mezhenger (Fig 3, Fig S11, Extended Table 5). These inferences are consistent with a high degree of intermarrying among the Chabu and Mezhenger, as has been proposed (Gopalan et al 2019; Anbessa & Unseth, 1989), and/or that these two groups split within the last ~900 years. For the Negede-Woyto, the other group in this study for which there is no established linguistic classification in Ethnologue, we infer a relatively high amount of Egyptian-like ancestry (Fig 3, Fig S11), which is consistent with the group's own origin narrative of a migration from Egypt by way of the Abay river (REF?). The ancestry proportions and admixture dates inferred in the Negede-Woyto are similar to those in the Beta Israel and Agaw, whom some scholars have proposed possible genealogical relationships with (Legesse, 2013), and show the highest average similarity to AA Semitic speakers ($p\text{-val} > 0.05$; Fig S9).

A caveat to the interpretation that groups with similar inference under the “Ethiopian-external” analysis share similar recent ancestry is that this analysis will have decreased (or no) power to discriminate between Ethiopian groups that indeed have separate ancestral sources if we have not included relevant non-Ethiopian groups to represent these separate sources. The large number of non-Ethiopian groups included in this sample, particularly those geographically proximal to Ethiopia, diminishes this possibility, but more samples from other sources, in particular DNA from ancient individuals in Ethiopia, may increase precision in identifying ancient ancestral differences between Ethiopians using these techniques. ”

19. Page 13 Lines 5-7: “These observations suggest that the first three tiers of Ethiopian language classifications at www.ethnologue.com are genetically -- in addition to linguistically - - separable on average, and that these genetic differences may not solely be attributable to

recent isolation”. I’m not following how they reached this conclusions. Do they observe statistically significant differences?

Yes, these differences are statistically significant (p -value < 0.001) – we have edited the sentence to report this (see Fig S9, SI section 4).

20. Are all cultural traditions independent of each other? If any are not, did you account for correlation between traditions in your analysis?

Our cultural similarity score between two groups counts the number out of 31 practices for which both groups report participating in that practice or both groups report not participating in that practice. We now provide this cultural reporting information in Extended Table 10. Our aim is to test whether genetic similarity is associated with groups' reporting. It is difficult to select any particular correlation cutoff here, because even two traits differing only by the reporting practice of a single group (but otherwise identical) carry information. Based on this reviewer's comment, we did check how often two traits (i.e. columns of Extended Table 10) are exactly identical, to see if we are indeed replicating information. Three columns are exactly identical with each other, due to only a single group reporting participation in each of these three practices (all other groups are unknown for these three traits). As only one group reports these three traits, these three traits do not contribute to our cultural similarity score.

Of further note, we test each of the six cultural traditions noted in Fig 5 separately, so that dependence does not matter unless determining a p -value cutoff – here we list a 0.05 threshold as providing nominal significance, since an appropriate threshold (i.e. based on the number of tests) is unclear, given groups overlap across different traits.

Minor Comments:

1. Page 17 Lines 5-8: “Therefore, to test for evidence of such recent intermingling between two 6 groups, we assess whether some pairings of individuals, one from each group, have average inferred 7 MRCA segments that are over 2.5 centimorgans (cM) longer than the median length of average inferred MRCA segments across all such pairings of individuals from the two groups (Fig 7).”, They should explain why they chose 2.5 cM.

The aim is to choose a value that is large enough to distinguish shared haplotype segments from background noise. Ideally it should not be too critical which value is chosen, given we aim to assess whether the trend (however defined) occurs more among groups sharing a cultural practice relative to randomly selected groups. Previously we reported how often this trend occurred among the 46 SNNPR groups used in this analysis. However, to better quantify whether there is increased evidence of the trend among groups that report male/female circumcision and sororate/cousin marriages, we now performed 10,000 re-samples of the 46 SNNPR groups and report these new results in Table S11. To specifically address the reviewer's concern, instead of only 2.5cM we tried seven different values (2, 2.5, 3, 3.5, 4, 4.5, 5cM) to assess robustness. As shown in the new Table S11, the proportion of re-samples for which the trend occurs as or more often than that observed in the raw data ranges from 0.00315-0.11788 depending on the length used. In the main text, we now write: “Groups sharing one of these six traits in common showed higher average genetic similarity than that expected based on linguistic affiliation and spatial distance (Fig 5), and we see increased evidence of recent intermixing among groups reporting male/female circumcision and sororate/cousin marriages relative to other SNNPR groups (Fig 5, Table S11, see Methods, SI section 5).”

2. Page 14 Lines 16-17: “The other group with uncertain linguistic classification in our study, the NegedeWoyto, are significantly differentiable ($p\text{-val} < 0.001$).” If all clusters, which basically correspond with ethnic groups, are statistically significant based on fineSTRUCTURE, how do we interpret this finding?

The NegedeWoyto indeed do cluster separately from other groups (Fig S10), which is consistent with our “Ethiopia-internal” results (Fig S8a). An advantage of the latter is that we can compare it to the “Ethiopia-external” results, which cannot be clustered in the same way using fineSTRUCTURE. As we note in the revised main text “However, under the “Ethiopia-external” analysis, ... the Negede-Woyto are not significantly distinguishable from multiple ethnic groups representing all three AA branches (Fig 2b, Fig S8b, Fig S9).”

3. Page 22 Lines 12-17: “On the other hand, the increased amount of ancestry related to West and Central African groups (type (b) clusters) suggest independent recent DNA contributions (<4200 years ago) from individuals carrying ancestry related to the migrations of Bantu speaking peoples from West and Central Africa into East and South Africa, which started around 5,000 KYA but are not believed to have reached present-day Ethiopia during the initial migrations (Vansina, 1995; Ashley, 2010; Clist, 1987)”. Which initial migrations are they referring to?

We have removed this wording in the edited manuscript.

4. Page 22 Lines 17-21: “Clusters predominantly consisting of Nilo-Saharan speakers are either in type (b) clusters or unclear, with each showing a relatively high degree of ancestry related to West/Central Africans and an inferred admixture date more recent than 1600 years ago, perhaps reflecting these groups recently intermixing with other non-Nilo-Saharan speakers in Ethiopia.” Which other non-Nilo-Saharan groups are they referring to?

We have rephrased this: “On the other hand, inferred admixture dates in groups with varying amounts of ancestry related to Bantu and Nilotic speakers are dated to <1600 years ago, with the exception of the NS-speaking Kwegu (~1700 years ago), which may reflect recent intermixing of NS-speakers with other Ethiopians. Such recent intermixing is consistent with mixed ancestry signals we see in some NS groups (e.g. see Berta and Meinit clusters 16, 49-50 in Fig 3, Fig S11). ”

5. Page 19 Lines 11-13: “We provide evidence that these patterns may be explained in part by the recent intermingling of individuals from groups sharing the same cultural practices”. Does intermingling refer to admixture or cultural exchange? Can you define this more clearly?

We have now changed intermingling to intermixing anytime we use it in the revised manuscript.

6. Pages 14-15 Lines 23-2. “Scholars have also proposed possible genealogical relationships with the Beta Israel and/or Agaw (Legesse, 2013); we observe a high genetic similarity between Negede Woyto and each of these groups (Fig S2b) though with some degree of recent isolation among them (Fig S2a).” Is this genetic sharing higher compared to sharing with

other groups?

They are indistinguishable (p -value > 0.05) from each of the Beta Israel and Agaw under the “Ethiopia-external” analysis (Fig S7b), but they also show this with other groups. We have edited this to: “The ancestry proportions and admixture dates inferred in the Negede-Woyto are similar to those in the Beta Israel and Agaw, whom some scholars have proposed possible genealogical relationships with (Legesse, 2013), and show the highest average similarity to AA Semitic speakers (p -val > 0.05 ; Fig S9).”

7. Your website(https://www.well.ox.ac.uk/~gav/work_in_progress/ethiopia/v5/index.html) has the potential to be very beneficial for other researchers. However, the labels on the graphs are too small and there should be some added documentation on how to make comparisons between groups. Is it possible to download any data or can you only look at the graphs?

For the graph labels, we assume the reviewer is referring to the “Mean between-group TVD” at the bottom right? While labels at the bottom are indeed challenging to read, if users hover over any particular bar, the name becomes enlarged and should be easily readable. We have also made a “Help” pull-down menu at top right that addresses how to make comparisons between groups.

This per group TVD information is also available in Extended Tables 3-4, as well as F_st in Extended Table 9.

8. Page 27 Lines 18-19: “In each analysis, for painting (2) we used ten painting samples inferred by CHROMOPAINTER per haploid of each target individual”. Why did you choose 10 painting samples?

Using 10 samples has proven sufficient in previous work. We've now added “following Hellenthal et al. (2014), ...”.

9. Page 11 Lines 4-11: “Strikingly, the SOURCEFIND-inferred proportion of DNA that each Ethiopian cluster matches to the 4,500-year-old Ethiopian Mota also showed a significant (linear regression p -value < 0.0001) decrease with increasing spatial distance between the average location of individuals in that cluster (based on birthplace) and where Mota was discovered (Fig 4cd, Table S7). This is consistent with the preservation of substantial population structure over the past 4.5k years in the region of the Gamo Highlands where Mota was found (Gallego-Llorente et al., 2015, Gopalan et al., 2019).” Can you place this degree of population continuity in context with other examples of population continuity in the literature, such as Posth et al 2018? Is this the longest example of population continuity in Africa?

We do not actually infer complete continuity, given we detect admixture in most Ethiopian groups that is more recent than 4.5kya, and infer in all groups at least some degree of ancestry more recently related to other groups than Mota. Instead we are suggesting some degree of the original population structure remains, rather than total replacement, and so are avoiding using the term “continuity” as may be more appropriate in e.g. Posth et al 2018.

10. Pag 13 Lines 16-18: “Meanwhile AA Cushitic and AA Omotic speakers display a wide range of inferred dates and admixture proportions that fall between those inferred for NS and AA Semitic speakers (Fig 3b, Extended Table 5).” Expand on why this is important.

This observation suggests a high heterogeneity among groups speaking these languages; the revised

manuscript has removed this particular wording.

11. Pages 32-33 Lines 22-2: “However, we instead use our approach of averaging (1 – □□□□□) across individuals because of the considerable reduction in computation time when performing large numbers of permutations.” Do you expect this to change your final results? If so, by how much? This is a rather ad hoc method for analyzing genetic distance between populations. It’s not clear why other more standard approaches weren’t used.

We have now included pairwise F_{ST} among Ethiopian groups (Extended Table 9), which we note are correlated (Pearson's $r = 0.64$, Mantel-test p -value < 0.000001 , as noted in the revised main text) with our “Ethiopia-internal” TVD analysis (Extended Table 3, Fig S8a). However, F_{st} does not seem to as clearly capture some of the subtle differences among groups, e.g. endogamy effects in Ari Potters, suggesting reduced power as has been observed by other researchers comparing the two (Busby et al 2016).

12. Page 2 Line 7-10: “we disentangle the effects of geographic distance, elevation, and social factors upon shaping the genetic structure of Ethiopians today. We provide examples of how social behaviours have directly --and strongly – increased genetic differences among present-day peoples.” This language is too strong (“disentangle” and “directly” and “strongly”) given the many uncertainties discussed above.

We would argue that such factors can have a strong impact, for example both our genetic distance measures (Fig 2a, FigS8a) and F_{st} metrics (Extended Table 9) between the Ari Smiths/Potters and other groups are atypically high. See also the high amounts of homogeneity among minority discriminated groups (Fig S3). Given genetic differences disappear under the “Ethiopia external” analysis (Fig 2a, Fig S8b), with many groups showing similar admixture inference (Fig 3, Fig S11) indicative of sharing recent ancestry, relatively recent changes in social behaviour offer the most parsimonious explanation for how these genetic differences have arisen.

13. Page 17 Line 5: “Therefore, to test for evidence of such recent intermingling between two”. I think this is supposed to be “intermingling”

Thanks for noticing, though based on this reviewer's other previous suggestion, we have replaced the term “intermingling” with “intermixing” now.

14. Page 28 line 19: “For all individuals we recorded their, their parents’, paternal grandfather’s”. “their” is repeated twice

This is correct we think, as the first “their” refers to the actual subject's information.

15. Page 30 Lines 8-9: Specify the exact file name that was downloaded from this ftp server “genetic map build 37 available on the 1000 Genomes Project website (<http://ftp.1000genomes.ebi.ac.uk/vol1/>).”

As we now specify, these are available at <https://github.com/johnbowes/CRAFT-GP/find/master>.

16. Page 6 Line 12: “For a set of pre-defined groups (see below)”. Please specify the exact section that describes these groups

We have removed instances of “below” that do not refer to the same section, now providing more specific information.

17. Page 9 Line 19: Cite a source that supports the use of a 28 year generation time. This is a very reasonable generation time, however there should be a cited source that supports this value.

Thanks for noticing this – we have now provided one: J.N. Fenner 2005, Am J Phys Anthropol 128:415-423

18. Note that the anthropological literature refers to the “Shabo” as “Chabu”. Shabo is considered derogatory.

We have changed this throughout. We note the labels we use here were entirely based on those provided by local interpreters in Ethiopia, as noted in SI section 2, with some changes suggested by Ethiopian academics during our lecture tours.

Reviewer #3 (Remarks to the Author):

The manuscript by Lopez et al analyze the genetic landscape of Ethiopia by genotyping over 1,200 Ethiopian individuals and analyzing the data for population structure and admixture, using mostly chromopainter based techniques. They analyze the genetic data with respect to geography, linguistic data and various socio-cultural aspects. They define the most important factors governing population structure within Ethiopia by Ethiopia internal analyses. They also infer demographic history of Ethiopia in relation to external groups and identify and describe important past migrations into the region. The analyses are robust and very detailed and interesting novel aspects governing genetic structure in Ethiopia were uncovered and discussed. The manuscript is well written and the ideas and conclusions are explained well. However, I still felt there is some room for improvement with regards to some additional explanation, clarification and discussion. I added detailed comments below, which the authors may consider to improve clarity and information in the manuscript.

Thank you for your positive assessment of our work.

Comments:

1. In general the article feels very detailed and long. It is nice with detailed and well-motivated results and discussion, however, perhaps some whole sections can be moved to supplementary notes and just referenced with a line or two from the text.

We have made efforts to streamline the manuscript and have reduced the main text from ~5900 to ~4900 words.

2. At the start of the result section it feels like a lot of methods are explained – which can be shortened and moved to the method section

We have shortened this description considerably, e.g. moving the original Fig 2A to the supplementary material (Fig S4).

3. On page 8 line 11-13 Regarding the way the populations in Figure 3B were sorted and the sentence:

“Broadly, ancestry inference places the Ethiopian clusters on a northeast to southwest cline corresponding to increasing ancestry related to West African groups and decreasing ancestry related to present-day Egyptian groups, consistent with geography (Fig 3).”

This sorting feels very subjective. The authors don’t support the correlation of these

components with geography statistically and this correlation is also not very apparent when just looking where populations are on the map in Figure 3A. If you sort by any component in the clustering analysis you will see a cline and the authors don't motivate why they chose West Africa and Egyptian components.

We have removed this sentence, and instead divide our new (simplified) Fig 3 based on sources of admixture inferred in common.

4. Page 12 lines 12-17. Are the admixture date inferences similarly sensitive to detect older dates compared to recent dates?

All things equal (e.g. sources and proportions of admixture), it is more challenging to detect older admixture events relative to recent ones. Older tests typically have larger confidence intervals to reflect this. Encouragingly, however, our newly added simulations mimicking admixture signals seen in Fig 3 show high accuracy (see SI section 3, Fig S4).

5. Page 14 line 7-14. The authors should outline more clearly how their inferences about the Chabu compare with Gopalan 2019

We have amended these lines to read “The Chabu, a hunter-gatherer group and linguistic isolate, exhibit the strongest overall degree of genetic differentiation from all other ethnic groups, consistent with previous analyses highlighting their genetic isolation (Gopalan et al 2019).”

Later we note: “These inferences are consistent with a high degree of intermarrying among the Chabu and Mezhenger, as has been proposed (Gopalan et al 2019; Anbessa & Unseth, 1989), and/or that these two groups split within the last ~900 years.”

6. Regarding the sentence “The other group with uncertain linguistic classification in our study, the Negede Woyto, are significantly differentiable (p-val < 0.001) from all other ethnic groups under the “Ethiopian internal” analysis, and cluster separately (Fig 2b, Fig S2a, Fig S3).”

In figure 2b this group have very similar cluster assignments to the groups that surrounds it on the clustering analysis plot.

We agree Negede Woyto seem genetically similar to other Ethiopian ethnic groups (the revised Fig S8), despite showing significant differences. We have greatly reduced the number of groups in our revised Fig 2b to highlight e.g. the Negede Woyto, which should help make our points more clear, and we have revised the main text: “The second example concerns the two sampled groups in our study for which Ethnologue ascribes no linguistic classification, the Chabu and Negede-Woyto. Each are significantly differentiable (p-val < 0.001) from all other ethnic groups under the “Ethiopia-internal” analysis (Fig 2b, Fig S8a)... However, under the “Ethiopia-external” analysis ... the Negede-Woyto are not significantly distinguishable from multiple ethnic groups representing all three AA branches (Fig 2b, Fig S8b, Fig S9).”

7. P18 line 5. “In particular we show a strong concordance between genetic differences and geographic distance among individuals (Fig 4), similar to that shown previously among peoples sampled from European (Novembre et al., 2008, Leslie et al., 2015) and worldwide countries (Li et al., 2008).”

Are there perhaps some values to compare whether the correlations are similar magnitudes? I.e. Procrustes factors or mantel tests?

This is an interesting question. While we did consider mantel tests, one complication is that our

“Ethiopian-internal” analysis, which would be most comparable to these studies, shows an exponential rather than linear decay (assumed by Mantel tests) between genetic and geographic distance (Fig 1b). We now provide coefficients describing our rates of decay and/or slope of decay for these analyses in new Tables S4-S5, and envisage direct comparisons like this as an important avenue for future work.

8. P18 line 8. I was surprised to see Mota quoted here as a farmer individual – is this correct?

This is indeed a typo and we have changed this; thanks for pointing it out.

9. RoH and selection scans are not included in the results section but only discussed in the Discussion section. Please introduce results first in result section and then discuss in discussion section.

You are right, we now first mention these results (now Fig S3) in the first paragraph in Results: “Runs-of-homozygosity (Chang et al., 2015) and proportions of genome shared identical-by-descent (IBD) (Browning and Browning, 2011) among individuals vary substantially across Ethiopian ethnic groups (Fig S3ab). Ethiopia’s two largest ethnicities, Amhara and Oromo, have the lowest levels of IBD-sharing (Fig S3a), and we observe a significant ($p\text{-val}<0.001$) decrease of homozygosity versus increasing population census size across ethnic groups in the SNNPR (Fig S3c; census from 2007: The Council of Nationalities, Southern Nations and Peoples Region, 2017).”

10. Page 21-22. Results of the Ethiopian external analysis are discussed extensively in comparison to previously published results – however no mention or reference is made to the Prendergast 2019 aDNA study. This study on ancient HG, pastoralist and farmers from East Africa made various hypotheses about the movements of North African/Levant components into East Africa in relation to the introduction of pastoralism. These findings are important to discuss in the context of the current work. Also the current study only included the Mota aDNA individual and include none of the aDNA data for East Africa published by Skoglund 2017 and Prendergast 2019.

The study from Prendergast et al 2019 makes a valuable contribution to aDNA data from east Africa but was not released when this paper was submitted. However, we have now incorporated samples published by Prendergast et al 2019 in our re-analysis. To aid comparison, we also re-framed our SOURCEFIND analysis to mimick their Figure 3, as we describe in our revised SI section 5. We now also include the aDNA samples from Kenya and Tanzania reported in Prendergast 2019 for our GLOBETROTTER analyses to date admixture (middle of Fig 3). However, we remove these aDNA samples from our SOURCEFIND analyses, for the reasons described below. In addition, we did not include the Skoglund 2017 samples due to low coverage, which precludes using haplotype information. However, we did include the high coverage South African aDNA sample “BallitoBayA” from Schlebusch 2017; this sample was not matched to by the Ethiopians included here.

Relating to this, we have also added the following to our revised Discussion: “To facilitate comparison, our SOURCEFIND analysis included reference groups related to the four primary sources of ancestry for ancient and present-day East African groups reported in Prendergast et al 2019 (see SI section 5). We excluded their aDNA samples as reference groups, because they reported them to be admixed by these four sources. While using different reference groups and techniques complicates direct comparisons, our inferred sources of ancestry broadly agree with theirs. For example, the Agaw (clusters 65, 67) have relatively more Levant-like ancestry (which we match most closely to Egypt), the Ari (clusters 22, 24, 27; called Aari in Prendergast et al 2019),

have relatively more Mota-like ancestry and the Ethiopian Mursi (cluster 2) have relatively more Dinka-like ancestry (Fig 3, Fig S11). Simulations mimicking the admixture inferred here show high accuracy in inferred dates and sources, though illustrate an inherent limitation that older dates of admixture (e.g. those reported in Prendergast 2019) may be masked by more recent admixture (Fig S5). Thus complex intermixing events, such as those exhibited here, can be difficult to dissect fully with these approaches and sample sizes, e.g. distinguishing between multiple pulses or continuous admixture. A potential example are the NS-speaking Berta (clusters 49, 50), in which we infer only a single recent date of admixture but whom have complicated sources of ancestry that suggest multiple events (Fig 3, Fig S11).”

We explain further in our revised SI section 5: “Our choice of reference populations is informed by findings reported by Prendergast 2019 that studied ancestry in East Africans (including Ethiopians), in that we included surrogates related to the four primary sources of ancestry they detected, i.e. populations from West Eurasia (representing what Prendergast 2019 refer to as “EN1” in their Figure 3), the Sudanese Dinka (“EN2” in their Figure 3), the 4.5kya Ethiopian “Mota” (“E.African forager-related” in their Figure 3) and Bantu-speaking groups (“W.African-related” in their Figure 3). We excluded the aDNA samples from Kenya and Tanzania reported in Prendergast 2019 in our SOURCEFIND analysis, because (i) the authors inferred each of those aDNA samples to be admixed descendants of the four primary sources mentioned above and (ii) to enable comparison to the Prendergast findings (e.g. their Figure 3).....

....Though using different surrogates and techniques complicates direct comparisons, our inferred sources of ancestry broadly agree with those reported for present-day Ethiopian groups by Prendergast 2019, in particular results reported in their Figure 3. For example, the Agaw (clusters 65, 67 in Fig 3, Fig S11) have relatively more Levant-like ancestry (which we match most closely to Egypt), the Ari (clusters 22, 24, 27; called Aari in Prendergast 2019) have relatively more Mota-like ancestry and the Ethiopian Mursi (cluster 2) have relatively more Dinka-like ancestry. Also consistent with their results, in general we find NS-speakers to have more Dinka-related ancestry than AA speakers. Furthermore, we infer mixture between Mota-like and Levant-like sources as old as around 4000 years ago among some AA speaking populations, though we note our study of present-day populations may miss some of the older admixture events reported in that study.”

11. In the materials and methods it says the Malawi samples from Skoglund et al were included but they are not shown in the map.

Thank you for pointing this out; we have now corrected this (new Figure 3, top-right).

12. For the PCAs in the supplement please include the variation explained by each PC. Also showing more PC's than just the first two can be very interesting regarding additional population structure

We have done so, providing PCs 1-8 for all including samples (Fig S2A) and African samples only (Fig S2B), also highlighting aDNA samples from Prendergast et al 2019, Schlebusch et al 2017, Lipson et al 2020 in addition to the 4.5kya Ethiopian Mota (Gallego-Llorente et al., 2015).

13. Please explain clearer in the introduction or discussion that the occupation specialization does not apply to all population groups in Ethiopia

We have clarified this in Results where we first mention the Ari/Wolayta divisions (Fig 2a) as follows: “This suggests that occupation is a better proxy for genetic homogeneity than ethnicity in

these groups, consistent with caste-like occupational classes observed in these groups (Freeman & Pankhurst, 2003).”

14. It was not clear whether authors controlled for possible influential co-variates (e.g. language, geography) when correlation with cultural traits were tested.

Yes, we used partial Mantel tests to adjust for these covariates when testing for an association with culture, in the new Table S9. To clarify in the revised text: “We found a significant association between genetic similarity and reporting shared cultural traits among SNNPR groups under the “Ethiopia-internal” analysis (Mantel-test p-value < 0.01; Table S9), which remained after accounting for geographic or elevation distance (partial Mantel-test p-value < 0.02; Table S9) or language group (partial Mantel-test p-value < 0.01; Table S9).”

15. Some of the colors on figure 1 are too close to each other and are difficult to distinguish in print

We have removed the original Fig 1B. We have also made the purple darker for the Chabo in our revised Figure 1A.

16. Figure 3. It is not clear how the number of sources to base the clustering on was chosen. From comparison to the box in figure 1b it seems the top 8 were chosen – but it is not clear why the cutoff was at 8. For instance, it caused the central Afr HGs to be grouped with the “other” fraction – which could have been interesting to have separate

The barplots in Figure 3 make use of separate population labels, rather than clusters, which were tested using SOURCEFIND. As we make clear in the revised main text: “Out of 284 reference populations, SOURCEFIND infers only 11 contributed >5% towards describing ancestry patterns within any of these 67 clusters: the 4,500-year-old Ethiopian Mota and ten present-day groups from Chad, Egypt, Israel, Kenya (2 groups), Somalia, Sudan, Tanzania, Uganda and Yemen (Fig 3, Fig S11, Extended Table 5).” We highlight the contributing groups in the revised Fig 3 and Fig S11, naming them by ethnic label and language and showing their location in the top-left of Fig 3.

17. Figure 4. It would be nice to include information on the gradients of the lines together with the residuals, in addition to p-values.

As it was cluttered to include all of this information on the figures, we have moved this information (in the new manuscript Fig 1B, Fig S6, Fig S12) to new Tables S4, S5 and S10.

18. Figure 6 – Referring back to my comment 14 – it was not clear if confounding effects were controlled for here

Fig 6ab has been removed from the revised manuscript (Table S9 now) and Fig 6cd has been moved to Fig S13. As noted in the revised Table S9 and Table S12, confounding effects (spatial distance, language) were controlled for.

19. Figure 7 – please add information on sample sizes. Again - referring to comment 14

We have added sample sizes to this figure now (now Fig 5). As noted in the revised legend: “The bottom right of each heatmap shows the increase (red) or decrease (blue) in average genetic similarity relative to that expected based on the ethnicities’ language classifications (key in bottom right heatmap), after accounting for the effects of spatial distance, between every pairing of ethnicities who reported practicing (“Y”) the given trait (axis labels colored by language group as in

Fig 1a; group labels given in Table S1).” Though the boxplots don't show this, we now note in the revised Materials and Methods (in the “Genetic similarity versus cultural distance” sub-section): “(These p-values remained after first adjusting for spatial distance as described in this paragraph.)”

Reviewers' Comments:

Reviewer #2:

Remarks to the Author:

1) It is incorrect to state that Tanzania and Cameroon did not have institutions to consult for the approval of sample collection. For reference see "International Compilation of Human Research Standards", 2016 Edition. Compiled By: Office for Human Research Protections U.S. Department of Health and Human Services where they clearly state institutions that were formed in Cameroon and Tanzania prior to the collection of these samples.

a. Cameroon:

i. "The Cameroon Ministry of Public Health created the Cameroon National Ethics Committee (CNEC) in 1987 to review biomedical research conducted in Cameroon."

<https://www.ccghr.ca/resources/harmonization/cameroon/cameroon-research-ethics/>

b. Tanzania:

i. National Institute for Medical Research, Act of Parliament No. 23, of 1979.

1. "The responsibility for controlling, coordinating, conducting and promoting the conduct of health research and dissemination of research results in Tanzania is vested upon NIMR by the Act of Parliament No.23 of 1979 (Appendix III). It follows therefore that in order to conduct health research in Tanzania permission must be sought from NIMR and under no circumstances can any research involving human beings be conducted or initiated in Tanzania without the permission of NIMR", and "Further, Section 12 empowers NIMR to call for information on medical research by requiring every person or body of persons engaged in medical or other allied scientific research within Tanzania to furnish to it such information relating to medical or other allied scientific research as the Institute may specify. The section spells out condition to be complied that, every person or body of persons required to furnish information under Section 12 (1) shall comply with the requirement and any person or body or persons, who refuses or fails to comply with that requirement shall be guilty of an offence." from appendix three of the TANZANIA NATIONAL HEALTH RESEARCH .FORUM GUIDELINES OF ETHICS FOR HEALTH RESEARCH IN TANZANIA Prepared by the National Health Research Ethics Committee Published in Dar es Salaam Second Edition 2009

2) The ethical approval from UCL is still difficult to follow. The authors in their response mentioned that sampling began in 1996, however their ethical approval form from UCL only states approval starting in 1998. Furthermore, the document they provide states approval for 300 samples, yet the manuscript presents 1200 Ethiopians. It is also confusing from the information presented here what populations the permits allowed to be sampled. Furthermore, this document does not mention approval for sampling from Cameroon. It only provides approval for sampling from Ethiopia and Sudan.

3) The consent form is concerning for multiple reasons: 1) no place for the participant to sign, which questions whether verbal or written acknowledgement of consent was given. If verbal consent was given, then there must be audio recordings of that consent. 2) the statement of there being no risks to participants is false. The possible loss of anonymity is a risk factor when contributing your DNA (Erlich et al 2018, PMID: 30309907), 3) It is stated that a participant can withdraw their consent at any time, however there is no provided mechanism or contact information if an individual chooses to do so.

4) Is David Reich's website a secure database to protect against the improper use of these samples compared to the prior mentioned strategy in the Gene Expression Omnibus? The deposition of these samples in an appropriate national public repository is essential. Furthermore, the added restriction of not performing medical research should be added to the data use agreement, as it is stated these samples were collected for population history studies only: "As a purely academic endeavour with no medical or commercial component." Finally, what plans are in place on David Reich's website to ensure investigators adhere to the data use agreement?

5) "The practices listed below are reported in (The Council of Nationalities, Southern Nations and Peoples Region, 2017), with groups' reports regarding them provided in Extended Table 10. No independent verification has been attempted. Explanation of their nature is based on AT's interactions during collection seasons and knowledge of relevant publications and unpublished dissertations and

theses." There needs to be greater details about the collection of this data. How were accounts of personal interactions standardized across multiple groups? If it is not possible to peer review the cultural data collected, then it should not be included in the study. Furthermore, most dissertations are included at some level at a University's library. Even if not published in a peer reviewed journal, they still need to be cited or described at some level.

6) The statement "We provide evidence of how social behaviours have directly -- and strongly -- increased genetic differences among present-day peoples." is too strong. While a correlation may be observed, that does not mean that social behavior is the primary cause of genetic differences among populations.

7) It is unclear if the MCMC chain probabilities presented are sufficiently robust to support the degree of clustering seen in the data. "As a measure of average cluster certainty, we note that for the 70 of these final cluster assignments with ≥ 5 individuals, on average individuals from different clusters were classified into the same cluster in only 0.7% of MCMC samples. In contrast, this average is 32% for individuals assigned to the same final cluster." 0.7% seems like an acceptable number, but is an average of 32% an acceptable value? What is the distribution of these values, are there some clusters that are very low confidence whereas others that are high confidence?

8) The comment regarding differences in sample sizes was not properly answered as it was not stated how having different numbers of samples in your reference groups can impact the calculation of ancestry proportions and migration time. Differences in reference group sizes can be problematic in certain methods, but it is not explained what is expected for the methods used here.

9) The importance of the Mota sample was not made clear in the main text, and will be confusing to a reader that is not informed about African ancient DNA.

10) "However, these genetic differences disappear under the "Ethiopia-external" analysis (Fig 2a, Fig S8b, Extended Table 4), suggesting such endogamy/isolation has not persisted for long periods. This in turn suggests that recent practices are rapidly increasing the genetic differentiation (e.g. measured by F_{st}) between discriminated peoples and other Ethiopians." Alternative explanations need to be considered. Why does the strong genetic drift have to be attributed to practices involving discriminated populations? What other scenarios could be simulated that could also explain this finding? Furthermore, the next paragraph refers to an example where iron-working is determined to be the cause of population differentiation within the last 4200 years. 4200 years is too long of a time frame to suggest that iron working is the only possible explanation to explain these differences. Alternative explanations need to be considered.

11) The simulations were helpful, however the question regarding alternative scenarios explaining the inferences made in the real data were not addressed. Therefore, it is still questionable as to if some of the demographic models presented here based on similar admixture time and proportions are the best fitting model for these populations. The simulations also helped demonstrate that scenarios with > 2 admixture events are difficult to model with this method. However, it is not clear how many of the populations this limitation applies to. It is also not evident how having highly admixed reference populations impacts this methodology, and having a simulation examining differing levels of admixture in the reference populations would be beneficial. The choice of simulated populations is confusing, and it is not clear how the reference populations are consistent with the dataset, given that there is minimal South Asian (Brahui) ancestry in Ethiopia, and the Buganda do not likely represent a good comparison to the admixture dynamics seen in modern Ethiopian populations since the Buganda are relatively genetically homogenous (Gurdasani et al 2019, PMID: 31675503, I think in this paper "Buganda" is called "Baganda").

12) ADMIXTURE would be a helpful comparison when estimating ancestry proportions. I do agree that the interpretation of ADMIXTURE is very prone to over-interpretation, however I would then ask why the authors current methodology is not prone to over-interpretation?

13) The Sengwer are stated as being Bantu speaking on Page 10 Line 247, however the Sengwer's language is a dialect of Marakwet, which is a Nilo-Saharan speaking language (Ethnologue).

Minor Comments:

1. Page 37 Lines 947 - 949: "Between each pairing of 46 sampled SNNPR ethnic groups, we calculated a cultural similarity score as the number of practices, out of 31 reported in the SNNPR book (The

Council of Nationalities, Southern Nations and Peoples Region, 2017) and described in SI Section 5" I think they mean SI Section 6.

2. Shabo, instead of Chabu, still appears in a few figures: Figure 1A, 6B, 12A/B, and 15.

3. Page 13 Lines 335-339: "For the Negede-Woyto, the other group in this study for which there is no established linguistic classification in Ethnologue, we infer a relatively high amount of Egyptian-like ancestry (Fig 3, Fig S11), which is consistent with the group's own origin narrative of a migration from Egypt by way of the Abay river." A citation is required for this origin narrative.

4. The blue and green borders in figure 3 are hard to determine their meaning when reading the figure and caption. I suggest adding your description of their meaning that is included in the response to Reviewer 1 Comment Major Comment 1: "In particular we enclose the reference populations representing one of the inferred admixing sources with a thick blue line. In Ethiopian groups with >2 inferred sources, we also enclose the reference populations representing the second source with a thick green line"

5. Intermixing is better than intermingling, however it is confusing if this term intends to imply different evolutionary processes than admixture or gene flow.

6. Why is there only an estimate of admixture using the Ethiopia-Internal (red plus sign) for Population 4 in Figure S5D. Is it that the Ethiopia-Internal analysis cannot be used for estimating the dates of the other ones or is it the exact same value?

Reviewer #3:

Remarks to the Author:

I read through the reviewers reply and manuscript with changes of Lopez et al. I am happy with the changes overall and feel the authors addressed all my comments satisfactorily.

Comments to Editor:

We have tracked changes to our revisions throughout the main text and Supplementary Information. For clarity, we have not tracked minor changes attributable to our re-analysis for this submission, such as re-wordings or corrected typos or numbers (e.g. sample sizes) that have changed very slightly in the text and Supplementary Tables. We note that all main conclusions remain the same as in the previous submission.

Comments to Reviewer:

We thank the reviewer for these comments. We address and reply to all individual comments below, with the reviewer's comments in bold and our responses underneath.

Reviewer 2 comments:

1) It is incorrect to state that Tanzania and Cameroon did not have institutions to consult for the approval of sample collection. For reference see “International Compilation of Human Research Standards”, 2016 Edition. Compiled By: Office for Human Research Protections U.S. Department of Health and Human Services where they clearly state institutions that were formed in Cameroon and Tanzania prior to the collection of these samples.

a. Cameroon:

i. “The Cameroon Ministry of Public Health created the Cameroon National Ethics Committee (CNEC) in 1987 to review biomedical research conducted in Cameroon.”
<https://www.ccghr.ca/resources/harmonization/cameroon/cameroon-research-ethics/>

b. Tanzania:

i. National Institute for Medical Research, Act of Parliament No. 23, of 1979.

1. “The responsibility for controlling, coordinating, conducting and promoting the conduct of health research and dissemination of research results in Tanzania is vested upon NIMR by the Act of Parliament No.23 of 1979 (Appendix III). It follows therefore that in order to conduct health research in Tanzania permission must be sought from NIMR and under no circumstances can any research involving human beings be conducted or initiated in Tanzania without the permission of NIMR”, and “Further, Section 12 empowers NIMR to call for information on medical research by requiring every person or body of persons engaged in medical or other allied scientific research within Tanzania to furnish to it such information relating to medical or other allied scientific research as the Institute may specify. The section spells out condition to be complied that, every person or body of persons required to furnish information under Section 12 (1) shall comply with the requirement and any person or body or persons, who refuses or fails to comply with that requirement shall be guilty of an offence.” from appendix three of the TANZANIA NATIONAL HEALTH RESEARCH .FORUM GUIDELINES OF ETHICS FOR HEALTH RESEARCH IN TANZANIA Prepared by the National Health Research Ethics Committee Published in Dar es Salaam Second Edition 2009

Regarding Cameroon, samples were collected in accordance with Cameroon law under Cameroon Research Permits 0188/MINREST/B00/D00/D10/D12, 317/MINREST/B00/D00/D10 and University Yaounde I procedures. We have updated the “Samples” section of the Materials and Methods to make our consent and ethical procedures more clear (see our reply to comment 2 below).

Regarding Tanzania, we note the NIMR that the reviewer refers to has no remit outside of issues involving “medical research” (see <https://www.nimr.or.tz/institute-profile/>), which is not the focus

of the data collected in this study. With respect to APPENDIX THREE of the TANZANIA NATIONAL HEALTH RESEARCH FORUM GUIDELINES OF ETHICS FOR HEALTH RESEARCH IN TANZANIA, the reference to Clause 1 makes clear in every sub clause that only medical research is covered. At any rate we have now re-analysed all of our data after removing these unpublished Tanzanian samples, noting that doing so does not alter our conclusions.

2) The ethical approval from UCL is still difficult to follow. The authors in their response mentioned that sampling began in 1996, however their ethical approval form from UCL only states approval starting in 1998. Furthermore, the document they provide states approval for 300 samples, yet the manuscript presents 1200 Ethiopians. It is also confusing from the information presented here what populations the permits allowed to be sampled. Furthermore, this document does not mention approval for sampling from Cameroon. It only provides approval for sampling from Ethiopia and Sudan.

To clarify, the ethical approval form seen by the reviewer refers to approval for our study on Ethiopian genetics in our submitted manuscript, which did not involve any new sampling. This approved study included comparing genetic patterns in Ethiopians (and Sudanese samples not included in this submission) to those that had not yet been genotyped from other Sub-Saharan African populations that were part of the same previously approved collections. This approval form refers to two other approvals given to research that was conducted prior to our study:

(1) The 300 samples that the reviewer refers to were approved as part of a previous study on Ethiopian genetics that was separate from our current submitted manuscript. Results from this separate study were published in doi: [10.1016/j.ajhg.2015.04.019](https://doi.org/10.1016/j.ajhg.2015.04.019). The UCL ethics committee referred to the ethical approval given for this previous study (in June 2010) as part of their reasoning for approving the project in our current submitted manuscript.

(2) The “approval starting in 1998” the reviewer mentions refers to previous approval by the Joint UCL/UCLH Committees on the Ethics of Human Research: Committee A and Alpha, REC reference number 99/0196 (Chief Investigator MGT). This approval covers the collection and analyses of samples from numerous locations in Sub-Saharan Africa, Europe and West Asia occurring subsequent to 1998. It also covers the analysis of samples from collections that started in 1996 that were collected by researchers at The Hebrew University, Israel, University of Leeds, University College London and School of Oriental and African Studies University of London.

We have clarified this ethical approval information in the “Samples” section of our revised Materials and Methods:

“Local permissions were obtained in all cases where applicable local ethical approval and regulations existed, e.g. Cameroon, Ministry of Higher Education and Scientific Research, Permits 0188/MINREST/B00/D00/D10/ D12 and 317/MINREST/B00/D00/D10 and University of Yaounde I; Ethiopia, Ethiopian Science and Technology Commission. Sample collection/usage for all unpublished data included in this study were approved by the UK ethics committee London Bentham REC (formally the Joint UCL/UCLH Committees on the Ethics of Human Research: Committee A and Alpha, REC reference number 99/0196, Chief Investigator MGT). The analyses

reported here were approved by UCL REC (Project ID: 5188/001).”

3) The consent form is concerning for multiple reasons: 1) no place for the participant to sign, which questions whether verbal or written acknowledgement of consent was given. If verbal consent was given, then there must be audio recordings of that consent. 2) the statement of there being no risks to participants is false. The possible loss of anonymity is a risk factor when contributing your DNA (Erlich et al 2018, PMID: 30309907), 3) It is stated that a participant can withdraw their consent at any time, however there is no provided mechanism or contact information if an individual chooses to do so.

- a) Collections were in accordance with best practice at the time.
- b) We agree with the potential for loss of anonymity that others have highlighted. Therefore, analogous to the hundreds of similar studies that have analysed and made available genomewide autosomal data, we are requiring researchers who access these data to provide a signed letter stating that they will make no attempt to identify study participants or use the data commercially (see response to 4 below), thus adhering to participants' informed consent.
- c) There was no ethical requirement for signed consent or a recording. Either would have increased the risk that an individual could be linked to the DNA, which is forbidden and potentially impossible.
- d) While the collection was being made consent could have been withdrawn at any time. Once the swab was taken and the collection procedure completed, no participation by the donor was possible since there was no link between the sample and the individual. The individual could withdraw at any time during their participation. This protocol was approved by the UCL ethics committee (REC reference number 99/0196, Chief Investigator MGT).
- e) In addition to the approval given by UCL ethics for the original sample collections, the work in our submitted paper was approved by the UCL ethics committee (UCL REC 5188/001). We also note that genetic data from individuals sampled as part of these collections, using these or analogous consent forms, have been published in 50 papers from 1997-2020 including in Nature and other Nature journals.

In summary, we believe the ethical and consent procedures underlying this study reflect best practice. Samples were collected legally, with ethical approval granted prior to the analyses of all data included in this project. We note that the editor has informed us that our information has been found to satisfy the journal requirements for informed consent and ethical approval.

4) Is David Reich’s website a secure database to protect against the improper use of these samples compared to the prior mentioned strategy in the Gene Expression Omnibus? The deposition of these samples in an appropriate national public repository is essential. Furthermore, the added restriction of not performing medical research should be added to the data use agreement, as it is stated these samples were collected for population history studies only: “As a purely academic endeavour with no medical or commercial component.” Finally, what plans are in place on David Reich’s website to ensure investigators adhere to the data use agreement?

We agree data release is an important issue. As is the case with data published from many other international studies made available on David Reich's website, maintained by Harvard University, we will give researchers the following guidance (in italics) prior to downloading the data:

Since our research permits for the collection of the samples analyzed in this dataset only cover studies of population prehistory, we have to stipulate some conditions before granting access to the data, as follows:

To access this dataset, you need to send David Reich a PDF of a signed letter containing the following language:

- (a) I will not distribute the samples marked "signed letter" outside my collaboration,
- (b) I will not post data from the samples marked "signed letter" publicly
- (c) I will make no attempt to connect the genetic data for the samples marked "signed letter" to personal identifiers
- (d) I will use the data from the samples marked "signed letter" only for studies of population history
- (e) I will not use the data for samples marked "signed letter" for commercial purposes

This is sufficient to meeting the provided consents, and we believe it meets best practice and is desirable in terms of publicising these data for the research community and maximising its use. As we have mentioned, data from well over 50 international studies have been made available on this website in the manner we are doing. We have updated the "Acknowledgements" to note that:

"Genotype data, birthplace information and self-reported group label, first language, second language and religious affiliation for each individual are available for non-commercial use at <https://reich.hms.harvard.edu/downloadable-genotypes-present-day-and-ancient-dna-data-compiled-published-papers>."

5) "The practices listed below are reported in (The Council of Nationalities, Southern Nations and Peoples Region, 2017), with groups' reports regarding them provided in Extended Table 10. No independent verification has been attempted. Explanation of their nature is based on AT's interactions during collection seasons and knowledge of relevant publications and unpublished dissertations and theses." There needs to be greater details about the collection of this data. How were accounts of personal interactions standardized across multiple groups? If it is not possible to peer review the cultural data collected, then it should not be included in the study. Furthermore, most dissertations are included at some level at a University's library. Even if not published in a peer reviewed journal, they still need to be cited or described at some level.

We assume the reviewer is referring to the SNNPR book. To clarify, the contents of the book are the raw data. We have included references to both the Amharic version and the English translation of this book:

- a) Council of Nationalities (CoN), Southern Nations, Nationalities and Peoples' Regional (SNNPR) April 2017. *A Profile of the Nations, Nationalities and Peoples of the Southern Region*, Second Edition (December 2004EC), Translated from Amharic and sub-edited by Ayele Tarekegn and Neil Bradman, Published by the Council of Nationalities with financial support from the German Co-operation (GIZ), Addis Ababa: Berhanena Selam Printing Enterprise.
- b) Council of Nationalities (CoN), Southern Nations, Nationalities and Peoples' Regional (SNNPR) *Miazia* (April) 2008EC (in Amharic). *Yedebub biheroach, biheresebochna Hizboch Profile* (title translated as *A Profile of the Nations, Nationalities and Peoples of the Southern Region*), Third Edition, Published by the CoN, SNNPR, Addis Ababa: Berhanena Selam Printing Enterprise.

To make this clearer, we have removed from the revised Supplementary Information (SI section 6): “No independent verification has been attempted. Explanation of their nature is based on AT’s interactions during collection seasons and knowledge of relevant publications and unpublished dissertations and theses.”

Also, to better highlight the utility of this resource, both in this paper and in future work, in the revised Discussion we highlight an example where oral traditions reported in this book are associated with genetic patterns we infer:

“Future work can compare these and other published genetic results (e.g. Scheinfeldt et al 2019, Gopalan et al 2019, Prendergast et al 2019) to oral histories recorded for various ethnic groups. For example, some Mezhenger report that their ancestors originally migrated from Sudan to the present-day Gambella Regional State where Anuak lived, after which they migrated with the AA Omotic-speaking Sheko for a period before settling in their present-day homeland (The Council of Nationalities, Southern Nations and Peoples Region, 2017). Consistent with this, in the “Ethiopia-external” analysis the Mezhenger have high inferred ancestry matching to the Sudanese Dinka (Fig 3, Fig S12, Extended Table 5), and in the “Ethiopia-internal” analysis they have an inferred admixture event ~300-600 years ago among three sources that are best represented by clusters containing the Anuak, Sheko and other NS-speaking groups near the Mezhenger (Extended Table 6).”

6) The statement “We provide evidence of how social behaviours have directly -- and strongly -- increased genetic differences among present-day peoples.” is too strong. While a correlation may be observed, that does not mean that social behavior is the primary cause of genetic differences among populations.

We have adjusted this text in the Abstract: “We provide evidence of associations between social behaviours and increased genetic differences among present-day peoples.”

7) It is unclear if the MCMC chain probabilities presented are sufficiently robust to support the degree of clustering seen in the data. “As a measure of average cluster certainty, we note that for the 70 of these final cluster assignments with ≥ 5 individuals, on average individuals from different clusters were classified into the same cluster in only 0.7% of MCMC samples. In contrast, this average is 32% for individuals assigned to the same final cluster.” 0.7% seems like an acceptable number, but is an average of 32% an acceptable value? What is the distribution of these values, are there some clusters that are very low confidence whereas others that are high confidence?

Based on this suggestion, we now report our certainty score per cluster in a revised Extended Table 2 (“Finestructure certainty”). We note there was a slight inconsistency in our calculations for our cluster certainty score in the previous submission, in that we were checking for consistency across fineSTRUCTURE's inference of 184 clusters rather than the 78 final clusters we used after merging some of these 184. We have adjusted this, which resulted in the 0.7% and 32% figures quoted here improving to 0.7% and 44.7%, respectively. We explain this score, and our motivation for using clusters, in our revised text in the “Classifying Ethiopians into genetically homogeneous clusters” section of Materials and Methods:

“We followed Leslie et al 2015 to generate a measure of cluster certainty using the last 100

fineSTRUCTURE MCMC samples. In particular for each of these 100 MCMC samples, we assigned a certainty score for each individual i being assigned to each final cluster j (out of 78) as the percentage of individuals assigned to the same cluster as individual i in that MCMC sample that are found in final cluster j . (For each individual i , note these percentages sum to 100% across the 78 final clusters.) For each combination of individual and final cluster, we averaged these certainty scores across all 100 MCMC samples. For each of our 78 final clusters, in Extended Table 2 we report the average certainty score of being assigned to that cluster across all individuals assigned to that cluster. This average certainty score had a mean of 44.7% across all clusters (range: 5.6-88.8%). For comparison, the average certainty score of being assigned to a cluster other than the final classification we used had a mean of 0.7% across all clusters (range: 0.1-1.2%). We note that clusters do not necessarily correspond to distinct groups that split from one another in the past, but instead provide a convenient means to increase power and clarity of ancestry inference by (i) merging people with similar genetic variation patterns, and (ii) separating individuals of the same self-identified label that have different genetic variation patterns.”

8) The comment regarding differences in sample sizes was not properly answered as it was not stated how having different numbers of samples in your reference groups can impact the calculation of ancestry proportions and migration time. Differences in reference group sizes can be problematic in certain methods, but it is not explained what is expected for the methods used here.

To address this, we have added text in SI section 5 noting the potential implications of varying reference sample size on inferred ancestry proportions. In brief, theoretically groups with fewer samples may be favored due to a potential inability to measure drift accurately in such groups. However, results across Ethiopians are still comparable (e.g. Fig 3, Fig S12), and inferred proportions/sources in the simulations are very accurate (Fig S5c) despite using 271 reference populations with sample sizes ranging from 1 to 100. This mimicks previous observations (e.g. Hellenthal et al 2014). The full text we are referring to (SI section 5):

“A second caveat of our SOURCEFIND analysis is that five of the 12 present-day groups contributing >5% (Chad_Bulala, Kenya_Elmolo, Kenya_Sengwer, Kenya_Rendille, Tanzania_Iraqw) had only two samples. When painting an individuals' genome using CHROMOPAINTER, an individual cannot match to itself. Thus each of these four populations matched to only one sample from their own population via CHROMOPAINTER, which may mitigate signals of isolation (e.g. due to endogamy) in that population, relative to groups that can match to a greater number of individuals from their own labeled group. By mitigating signals of endogamy effects, such reference populations can potentially be favored as an ancestral source in SOURCEFIND analysis, which aims to find the reference populations with painting patterns that

most closely match those of the target (in this case Ethiopian) cluster. This may also explain why Mota is favored, as it has only a single sample and hence no means of measuring endogamy under this approach. Nonetheless, comparisons among Ethiopian clusters are still meaningful when conditioning on this set of references, as each cluster was analysed in the same way. In general, we note that surrogate populations with high degrees of isolation (e.g. due to endogamy) may be less likely to be selected as representative of an ancestral source, which is one way SOURCEFIND likely differs from e.g. a f3 outgroup test (Patterson et al 2012). But arguably such surrogates should be downweighted, as – due to recent isolation – the genetic make-up of such surrogates likely no longer well-reflects the ancestral source population.”

Regarding the impact of sample size on inferred admixture dates (“migration time”), we are not aware of any biases (theoretical or observed in practice) from having differential sample sizes in the reference groups when using GLOBETROTTER. Supporting this, our simulated populations here (SI section 3) inferred very accurate dates via GLOBETROTTER (Fig S5d) when using 116 reference populations with sample sizes ranging from 1 to 100. In general inferred dates can be less precise with fewer target (i.e. admixed) individuals, though this depends on which groups mixed and when. For example, single individuals can have their dates inferred very accurately in some cases (Chacon-Duque et al 2018). When re-analysing for this revision, to be conservative we excluded clusters with ≤ 5 individuals from the “Ethiopia-internal” analysis, as this analysis is trying to pick up more subtle admixture relative to the “Ethiopia-external” analysis and may struggle with few individuals. We also have added the following text to SI section 5 that describes how, to be conservative, we do not report GLOBETROTTER results where signals in the probability curves are unclear:

“We also visually inspected the probability curves (e.g. Fig S15) to assess whether the conclusions (i)-(iv) that GLOBETROTTER reports fit the data. Based on this visual inspection, and using the parameters highlighted below from the GLOBETROTTER *main.txt output files, we made some slight alterations to GLOBETROTTER's reported conclusions. In particular, to be conservative we do not report GLOBETROTTER results for clusters where “r2.oneevent”, which assess the overall evidence of admixture (on a 0-1 scale), was < 0.34 , as such clusters had noisy probability curves. Also, for the “Ethiopia-internal” analysis, which is picking up more subtle admixture between genetically similar groups (i.e. between Ethiopian groups), we did not analyse clusters with ≤ 5 individuals, and we do not report results for two clusters (Eth_al, Eth_cr) with “r2.oneevent” < 0.5 that had noisy probability curves. In addition to these omissions, we slightly altered GLOBETROTTER's default threshold for concluding “one-date, multiway” over “one-date” from “fit.1event < 0.975 ” to “fit.1event < 0.98 ”, which changed the conclusion from “one-date” to “one-date, multiway” for four clusters (Eth_ab, Eth_ap, Eth_ar, Eth_bh) in the “Ethiopia-external” analysis and one cluster (Eth_ax) in the “Ethiopia-internal” analysis (Extended Table 5-6). Finally, we visually inspected clusters for which “maxScores.2events” $\in (0.3, 0.35]$, which is indicative of multiple-dates of admixture but does not meet GLOBETROTTER's default criterion of “maxScores.2events” > 0.35 for concluding “multiple-dates”. In some of these cases, two admixture dates appeared to fit the data notably better than one date; i.e. the red line in the GLOBETROTTER *pdf file output was a better fit to many probability curves relative to the green line. Thus we changed the conclusion from “one-date” of admixture to “multiple-dates” of admixture for three clusters (Eth_ag, Eth_ak, Eth_as) under the “Ethiopia-external” analysis and for three clusters (Eth_ag, Eth_ak, Eth_bi) under the “Ethiopia-internal” analysis (Extended Table 5-6). We note that other clusters may have multiple dates of admixture that we miss here, and that more data from Ethiopians will help to clarify these admixture signals in the future.”

In addition to these changes, we also tested the robustness of the genetic similarity results based on chromosome painting to varying sample sizes (e.g. those results reported in Fig S8a and Extended Table 3) for the “Ethiopia-internal” analysis, as described in our revised Materials and Methods:

“As individuals are not allowed to match to themselves under the CHROMOPAINTER model, one potential issue with our paintings of Ethiopians under the “Ethiopia-internal” analysis is that each Ethiopian is allowed to match to one less individual in the cluster to which it is assigned relative to Ethiopians outside that cluster. For example, if cluster *A* contains ten Ethiopians, each of those Ethiopians are allowed to match to nine people from cluster *A* under the “Ethiopia-internal” analysis, while Ethiopians outside of cluster *A* are matched to all ten. This may create a slight discrepancy in the f_k^i values among Ethiopians for the 78 elements of *k* representing the Ethiopian clusters, which in turn may affect differences in TVD among Ethiopian group labels. To test this, we repeated the above using an alternative “Ethiopia-internal” painting where each Ethiopian is matched to all other Ethiopians from their cluster and $n_k - 1$ Ethiopians from each other Ethiopian cluster *k* after randomly removing one individual, while matching to all individuals from every non-Ethiopian group as before. This gives a $K=346$ length vector of f_k^i values for each Ethiopian *i* as before, but where each Ethiopian now has been painted against the same numbers of individuals from the *K* groups. We found that results change very little, e.g. with the TVD values among all pairwise combinations of Ethiopian groups (Fig S8A, Extended Table 3) having correlation $r > 0.999$. This likely reflects how, for the given sample sizes in the *k* clusters, removing one individual from a cluster *k* results in people matching slightly more to the remaining $n_k - 1$ individuals in that cluster, so that the total matching to *k* remains relatively unchanged. For comparison, in Extended Table 3 we provide columns at the far right end showing which groups were the closest match under this alternative “Ethiopia-internal” analysis; we note there are few changes relative to the original “Ethiopia-internal” analysis.”

9) The importance of the Mota sample was not made clear in the main text, and will be confusing to a reader that is not informed about African ancient DNA.

We have edited the main text (“Introduction”) to emphasize its importance as the only presently available aDNA sample from Ethiopia:

“We compared SNP patterns in each present-day Ethiopian to those in all other present-day Ethiopians and to the 4,500 year-old Ethiopian sample “Mota”, a forager from southern Ethiopia that represents the only presently available ancient DNA sample from the country (Gallego-Llorente et al., 2015).”

10) “However, these genetic differences disappear under the “Ethiopia-external” analysis (Fig 2a, Fig S8b, Extended Table 4), suggesting such endogamy/isolation has not persisted for long periods. This in turn suggests that recent practices are rapidly increasing the genetic differentiation (e.g. measured by *Fst*) between discriminated peoples and other Ethiopians.” Alternative explanations need to be considered. Why does the strong genetic drift have to be attributed to practices involving discriminated populations? What other scenarios could be simulated that could also explain this finding? Furthermore, the next paragraph refers to an example where iron-working is determined to be the cause of population differentiation within the last 4200 years. 4200 years is too long of a time frame to suggest that iron working is the only possible explanation to explain these differences. Alternative explanations need to be considered.

We have edited this paragraph the reviewer references in order to make this clearer and to de-emphasise a causal association:

“...minority discriminated groups included in this study, such as the Manjo from Kefa Sheka (Freeman & Pankhurst, 2003), the Manja from Dawro (Dea, 2007), the Ari/Wolayta Blacksmiths/Potters/Tanners (Biasutti, 1905; Pankhurst, 1999), the Chabu and the Negede-Woyto (Teclehaimanot, 1984; Legesse, 2013; Dira & Hewlett, 2017), each show relatively low genetic similarity to other Ethiopians using F_{st} (Extended Table 9) and under the “Ethiopia-internal” analysis (Fig 2a, Fig S8a, Extended Table 3). However, these genetic differences disappear under the “Ethiopia-external” analysis (Fig 2a, Fig S8b, Extended Table 4), suggesting that the high levels of genetic differentiation between discriminated peoples and other Ethiopians (e.g. measured by F_{st}) have arisen through their relatively recent isolation from other groups. Consistent with this isolation, these groups also exhibit signatures of recent endogamy as reflected by higher degrees of genetic homogeneity (Fig S3ab) with each of these groups separating into distinct clusters in ADMIXTURE analysis (Fig S11, Lawson et al 2018).”

The “next paragraph” sentence which the reviewer refers to is “This corresponds to the time period during which iron working is thought to have first appeared in Ethiopia (Phillipson, 2005) and supports the marginalisation theory of their origins (Lewis, 1962), which is consistent with findings from previous genetic studies (van Dorp 2015, Gopalan 2019).” We do not believe this sentence suggests iron working is the only possible explanation of the patterns we observe, only that iron working appeared around this time, so we prefer this phrasing.

11) The simulations were helpful, however the question regarding alternative scenarios explaining the inferences made in the real data were not addressed. Therefore, it is still questionable as to if some of the demographic models presented here based on similar admixture time and proportions are the best fitting model for these populations. The simulations also helped demonstrate that scenarios with > 2 admixture events are difficult to model with this method. However, it is not clear how many of the populations this limitation applies to. It is also not evident how having highly admixed reference populations impacts this methodology, and having a simulation examining differing levels of admixture in the reference populations would be beneficial. The choice of simulated populations is confusing, and it is not clear how the reference populations are consistent with the dataset, given that there is minimal South Asian (Brahui) ancestry in Ethiopia, and the Buganda do not likely represent a good comparison to the admixture dynamics seen in modern Ethiopian populations since the Buganda are relatively genetically homogenous (Gurdasani et al 2019, PMID: 31675503, I think in this paper “Buganda” is called “Baganda”).

The Buganda are indeed the Baganda; we thank the reviewer for pointing this out and have now used consistent naming (Baganda) throughout. We agree there are many possible demographic histories that could explain the data. Our simulations aim to explain patterns observed in the data while incorporating several realistic complicating factors. In particular we included bottleneck effects, multiple waves of intermixing, recent intermixing among “Ethiopian-like” groups, unequal reference group sample sizes, and unsampled admixing source populations. To further mimic our real data we chose to mix an East African source (Baganda of Uganda) with a West Eurasian source (comprised of four populations), a signal comparable to that we see in many of our Ethiopian samples. We now note this motivation in SI section 3:

“This admixture scenario is meant to reflect mixture between an East African source (B) and a West Eurasian source (A), as is seen in many of our Ethiopian populations.”

While we agree there may be better source populations to approximate Ethiopian ancestry, we are

limited by available sample sizes. E.g. we choose the Baganda because we had 96 sampled individuals that enabled generating simulated individuals whose data were less correlated (i.e. their genome-wide DNA patterns could be derived as mosaics of different Baganda “ancestors”). Our motivation for choosing our West Eurasian sources ({BedouinA of Israel, Brahui of Pakistan, Egyptian_Comas, Palestinian}) was similar, in that we aimed to compile a relatively large sample size (86 total individuals) using a relatively small number of populations (four).

On the reviewer's point regarding “differing levels of admixture,” our simulations here show that GLOBETROTTER/SOURCEFIND select the best genetic proxies (out of 271 available reference populations) for the admixing sources (Fig S5c). Four of five populations we used to simulate, and six of nine reference populations inferred to contribute >1% to these simulated populations, were found by Hellenthal et al 2014 to have ~2-14% admixture from Africa or East Asia, while the other four were not tested in that paper. Despite our reference and source populations having these varying levels of admixture, the inference in these simulations were still very accurate (Fig S5), as has been seen previously in other simulations (Hellenthal et al 2014).

12) ADMIXTURE would be a helpful comparison when estimating ancestry proportions. I do agree that the interpretation of ADMIXTURE is very prone to over-interpretation, however I would then ask why the authors current methodology is not prone to over-interpretation?

In this revision, we have included results from a new ADMIXTURE analysis applied to the Ethiopian individuals (Fig S11), which we now mention throughout the text to support findings regarding genetic structure and clustering (see tracked changes).

We agree that inferring ancestry proportions using ADMIXTURE, e.g. through supervised or unsupervised analyses that make use of all Ethiopian and non-Ethiopian data considered here, could be an important area of future work. However, in general we believe ancestry proportions are prone to over-interpretation, regardless of the approach, which is why we have provided important caveats on this in Section SI 5 (see our reply to comment 8). Nonetheless, comparing inferred proportions across groups can be informative, in terms of the relative amounts of ancestry related to each reference population. In addition, the consistency across our 68 Ethiopian clusters in Figure 3 and Figure S12 is striking, e.g. in that only 13 of these 275 reference populations contribute >5% to any of these 68 clusters, even after adding samples from more geographically near populations as suggested previously by the reviewers.

13) The Sengwer are stated as being Bantu speaking on Page 10 Line 247, however the Sengwer’s language is a dialect of Marakwet, which is a Nilo-Saharan speaking language (Ethnologue).

We thank the reviewer for finding this mistake, which we have now corrected (throughout the text as necessary and in Fig 3, Fig S12). We note that Markweeta are referred to as Nilotic in Glottolog, so we use that here.

Minor Comments:

1. Page 37 Lines 947 - 949: “Between each pairing of 46 sampled SNNPR ethnic groups, we calculated a cultural similarity score as the number of practices, out of 31 reported in the SNNPR book (The Council of Nationalities, Southern Nations and Peoples Region, 2017) and described in SI Section 5” I think they mean SI Section 6.

We thank the reviewer for finding this mistake, which we have now corrected.

2. Shabo, instead of Chabu, still appears in a few figures: Figure 1A, 6B, 12A/B, and 15.

We thank the reviewer for noting this mistake, which we have now corrected. We note that this was an issue in Figures 1A, S1, S7AB and S10 in the revised version of the manuscript we are submitting.

3. Page 13 Lines 335-339: “For the Negede-Woyto, the other group in this study for which there is no established linguistic classification in Ethnologue, we infer a relatively high amount of Egyptian-like ancestry (Fig 3, Fig S11), which is consistent with the group’s own origin narrative of a migration from Egypt by way of the Abay river.” A citation is required for this origin narrative.

We thank the reviewer for pointing this out, and we have added the citation (Teclehaimanot 1984).

4. The blue and green borders in figure 3 are hard to determine their meaning when reading the figure and caption. I suggest adding your description of their meaning that is included in the response to Reviewer 1 Comment Major Comment 1: “In particular we enclose the reference populations representing one of the inferred admixing sources with a thick blue line. In Ethiopian groups with >2 inferred sources, we also enclose the reference populations representing the second source with a thick green line”

We have added this information to the Fig 3 legend.

5. Intermixing is better than intermingling, however it is confusing if this term intends to imply different evolutionary processes than admixture or gene flow.

We have confirmed that every instance of “intermixing” in the text refers to admixture.

6. Why is there only an estimate of admixture using the Ethiopia-Internal (red plus sign) for Population 4 in Figure S5D. Is it that the Ethiopia-Internal analysis cannot be used for estimating the dates of the other ones or is it the exact same value?

We thank the reviewer for pointing this omission out. We have added the “Ethiopia-internal” date inference for sims 2 (“40% BN”) and 3 (“20% no Dem”), which closely match the truth, to Fig S5D. For sim 1 (“40% Exp”), GLOBETROTTER failed to detect admixture, presumably due to masking from its ancestry patterns being notably similar to those in simulation 4 (“26% 3-date”). We have updated the text in SI section 3 (and Fig S5D legend) to describe these additional results:

“Under the “Ethiopia-internal” analogue, inferred dates for simulations (2) and (3) closely match the truth, while GLOBETROTTER failed to detect admixture in simulation (1), presumably due to masking since its ancestry patterns are similar to those in simulation (4).”

Reviewers' Comments:

Reviewer #3:

Remarks to the Author:

(I was asked by the journal to assess author responses to another reviewer that were not able to look at responses again)

I went through the Hellenthal rebuttal and think they generally answered the reviewers' comments well.

I however have two additional comments - one minor and one I feel quite strong about:

Regarding point 13: The reviewer point out that Sengwer is a Nilo-Saharan language rather than a Bantu language. The authors replied that they used Nilotic as the linguistic group rather than Nilo-Saharan. My comment: Since Nilotic is a sub-group of Nilo-Saharan the authors should still indicate that major linguistic group. They can use Nilotic (Nilo-Saharan) for example.

Regarding point 4: I strongly agree with the original reviewer that the deposition of the data in an appropriate national public repository is essential. The argument by the authors that David Reich's lab database is used by many studies does not really answer to the reviewers' concern. Keeping data on personal lab databases is not justified while public repositories (e.g. nih-dbGaP, ebi-EGA, ebi-ArrayExpress) are available at no charge. There is no guarantee that the Reich database will remain in existence for the foreseeable future. It might be shut down in 10 or 20 years and then access to data is no longer possible - as we have seen with many lab webpages and databases. For the long time preservation of data and information it is crucial that public repository deposition is a strong requirement for publication. Furthermore, since public repositories have the facility of establishing Data Access Committees that can ensure correct usage data - there is no reason not to make use of these resources. If the authors really want to have the data in the Reich database they still can have it there - but then also deposit it on a public repository.

The authors addressed the rest of the original reviewers' comments sufficiently

Comments to Reviewer:

We thank the reviewer for these comments. We address and reply to all individual comments below, with the reviewer's comments in bold and our responses underneath.

Reviewer #3 (Remarks to the Author):

(I was asked by the journal to assess author responses to another reviewer that were not able to look at responses again)

I went through the Hellenthal rebuttal and think they generally answered the reviewers' comments well.

I however have two additional comments - one minor and one I feel quite strong about:

Regarding point 13: The reviewer point out that Sengwer is a Nilo-Saharan language rather than a Bantu language. The authors replied that they used Nilotic as the linguistic group rather than Nilo-Saharan. My comment: Since Nilotic is a sub-group of Nilo-Saharan the authors should still indicate that major linguistic group. They can use Nilotic (Nilo-Saharan) for example.

Following this suggestion, we now refer to Sengwer as “NS Nilotic” (e.g. in Figure 3).

Regarding point 4: I strongly agree with the original reviewer that the deposition of the data in an appropriate national public repository is essential. The argument by the authors that David Reich's lab database is used by many studies does not really answer to the reviewers' concern. Keeping data on personal lab databases is not justified while public repositories (e.g. nih-dbGaP, ebi-EGA, ebi-ArrayExpress) are available at no charge. There is no guarantee that the Reich database will remain in existence for the foreseeable future. It might be shut down in 10 or 20 years and then access to data is no longer possible - as we have seen with many lab webpages and databases. For the long time preservation of data and information it is crucial that public repository deposition is a strong requirement for publication. Furthermore, since public repositories have the facility of establishing Data Access Committees that can ensure correct usage data - there is no reason not to make use of these resources. If the authors really want to have the data in the Reich database they still can have it there - but then also deposit it on a public repository.

Following this suggestion, we have made the data available on EGA (<https://ega-archive.org>), under accession number [EGAS00001005171](https://ega-archive.org/EGAS00001005171). We have now added the following “Data Availability” statement at the end of the main text:

Genotype data, birthplace information and self-reported group label, first language, second language and religious affiliation for newly genotyped individuals are available for non-commercial use at the European Genome-phenome Archive (EGA), which is hosted by the EBI and the CRG, under accession number [EGAS00001005171](https://ega-archive.org/EGAS00001005171).